# Proportional Response: Contextual Bandits for Simple and Cumulative Regret Minimization

**Sanath Kumar Krishnamurthy**
Management Science and Engineering
Stanford University
sanathsk@stanford.edu

**Ruohan Zhan**
Industrial Engineering and Decision Analytics
Hong Kong University of Science and Technology
rhzhan@ust.hk

**Susan Athey**
Graduate School of Business
Stanford University
athey@stanford.edu

**Emma Brunskill**
Computer Science Department
Stanford University
ebrun@cs.stanford.edu

## Abstract

In many applications, e.g. in healthcare and e-commerce, the goal of a contextual bandit may be to learn an optimal treatment assignment policy at the end of the experiment. That is, to minimize simple regret. However, this objective remains understudied. We propose a new family of computationally efficient bandit algorithms for the stochastic contextual bandit setting, where a tuning parameter determines the weight placed on cumulative regret minimization (where we establish near-optimal minimax guarantees) versus simple regret minimization (where we establish state-of-the-art guarantees). Our algorithms work with any function class, are robust to model misspecification, and can be used in continuous arm settings. This flexibility comes from constructing and relying on "conformal arm sets" (CASs). CASs provide a set of arms for every context, encompassing the context-specific optimal arm with a certain probability across the context distribution. Our positive results on simple and cumulative regret guarantees are contrasted with a negative result, which shows that no algorithm can achieve instance-dependent simple regret guarantees while simultaneously achieving minimax optimal cumulative regret guarantees.

## 1 Introduction

Learning and deploying personalized treatment assignment policies is crucial across domains such as healthcare and e-commerce [29, 25]. Traditional randomized control trials (RCTs), while foundational for policy learning [4, 7], can be inefficient and costly [30]. This motivates the study of adaptive sequential experimentation algorithms for the stochastic contextual bandit (CB) settings. The algorithm interacts with a finite sequence of users drawn *stochastically* from a fixed but unknown distribution. At each round, the algorithm receives a *context* (a user's feature vector), selects an action, and gets a corresponding reward. At the end of this adaptive experiment, the algorithm outputs a learned policy (mapping between contexts and actions).

Our algorithms are designed with the dual objectives of minimizing *simple regret* and *cumulative regret*. Simple regret quantifies the difference between the expected rewards achieved by the optimal policy and the policy learned at the conclusion of the experimental process. In contrast, cumulative regret encapsulates the summation of differences between the expected rewards generated by the

37th Conference on Neural Information Processing Systems (NeurIPS 2023).

optimal policy and the exploration policies employed at each sequential round of decision-making.[1] Although there are many settings where simple regret is an important consideration, the majority of research in the contextual bandit field has focused on the minimization of cumulative regret. To the best of our knowledge, there is no general-purpose computationally efficient algorithm for pure exploration objectives like simple regret minimization in the contextual bandit setting. Further, there has been relatively little work so far into algorithms that explore the trade-off between multiple objectives like cumulative regret and simple regret (though see [3, 9, 37] for studies that address this empirically or juxtapose minimizing cumulative regret with estimating treatment effects or arm parameters). Our work seeks to address these gaps. We show that there is a trade-off between simple and cumulative regret minimization (formalized later in a lower-bound result). To navigate this trade-off, we proposes a new algorithm called Risk Adjusted Proportional Response (RAPR) with a tuning parameter $\omega \in [1, K]$, which governs the weight placed on the two objectives.[2] The algorithm is general-purpose (in that it can address any user-specified reward and policy classes), ensures near-optimal guarantees, and is also computationally efficient.

**Types of guarantees.** In our analysis, we consider two different types of bounds on simple and cumulative regret, worst-case and instance-dependent guarantees. Here instance-dependent guarantees refer to bounds that surpass worst-case rates by exploiting instances with large gaps between the conditional expected rewards of the optimal and sub-optimal arms. Recent work by [11] has shown that it is not possible for contextual algorithms to have instance-dependent guarantees on cumulative regret (without suffering an exponential dependence on model class complexity); the authors instead develop algorithms that achieve minimax optimal (worst case optimal) cumulative regret guarantees (with square-root dependence on model class complexity). [26] developed the first general-purpose contextual bandit algorithm for pure exploration, and their algorithm achieved instance-dependent guarantees. They also show that instance-dependent best policy identification guarantees must come at the cost of worse than minimax optimal cumulative regret (discussed in detail later). We show a similar lower bound on cumulative regret for algorithms that achieve better instance-dependent simple regret guarantees, and propose the first family of algorithms that flexibly navigate such trade-offs.

**Overview of our guarantees.** The simple regret guarantees of RAPR are never worse than the minimax optimal rates (Theorem 2). Depending on the instance, RAPR achieves simple regret guarantees that are up to $O(1/\sqrt{\omega})$ times smaller compared to minimax optimal rates (Theorem 2). This improvement factor of $O(1/\sqrt{\omega})$ over minimax optimal rates is asymptotically achieved for instances where *realizability* holds (the reward model class is well specified) and the gap between the best and second best arm in terms of conditional expected reward is at least $\Delta > 0$ at every context (best-case instance in Theorem 2). RAPR provides these instance-dependent guarantees without the knowledge of any instance information. Unfortunately, the corresponding cumulative regret for the above instances is a factor of $O(\sqrt{\omega})$ times larger compared to minimax optimal rates (Theorem 1). The cumulative regret guarantees of our algorithm only degrade relative to the minimax optimal rate if the instance allows for better simple regret guarantees. Our lower bound (Theorem 3) considers the instances described above with $\Delta = 0.24$ (the gap between best and second best arm in terms of conditional expected reward). Theorem 3 shows that, for any algorithm that bounds the simple regret on these instances to $O(1/\sqrt{\omega})$ of the minimax optimal rates, its cumulative regret will be at least $\Omega(\sqrt{\omega})$ times the minimax optimal rates. RAPR thus achieves a near-optimal trade-off between guarantees on simple vs cumulative regrets when $T$ is large enough. The trade-off contrasts with non-contextual bandits, where successive elimination ensures improved (compared to minimax) instance-dependent guarantees for both simple and cumulative regret [10, 35].

**Types of CB algorithms.** Contextual bandit algorithms broadly fall into two categories: regression-free and regression-based. Regression-free algorithms create an explicit policy distribution, randomly choosing a policy for decision-making at any time-step [2, 5, 8, 26]. While these algorithms provide worst-case cumulative regret guarantees [2, 5, 8] or instance-dependent PAC guarantees for policy learning [26] without additional assumptions, they can be computationally intensive [11]: they require solving and storing the output of $\Omega(\text{poly}(T))$ cost-sensitive classification (CSC) problems [21] at every epoch (or update step). In contrast, regression-based algorithms [e.g., 1, 11, 34] construct a

---

[1]Our formal definition of simple regret compares against the best policy in our policy class, while our cumulative regret definition compares against the global optimal policy (induced by the true conditional expected reward model). The reason for this discrepancy is because we use a regression based approach (due to computational considerations) for constructing our exploration policies.

[2]$K$ is the number of arms for the finite arm setting.

conditional arm distribution using regression estimates of the expected reward, allowing for methods that need only solve $\mathcal{O}(1)$ regression or CSC problems at every epoch (or update step). Traditionally, these algorithms relied on realizability assumptions for optimal regret guarantees, but recent advances allow for misspecified reward model classes [6, 12, 22]. We develop regression-based algorithms and do not assume realizability. RAPR is the first general-purpose regression-based algorithm with attractive pure exploration (simple regret) guarantees.

**Overview of our algorithm.** We now describe the RAPR algorithm in more detail. We first define a surrogate objective for simple regret, the optimal cover, which is inversely proportional to the probability that the bandit exploration policy chooses the arm recommended by the unknown optimal policy. The optimal cover bounds the variance of evaluating the unknown optimal policy under our exploration policy. This surrogate objective can be minimized by appropriately designing our exploration policy/action selection kernels. To maintain the attractive computational properties of regression-based algorithms, RAPR does not construct an explicit distribution over policies as that distribution would have large support and would be computationally and memory intensive to maintain. Instead, the goal of minimizing the optimal cover is attained by directly constructing a distribution over arms for each arriving context. This in turn builds on a novel general-purpose uncertainty quantification at each context. Much of the existing literature constructs confidence intervals with point-wise guarantees, but existing approaches to constructing them rely on assumptions like linear realizability. For general function classes, these intervals may be too wide and are often computationally expensive to construct. To overcome this issue, we develop Conformal Arm Sets (CASs), which are a set of potentially optimal arms at each context. This uncertainty quantification is regression-based and computationally efficient to construct; it's general-purpose and shrinks at "fast rates" (with square-root dependency on expected squared error bounds for regression). Unfortunately, these sets come with some risk of not containing the arm recommended by the optimal policy at every context. Nevertheless, we can use this uncertainty quantification to construct a distribution over arms at each context that helps us minimize the optimal cover by balancing the benefits and risks of relying on these CASs. The unavoidable trade-off between our simple and cumulative regret guarantees is an artifact of these risky sets. Beyond allowing us to trade off simple and cumulative regret guarantees, the flexibility of the approach also helps us extend to continuous arm settings and allows us to handle model misspecification.

**Other Related Work.** Our work connects to the literature on pure exploration, extensively studied in MAB settings (see overview in [24]). [10, 15] study elimination-based algorithms for fixed confidence best-arm identification (BAI). [19, 32] study variants of Thompson Sampling with optimal asymptotic designs for BAI. [18] propose sequential halving for fixed budget BAI. Our algorithm provides fixed confidence simple regret guarantees and can be seen as a generalization of successive elimination [10] to the contextual bandit setting. The key technical difference is that it is often impossible to construct sub-gaussian confidence intervals on conditional expected rewards. The uncertainty quantification we use is similar to the notion of conformal prediction (see [36] for a detailed exposition). Until recently, pure exploration had been nearly unstudied in contextual bandits. [38] provide a static exploration algorithm that achieves the minimax lower bound on sample complexity for linear contextual bandits. [26] then provided the first algorithm with instance-dependent $(\epsilon, \delta)$-PAC guarantees for contextual bandits. This algorithm is regression-free (adapts techniques from [2]) and requires a sufficiently large dataset of offline contexts as input. Hence, unfortunately, it inherits high memory and runtime requirements [See 11, for a more detailed discussion]. However, these costs come with the benefit that their notion of instance dependence leverages structure not only in the true conditional expected reward (as in Theorem 2) but also in the policy class (similar to policy disagreement coefficient [13]). They also prove a negative result, showing that it is not possible for an algorithm to have instance-dependent $(0, \delta)$-PAC guarantees and achieve minimax optimal cumulative regret guarantees. Our hardness result is similar but complementary to their result, for we show a similar result for simple regret (rather than their $(0, \delta)$-PAC sample complexity).[3] Our work also recovers some cumulative regret guarantees for the continuous arm case [27, 40], with new guarantees on simple regret and robustness to misspecification. Note that our restriction to "slightly randomized" policies for the continuous arm case results in regret bounds with respect to a "slightly randomized" (smooth) benchmark [see 40, for smooth regret].

---

[3]In $(\epsilon, \delta)$ PAC sample complexity results, given an input $(\epsilon, \delta)$, the objective is to minimize the number of samples needed in order to output an $\epsilon$-optimal policy with probability at least $1 - \delta$ (a "fix accuracy, compute budget" setting). In contrast, in our simple regret case, we consider how to minimize the error $\epsilon$ as the number of samples increases.

## 1.1 Stochastic Contextual Bandits

We consider the stochastic contextual bandit setting, with context space $\mathcal{X}$, (compact) arm space $\mathcal{A}$, and a fixed but unknown distribution $D$ over contexts and arm rewards. $D_{\mathcal{X}}$ refers to the marginal distribution over contexts, and $T$ signifies the number of rounds or sample size. At each time $t \in [T]$[4], the environment draws a context $x_t$ and a reward vector $r_t \in [0,1]^{\mathcal{A}}$ from $D$; the learner chooses an arm $a_t$ and observes a reward $r_t(a_t)$. To streamline notation for discrete and continuous arm spaces, we consider a finite measure space $(\mathcal{A}, \Sigma, \mu)$ over the set of arms, with $K$ shorthand for $\mu(\mathcal{A})$.[5] For ease of exposition, we focus on the finite/discrete arm setting. Here $\mathcal{A} = [K]$ and $\mu$ is the count measure, and $\mu(S) = |S|$ for any $S \subseteq \mathcal{A}$. A (deterministic) policy $\pi$ maps contexts to *singleton arm sets* $\Sigma_1 := \{a | a \in \mathcal{A}\}$[6]. With some abuse of notation, we also let $\pi$ refer to the kernel given by $\pi(a|x) = I(a \in \pi(x))$. An action selection kernel (randomized policy) $p : \mathcal{A} \times \mathcal{X} \to [0,1]$ is a probability kernel that describes a distribution $p(\cdot|x)$ over arms at every context $x$. We let $D(p)$ be the induced distribution over $\mathcal{X} \times \mathcal{A} \times [0,1]$, where sampling $(x, a, r(a)) \sim D(p)$ is equivalent to sampling $(x, r) \sim D$ and then sampling $a \sim p(\cdot|x)$.

A reward model $f$ maps $\mathcal{X} \times \mathcal{A}$ to $[0,1]$, with $f^*(x, a) := \mathbb{E}_D[r_t(a)|x_t = x]$ denoting the true conditional expected reward model. Our algorithm works with a reward model class $\mathcal{F}$ and a policy class $\Pi$. For a given model $f$ and an action selection kernel $p$, we denote the expected instantaneous reward of $p$ with $f$ as $R_f(p)$. We write $R_{f^*}(p)$ as $R(p)$ to simplify notation when no confusion arises. The optimal policy associated with reward function $f$ is defined as $\pi_f$[7].

$$R_f(p) := \mathbb{E}_{x \sim D_{\mathcal{X}}} \mathbb{E}_{a \sim p(\cdot|x)} [f(x, a)], \text{ and } \pi_f \in \arg\max_{\pi} R_f(\pi).$$

The policy $\pi_f$ induced by $f \in \mathcal{F}$ is assumed to be within policy class $\Pi$ without loss of generality.[8] For any $S \subseteq \mathcal{A}$, with some abuse of notation, we let $f(x, S) = \int_{a \in S} f(x, a) d\mu(a)/\mu(S)$. Note that $\pi_f(x) \in \arg\max_{S \in \Sigma_1} f(x, S)$ for all $x$. The *regret* of a policy $\pi$ with respect to $f$ is the difference between the optimal value and the actual value of $\pi$, denoted as $\text{Reg}_f(\pi) := R_f(\pi_f) - R_f(\pi)$. Finally, we let $\pi^*$ denote the optimal policy in the class $\Pi$ and let $\text{Reg}_{\Pi}(\cdot)$ denote the regret with respect to $\pi^*$. That is, $\pi^* \in \arg\max_{\pi \in \Pi} R(\pi)$ and $\text{Reg}_{\Pi}(\pi) := R(\pi^*) - R(\pi)$.

**Objectives.** Contextual bandit algorithms adaptively construct action sampling kernels (exploration policies) $\{p_t\}_{t \in [T]}$ used to collect data over the $T$ rounds. At the end of the adaptive experiment, the adaptively collected data is used to learn a policy $\hat{\pi} \in \Pi$. We study two main objectives to measure quality of these outputs: [Objective 1] *Cumulative regret minimization.* Cumulative regret ($\text{CReg}_T$) is given by $\text{CReg}_T := \sum_{t=1}^T \text{Reg}_{f^*}(p_t)$. It compares the cumulative expected reward obtained during the experiment with the expected reward of the policy ($\pi_{f^*}$) induced by the true conditional expected reward ($f^*$). We seek to minimize cumulative regret which is equivalent to maximizing cumulative expected reward during the experiment. [Objective 2] *Simple regret minimization.* Simple regret is given by $\text{Reg}_{\Pi}(\hat{\pi})$. It compares the expected reward of the learnt policy $\hat{\pi} \in \Pi$ against the value of the optimal policy in the class $\Pi$. We seek to minimize simple regret which is equivalent to maximizing expected reward of the policy learnt at the end of the experiment. To understand the kind of exploration kernels ($\{p_t\}_{t \in [T]}$) that help with policy learning, we now identify a surrogate objective for simple regret (called optimal cover) that is in terms of the kernels used for exploration.

**Definition 1** (Cover). *Given a kernel $p$ and a policy $\pi$, we define the cover of policy $\pi$ under the kernel $p$ to be,*

$$V(p, \pi) := \mathbb{E}_{x \sim D_{\mathcal{X}}, a \sim \pi(\cdot|x)} \left[ \frac{\pi(a|x)}{p(a|x)} \right]. \tag{1}$$

*Additionally, for any pair of kernels $(p, q)$, we let $V(p, q) := \mathbb{E}_{x \sim D_{\mathcal{X}}, a \sim q(\cdot|x)}[q(a|x)/p(a|x)]$. Finally, we use the term optimal cover for kernel $p$ to refer to $V(p, \pi^*)$.*

The cover measures the quality of data collected under the action selection kernel $p$ for evaluating a given policy $\pi$ and bounds the variance of commonly used unbiased estimators for policy value

---

[4]For any $n \in \mathbb{N}^+$, we use notation $[n]$ to denote the set $\{1, ..., n\}$

[5]Here $\Sigma$ is a $\sigma$-algebra over $\mathcal{A}$ and $\mu$ is a bounded set function from $\Sigma$ to the real line.

[6]The introduction of $\Sigma_1$ is to allow for easy generalization to the continuous arm setting.

[7]subject to any tie-breaking rule.

[8]Note that $\Pi$ may contain policies that are not induced by models in the class $\mathcal{F}$.

[e.g., 2, 14, 39]. In particular, the cover under optimal policy $\frac{1}{T} \sum_{t=1}^{T} V(p_t, \pi^*)$ can be treated as a surrogate objective for simple regret minimization (proven in Appendix E.2), which is particularly instructional in designing our algorithm to minimize simple regret.

**Extending notation to continuous arms.** In the continuous arm setting, evaluating arbitrary deterministic policies can be infeasible without extra assumptions [28]. Thus, we focus on "slightly randomized" policies by generalizing $\Sigma_1$ to be the arm sets with measure one ($\Sigma_1 := \{S \in \Sigma | \mu(S) = 1\}$).[9] The granularity of these sets can be adjusted by scaling the finite measure $\mu$, which also affects the value of $K = \mu(\mathcal{A})$. We then continue defining policies be maps from $\mathcal{X}$ to $\Sigma_1$ and $\Pi$ is a class of such policies. We overload notation and define the induced kernel as $\pi(a|x) = I(a \in \pi(x))$, which is a valid definition since $\int_a I(a \in \pi(x))d\mu(a) = \mu(\pi(x)) = 1$. All the remaining definitions, including $R_f(\pi), \pi_f, \pi^*$ and $V(p, \pi)$, relied on these induced kernels and continue to hold. While there are some measure theoretic issues that remain to be discussed, we defer these details to Appendix A.

**Uniform sampling.** Our algorithm frequently selects an arm uniformly from a constructed set of arms. In the context of a set $S \subseteq \mathcal{A}$, uniform sampling refers to selecting an arm from the distribution $q(a) := I(a \in S)/\mu(S)$. This constitutes a probability measure since its integral over $\mathcal{A}$ equals 1. In the discrete arm setting, uniform sampling from a set $S \subseteq \mathcal{A}$ implies selecting an arm according to the distribution $I(a \in S)/|S|$.

## 1.2 Oracle Assumptions

Our algorithm relies on two sub-routines. For generality, we abstract away these sub-routines by stating them as oracle assumptions, for which we describe two oracles, EstOracle and EvalOracle, in Assumptions 1 and 2 respectively. The EstOracle sub-routine is for estimating conditional expected reward models (Assumption 1), and the EvalOracle sub-routine is for estimating policy values (Assumption 2) according to the true and estimated reward models.

These sub-routine tasks are supervised learning problems. Hence, the average errors for the corresponding tasks can be bounded in terms of the number of samples ($n$) and a confidence parameter ($\delta'$). The oracle assumptions specify the estimation rates. We let $\xi : \mathbb{N} \times [0, 1] \to [0, 1]$ denote the estimation rate for these oracles. For simplicity, we assume that they share the same rate and that $\xi(n, \delta')$ scales polynomially in $1/n$ and $\log(1/\delta')$. In order to simplify the analysis, we also require $\xi(n/3, \delta'/n^3)$ be non-increasing in $n$.[10] We now formally describe these oracle assumptions, starting with EstOracle.

**Assumption 1** (Estimation Oracle). *We assume access to a reward model estimation oracle (EstOracle) that takes as input an action selection kernel $p$, and $n$ independently and identically drawn samples from the distribution $D(p)$. The oracle then outputs an estimated model $\hat{f} \in \mathcal{F}$ such that for any $\delta' \in (0, 1)$, the following holds with probability at least $1 - \delta'$:*

$$\mathop{\mathbb{E}}_{x \sim D_{\mathcal{X}}} \mathop{\mathbb{E}}_{a \sim p(\cdot|x)} [(\hat{f}(x, a) - f^*(x, a))^2] \leq B + \xi(n, \delta')$$

*Where $B \geq 0$ is a fixed but unknown constant that may depend on the model class $\mathcal{F}$ and distribution $D$, but is independent of the action selection kernel $p$.*

In Assumption 1, the parameter $B$ measures the bias of model class $\mathcal{F}$; under realizability, $B$ equals 0. The function $\xi$ characterizes the estimation variance, which decreases with increasing sample size. *As long as the variance term (which shrinks as we gather more data) is larger than the fixed unknown bias ($B$), we have from Assumption 1 that the expected squared error for the estimated reward model is bounded by $2\xi$.* We use this bound on expected squared error to further bound how accurately the estimated reward model evaluates policies in the class $\Pi$ (Lemma 6). However, since $B$ is unknown, we need a test to detect when this policy evaluation bounds starts failing (which can only happen after the variance term gets dominated by the unknown bias term). To construct this test,

---

[9]Note that our restriction to "slightly randomized" policies for the continuous arm case results in regret bounds with respect to a "slightly randomized" (smooth) benchmark. Hence for the continuous arm case, our cumulative regret bounds translate to smooth regret bounds from [40] with $K = 1/h$. Where $h$ is the measure of smoothness in smooth regret (a leading objective for this setting).

[10]This ensures that $\xi_m$ defined in Lemma 1 is non-increasing in $m$ for any epoch schedule with increasing epoch lengths.

our algorithm relies on EvalOracle, which provides consistent independent policy value estimates and helps compare them with policy value estimates with respect to the estimated reward model.

**Assumption 2** (Evaluation Oracle). *We assume access to an oracle (EvalOracle) that takes as input an action selection kernel $p$, $n$ independently and identically drawn samples from the distribution $D(p)$, a set of $m$ models $\{g_i|i \in [m]\} \subseteq \mathcal{F}$, and another action selection kernel $q$. The oracle then outputs a policy evaluation estimator $\hat{R}$ of true policy value, and a set of $m$ policy evaluation estimators $\{\hat{R}_{g_i}|i \in [m]\}\}$ that estimate policy value with respect to the models $g_1, g_2, \ldots, g_m$ respectively. Such that for any $\delta' \in (0,1)$, the following conditions simultaneously hold with probability at least $1 - (m+1)\delta'$:*

- $|\hat{R}(\pi) - R(\pi)| \leq \sqrt{2V(p,\pi)\xi(n,\delta')} + 2\xi(n,\delta')/(\min_{(x,a) \in \mathcal{X} \times \mathcal{A}} p(a|x))$ *for all $\pi \in \Pi \cup \{q\}$.*

- $|\hat{R}_f(\pi) - R_f(\pi)| \leq \sqrt{2\xi(n,\delta')}$ *for all $\pi \in \Pi \cup \{q\}$ and for all $f \in \{g_i|i \in [m]\}$.*

When $\mathcal{F}$ and $\Pi$ are finite, one can construct oracles such that Assumptions 1 and 2 hold with $\xi(n,\delta') = \mathcal{O}(\log(\max(|\mathcal{F}|,|\Pi|)/\delta')/n)$. One example of such a construction is given by using empirical squared loss minimization for EstOracle, using inverse propensity scores (IPS) for estimating $R(\pi)$ in EvalOracle, and using the empirical average for estimating $R_f(\pi)$ in EvalOracle. The guarantees of these assumptions can be derived using Bernstein's inequality and union bounding. When $\mathcal{F}$ has pseudo-dimension [20] bounded by $d$ and $\Pi$ has the Natarajan-dimension bounded by $d$ [17, 16], one can construct oracles such that Assumptions 1 and 2 hold with $\xi(n,\delta') = \mathcal{O}(d\log(nK/\delta')/n)$.

## 2   Algorithm

---
**Algorithm 1** $\omega$ Risk Adjusted Proportional Response ($\omega$-RAPR)

---
**input:** Trade-off parameter $\omega \in [1, K]$, proportional response threshold $\beta_{\max} = 1/2$, and confidence parameter $\delta$ (used in definition of $\xi_m$).

1: Let $p_1(a|x) \equiv 1/\mu(\mathcal{A}) = 1/K$, $\hat{f}_1 \equiv 0$, $\alpha_1 = 3K$, $\tau_1 = 3$, and **safe** = **True**.
2: **for** epoch $m = 1, 2, \ldots$ **do**
3:     $\tau_m = 2\tau_{m-1}$.                                         ▷ Doubling epochs.
4:     **if safe then**
5:         **for** round $t = \tau_{m-1} + 1, \ldots, \tau_m$ **do**
6:             Observe context $x_t$, sample $a_t \sim p_m(\cdot|x_t)$, and observe $r_t(a_t)$.
7:         **end for**
8:         Let $S_m$ denote the data collected in epoch $m$.
9:         We split $S_m$ into three equally sized sets $S_{m,1}, S_{m,2}$ and $S_{m,3}$.
10:        Let $\hat{f}_{m+1} \leftarrow$ EstOracle$(p_m, S_{m,1})$, and let $C_{m+1}$ be given by Definition 2.
11:        Let $\eta_{m+1}$ be the solution to (5) and let $\alpha_{m+1} := 3K/\eta_{m+1}$.    ▷ $S_{m,2}$ is used here.
12:        Now let $p_{m+1}$ be given by (4).
13:        Let $\hat{R}_{m+1}, \{\hat{R}_{m+1,\hat{f}_i}|i \in [m+1]\} \leftarrow$ EvalOracle$(p_m, S_{m,3}, \{\hat{f}_i|i \in [m+1]\}, p_{m+1})$.
14:        **if** (2) does not hold. **then**
15:            $\hat{m}$, **safe** $\leftarrow m$, **False**.
16:        **end if**
17:     **else**
18:         **for** round $t = \tau_{m-1} + 1, \ldots, \tau_m$ **do**
19:             Observe context $x_t$, sample $a_t \sim p_{\hat{m}}(\cdot|x_t)$, and observe $r_t(a_t)$.
20:        **end for**
21:        Let $S_m$ denote the data collected in epoch $m$.
22:        Let $\hat{R}_{m+1}, \{\hat{R}_{m+1,\hat{f}_i}|i \in [\hat{m}]\} \leftarrow$ EvalOracle$(p_{\hat{m}}, S_m, \{\hat{f}_i|i \in [\hat{m}]\}, p_{\hat{m}})$.
23:     **end if**
24: **end for**

---

At a high level, our algorithm operates in two modes, indicated by a Boolean variable "**safe**". During mode one (**safe** = **True**), where estimated reward models are sufficiently accurate at evaluating policies in the class $\Pi$,[11] we use our estimated models to update our action selection kernel used

---

[11]Where the estimated reward models pass the misspecification test.

during exploration. During mode two (**safe = False**), where the condition for mode one no longer holds, we stop updating the action selection kernel used for exploration. Operationally our algorithm runs in epochs/batches indexed by $m$. Epoch $m$ begins at round $t = \tau_{m-1} + 1$ and ends at $t = \tau_m$, and we use $m(t)$ to denote the epoch index containing round $t$. We let $\hat{m}$ denote the critical epoch, at the end of which our algorithm changes mode (with "**safe**" being updated from "**True**" to "**False**"); we refer to $\hat{m}$ as the *algorithmic safe epoch*. For all rounds in epoch $m \leq \hat{m}$, our algorithm samples action using the action selection kernel $p_m$ defined later in (4). For $m > \hat{m}$, our algorithm samples action using $p_{\hat{m}}$ –the action selection kernel used in the algorithmic safe epoch $\hat{m}$.

We now describe the critical components of our algorithm. These include (i) data splitting and using oracle sub-routines; (ii) *misspecification tests*, which we use to identify the **safe**-mode switching epoch $\hat{m}$; and (iii) *conformal arm sets*, which presents a new form of uncertainty quantification that is critical in constructing $p_{m+1}$ at the end of each epoch $m \in [\hat{m}]$. Finally, we use these components to describe our final algorithm.

**Data splitting and oracle sub-routines.** Consider an epoch $m \in [\hat{m}]$. Let $S_m$ denote the set of samples collected in this epoch: $S_m = \{(x_t, a_t, r_t(a_t)) | t \in [\tau_{m-1}, \tau_m]\}$. Our algorithm splits $S_m$ into three equally-sized subsets: $S_{m,1}, S_{m,2}$ and $S_{m,3}$. Algorithm 1 outlines using these subsets and the oracles (described in Section 1.2) to estimate reward models and evaluate policies. Based on Assumptions 1 and 2, we bound the errors for these estimates in terms of $\xi_{m+1} = 2\xi((\tau_m - \tau_{m-1})/3, \delta/(16m^3))$, where $\delta$ is a specified confidence parameter. As we will see later, our algorithm relies on these bounds to test for misspecification and construct action selection kernels.

**Misspecification test.** We first discuss the need for our misspecification test. Note that Assumption 1 is flexible and allows our reward model class $\mathcal{F}$ to be misspecified. In particular, the squared error of our reward model estimate may depend on an unknown bias term $B$. To account for this unknown $B$, it is useful to center our analysis around the safe epoch $m^* := \arg\max\{m \geq 1 | \xi_{m+1} \geq 2B\}$, which denotes the last epoch where variance dominates bias. We show that for any epoch $m \in [m^*]$, the estimated reward model $\hat{f}_{m+1}$ is "sufficiently accurate" at evaluating the expected reward of any policy in $\Pi \cup \{p_{m+1}\}$. This property is critical in ensuring that the constructed action selection kernel $p_{m+1}$ has low exploration regret $\text{Reg}_{f^*}(p_{m+1})$ and a small optimal cover $(V(p_{m+1}, \pi^*))$. Since $B$ and $m^*$ are unknown, we need to test whether the estimated reward model is sufficiently accurate at evaluating these policies. When the test fails, the algorithm sets the variable "safe" to **False** and stops updating the action selection kernel used for exploration. The core idea for this test comes from [23] although its application to simple regret minimization is new, and the form of our test differs a bit. We now state our misspecification test (2). At the end of each epoch $m$, the test is passed if (2) holds:

$$
\max_{\pi \in \Pi \cup \{p_{m+1}\}} |\hat{R}_{m+1, \hat{f}_{m+1}}(\pi) - \hat{R}_{m+1}(\pi)| - \sqrt{\alpha_m \xi_{m+1}} \sum_{\bar{m} \in [m]} \frac{\hat{R}_{m+1, \hat{f}_{\bar{m}}}(\pi_{\hat{f}_{\bar{m}}}) - \hat{R}_{m+1, \hat{f}_{\bar{m}}}(\pi)}{40 \bar{m}^2 \sqrt{\alpha_{\bar{m}-1} \xi_{\bar{m}}}}
$$
$$
\leq 2.05 \sqrt{\alpha_m \xi_{m+1}} + 1.1 \sqrt{\xi_{m+1}},
$$
(2)

where $\alpha_{\bar{m}}$ empirically bounds $V(p_{\bar{m}}, \pi^*)$, the optimal cover for the action selection kernel used in epoch $\bar{m}$ (see (49)). The first term in (2) measures how well the estimated reward model $\hat{f}_{m+1}$ evaluates the policy $\pi$, and the second term accounts for under-explored policies (policies that have high regret under the reward model $\hat{f}_m$ would be less explored in epoch $m$).

**Conformal arm sets.** We proceed to introduce the notion of *conformal arm sets* (CASs), based on which we construct the action selection kernels employed by our algorithms. At the beginning of each epoch $m$, we construct CASs, denoted as $\{C_m(x, \zeta) | x \in \mathcal{X}, \zeta \in [0,1]\}$; here $\zeta$ controls the probability with which the set $C_m$ contains the optimal arm. The construction of these sets rely on the models $(\hat{f}_1, \ldots, \hat{f}_m)$ estimated from data up to epoch $m - 1$, as defined below.

**Definition 2** (Conformal Arm Sets). *Consider $\zeta \in (0,1)$. At epoch $m$, for context $x$, the arm set $C_m(x, \zeta)$ is given by (3).*

$$
C_m(x, \zeta) := \pi_{\hat{f}_m}(x) \bigcup \bar{C}_m(x, \zeta), \quad \bar{C}_m(x, \zeta) := \bigcap_{\bar{m} \in [m]} \tilde{C}_{\bar{m}}\left(x, \frac{\zeta}{2\bar{m}^2}\right),
$$
$$
\tilde{C}_{\bar{m}}(x, \zeta') := \left\{ a : \hat{f}_{\bar{m}}(x, \pi_{\hat{f}_{\bar{m}}}(x)) - \hat{f}_{\bar{m}}(x, a) \leq \frac{20 \sqrt{\alpha_{\bar{m}-1} \xi_{\bar{m}}}}{\zeta'} \right\} \quad \forall \bar{m} \in [m], \zeta' \in (0,1).
$$
(3)

Similar to conformal prediction (CP) [36, 33], CASs have marginal coverage guarantees. We show that with high probability, we have $\pi^*(x)$ lies in $C_m(x, \zeta)$ with probability at least $1 - \zeta$ over the context distribution. That is, $\Pr_{x \sim D_{\mathcal{X}}}(\pi^*(x) \in C_m(x, \zeta)) \geq 1 - \zeta$ with high-probability (see Appendix E.1). However, there is also a key technical difference. While CP provides coverage guarantees for the conditional random outcome, CASs provide coverage guarantees for $\pi^*(x)$ – which is not a random variable given the context $x$. Hence, intervals estimated by CP need to be wide enough to account for conditional outcome noise, whereas CASs do not. CASs also have several advantages compared to pointwise confidence intervals used in UCB algorithms. First, CASs are computationally easier to construct. Second, CAS widths have a polynomial dependency on model class complexity, whereas pointwise intervals may have an exponential dependence for some function classes [see lower bound examples in 13]. Third, pointwise intervals require realizability, whereas the guarantees of CASs hold even without realizability (as long as the misspecification test in (2) holds). However, it's important to remember that these benefits of CASs come with the risk of only covering $\pi^*(x)$ marginally over the context distribution – that is, these sets may not contain $\pi^*(x)$ at all $x$.

**Risk Adjusted Proportional Response Algorithm.** We now describe the design of our algorithm, which is summarized in Algorithm 1. The algorithm depends on the following input parameters: $\omega \in [1, K]$ which controls the trade-off between simple and cumulative regret, the proportional response threshold $\beta_{\max} = 1/2$, and confidence parameter $\delta$. The algorithm also computes $\eta_{m+1}$ (risk adjustment parameter for $p_{m+1}$), $\alpha_{m+1}$ (empirical bound on optimal cover for $p_{m+1}$), and $\lambda_{m+1}(\cdot)$ (empirical bound on average CAS size). At the end of every epoch $m \in [\hat{m}]$, we construct the action selection kernel $p_{m+1}$ given by (4).

$$p_{m+1}(a|x) = \frac{(1 - \beta_{\max})I[a \in C_{m+1}(x, \beta_{\max}/\eta_{m+1})]}{\mu\big(C_{m+1}(x, \beta_{\max}/\eta_{m+1})\big)} + \int_0^{\beta_{\max}} \frac{I[a \in C_{m+1}(x, \beta/\eta_{m+1})]}{\mu\big(C_{m+1}(x, \beta/\eta_{m+1})\big)} \mathrm{d}\beta.$$
(4)

At any context $x$, sampling arm $a$ from $p_{m+1}(\cdot|x)$ is equivalent to the following. Sample $\beta$ uniformly from $[0, 1]$, then sample arm $a$ uniformly from the set $C_{m+1}(x, \min(\beta_{\max}, \beta)/\eta_{m+1})$. A small $\beta$ results in a larger CAS and a higher probability of containing the optimal arm for the sampled context. However, uniformly sampling an arm from a larger CAS also implies a lower probability on every arm in the set. Sampling $\beta$ uniformly allows us to respond proportionately to the risk of not sampling the optimal arm while enjoying the benefits of smaller CASs. We refer to this as the *Proportional Response Principle*.

Similarly note that, a larger risk-adjustment parameter $\eta_{m+1}$ encourages reliance on less risky albeit larger CASs. We want to choose $\eta_{m+1}$ to tightly bound the the optimal cover (surrogate for simple regret), subject to cumulative regret constrains imposed by the trade-off parameter $\omega$. To do this, we first let $\lambda_{m+1}(\eta)$ be a high-probability empirical upper bound on $E_{x \sim D_{\mathcal{X}}}[\mu(C_{m+1}(x, \beta_{\max}/\eta))]$. Hence, using (49), we can upper bound the optimal cover ($V(p_{m+1}, \pi^*)$) by $\frac{\lambda_{m+1}(\eta_{m+1})}{1 - \beta_{\max}} + \frac{K}{\eta_{m+1}}$. Our choice of $\eta_{m+1}$ approximately minimizes this upper bound on the optimal cover, by choosing the largest feasible $\eta \in [\eta_m, \sqrt{\omega K / \alpha_m}]$ such that $\lambda_{m+1}(\eta) \leq \frac{K}{\eta}$ (see (5)). Note that this choice of $\eta_{m+1}$ balances the risk of a small $\eta$ (large $\frac{K}{\eta}$) with the benefits of a small $\lambda_{m+1}(\eta)$ (small $\frac{\lambda_{m+1}(\eta)}{1 - \beta_{\max}}$).

$$\lambda_{m+1}(\eta) := \min\left(1 + \frac{1}{|S_{m,2}|} \sum_{t \in S_{m,2}} \mu(\bar{C}_{m+1}\big(x, \frac{\beta_{\max}}{\eta}\big)) + \sqrt{\frac{K^2 \ln(8|S_{m,2}|(m+1)^2/\delta)}{2|S_{m,2}|}}, K\right),$$

$$\eta_{m+1} \leftarrow \max\left\{\eta_m, \max\left\{\eta = \frac{|S_{m,2}|}{n}\bigg| n \in [|S_{m,2}|], \eta \leq \sqrt{\frac{\omega K}{\alpha_m}}, \lambda_{m+1}(\eta) \leq \frac{K}{\eta}\right\}\right\}.$$
(5)

With $\eta_{m+1}$ chosen, the action selection kernel $p_{m+1}$ is decided. Now let $\alpha_{m+1} = 3K/\eta_{m+1}$, which is a high-probability empirical upper bound on $V(p_{m+1}, \pi^*)$. We then use $\alpha_{m+1}$ at the end of epoch $m + 1$ to construct CASs, compute the risk-adjustment parameter, and test for misspecification.

**Computation.** We have $\mathcal{O}(\log(T))$ epochs. At the end of any epoch $m \in [\hat{m}]$, we solve three optimization problems. The first is for estimating $\hat{f}_{m+1}$, which often reduces to empirical squared loss minimization and is computationally tractable for several function classes $\mathcal{F}$. The second is for computing the risk-adjustment parameter in (5) which can be solved via binary search. The third is for the misspecification test in (2), which can be solved via two calls to a cost-sensitive classification (CSC) solver (don't need this when assuming realizability, further if we only care about cumulative

regret, sufficient to use the simpler test in [22]). Finally, to learn a policy $\hat{\pi}$ at the end of $T$ rounds, we need to solve (8) using a CSC solver (under realizability we can set $\hat{\pi} = \pi_{\hat{f}_{m(T)-1}}$). Hence, overall, $\omega$-RAPR makes exponentially fewer calls to solvers compared to regression-free algorithms like [26].

## 3    Main Results

Our algorithm/analysis/results hold for both the discrete and continuous arm cases. As discussed before, minimizing optimal cover helps us ensure improved simple regret guarantees. Hence $\alpha_m \in [1, 3K]$ (the high-probability empirical upper bound on the optimal cover $V(p_m, \pi^*)$) will play a crititcal role thoughout this results section. We start with stating our cumulative regret bounds.

**Theorem 1.** *Suppose Assumptions 1 and 2 hold. Then with probability $1 - \delta$, $\omega$-RAPR attains the following cumulative regret guarantee. Here $\xi_{m+1} = \xi((\tau_m - \tau_{m-1})/3, \delta/(16m^3)$ for all $m$.*

$$CReg_T \leq \tilde{\mathcal{O}}\left( \sum_{t=\tau_1+1}^{T} \sqrt{\frac{K}{\alpha_{m(t)}} \frac{\alpha_{m(t)-1}}{\alpha_{m(t)}}} \left( \sqrt{KB} + \sqrt{K\xi_{m(t)}} \right) \right) \tag{6a}$$

$$\leq \tilde{\mathcal{O}}\left( \sqrt{\omega KBT} + \sum_{t=\tau_1+1}^{T} \sqrt{\omega K\xi_{m(t)}} \right). \tag{6b}$$

*Where we use $\tilde{\mathcal{O}}$ to hide terms logarithmic in $T, K, \xi(T, \delta)$.*

We start with discussing (6b). The first part $\sqrt{\omega KBT}$ comes from the bias of the regression oracle with model class $\mathcal{F}$ and will vanish under the realizability assumption. The second part $\sum_{t=\tau_1+1}^{T} \sqrt{\omega K\xi_{m(t)}}$, when setting $\omega = 1$, recovers near-optimal (upto logarithmic factors) minimax cumulative regret guarantees for common model classes, as demonstrated by the following examples.

**Corollary 1.** *We consider $\omega$-RAPR with appropriate oracles in the following cases and let $B$ denote the corresponding bias terms. When $\mathcal{F}$ and $\Pi$ are finite, $CReg_T \leq \tilde{\mathcal{O}}(\sqrt{\omega KBT} + \sqrt{\omega KT \log(\max(|\mathcal{F}|, |\Pi|)/\delta)})$ with probability at least $1 - \delta$. When $\mathcal{F}$ has a finite pseudo dimension $d$, $\Pi$ has a finite Natarajan dimension $d$, and $\mathcal{A}$ is finite, $CReg_T \leq \tilde{\mathcal{O}}(\sqrt{\omega KBT} + \sqrt{\omega KTd \log(TK/\delta)})$ with probability at least $1 - \delta$. Note that under realizability ($B = 0$), 1-RAPR achieves near-optimal minimax cumulative guarantees.*

In (6a), we observe that the multiplicative $\sqrt{\omega}$ cost to cumulative regret is only incurred if the empirical bound on optimal cover ($\alpha_m \in [1, 3K]$) can get small. That is, our cumulative regret bounds degrade only if our algorithm better bounds the optimal cover and thus ensures better simple regret guarantees. [12] We now provide instance dependent simple regret guarantees for our algorithm.

**Theorem 2.** *Suppose Assumptions 1 and 2 hold. For some $(\lambda, \Delta, A) \in [0, 1] \times (0, 1] \times [1, K]$, consider instances where for $1 - \lambda$ fraction of contexts at most $A$ arms are $\Delta$ optimal (i.e. (7) holds).*

$$\mathbb{P}_{x \sim D_{\mathcal{X}}} \left( \mu\left(\{a \in \mathcal{A} : f^*(x, \pi_{f^*}(x)) - f^*(x, a) \leq \Delta\}\right) \leq A \right) \geq 1 - \lambda. \tag{7}$$

*Let $m' = \min(\hat{m}, m(T)) - 1$. Let the learned policy $\hat{\pi}$ be given by (8) (equivalent to variance penalized policy optimization).*

$$\hat{\pi} \in \arg\max_{\pi \in \Pi} \hat{R}_{m(T)}(\pi) - \frac{1}{2}\sqrt{\alpha_{m'}\xi_{m(T)}} \sum_{\bar{m} \in [m']} \frac{\hat{R}_{m(T),\hat{f}_{\bar{m}}}(\pi_{\hat{f}_{\bar{m}}}) - \hat{R}_{m(T),\hat{f}_{\bar{m}}}(\pi)}{40\bar{m}^2\sqrt{\alpha_{\bar{m}-1}\xi_{\bar{m}}}}. \tag{8}$$

*Then with probability $1 - \delta$, $\omega$-RAPR has the following simple regret bound when $T$ samples.*

$$Reg_{\Pi}(\hat{\pi}) \leq \mathcal{O}\left( \sqrt{\alpha_{m'}\xi_{m(T)}} \right)$$

$$\leq \mathcal{O}\left( \sqrt{\xi_{m(T)} \min\left( K, A + K\lambda + \frac{K}{\omega} + \frac{K^{3/2}\omega^{1/2}}{\Delta}\sqrt{\xi_{\min(m^*,m(T)-1)-\lceil\log_2\log_2(K)\rceil}} \right)} \right).$$

---

[12]For large $t$, once our bounds on optimal cover can't be significantly improved, we have $\alpha_{m(t)}/\alpha_{m(t)-1} = O(1)$. Hence for large $T$, our cumulative regret is a factor of $\sqrt{K/\alpha_{m(T)+1}}$ larger than the near optimal minimax guarantees. As we will see in Theorem 3, this multiplicative factor is unavoidable.

Under (7), we can only argue that the expected (over context distribution) measure of $\Delta$ optimal arms is at most $(1 - \lambda)A + K\lambda = O(A + K\lambda)$. Hence for large $T$, the best we can hope for is instance-dependant simple regret guarantees that shrink/improve over minimax guarantees by a factor of $\mathcal{O}(\sqrt{(A + K\lambda)/K})$. We show that this is guaranteed by Theorem 2. Suppose $\omega = K$, $\mathcal{F}$ has a finite pseudo dimension bounded by $d$, and $\Pi$ has a finite Natarajan dimension bounded by $d$. The simple regret guarantee of Theorem 2 reduces to $\tilde{\mathcal{O}}(\min((\sqrt{Kd/T}, \sqrt{(A + K\lambda)d/T} + (K/\sqrt{\Delta})\sqrt{d/T}\sqrt[4]{B + d/T}))$. When the reward model estimation bias $B$ is small enough, the term $(K/\sqrt{\Delta})\sqrt{d/T}\sqrt[4]{B + d/T}$ is dominated by the remaining terms for large $T$. Hence, in this case, we get a simple regret bound of $\tilde{\mathcal{O}}(\sqrt{(A + K\lambda)d/T})$ for large $T$. As promised, this improves upon the minimax guarantees by a factor of $\mathcal{O}(\sqrt{(A + K\lambda)/K})$.

Note that Theorem 1 guarantees are better for $\omega$ closer to 1 whereas Theorem 2 guarantees are better for $\omega$ closer to $K$. Hence these theorems show a tradeoff between the cumulative and simple regret guarantees for $\omega$-RAPR. Theorem 3 shows that improving upon minimax simple regret guarantees for instances satisfying (7) may come at the unavoidable cost of worse than minimax optimal cumulative regret guarantees. This contrasts with non-contextual bandits, where successive elimination ensures improved (compared to minimax) gap-dependent guarantees for both simple and cumulative regret.

**Theorem 3.** *Given parameters $K, F, T \in \mathbb{N}$ and $\phi \in [1, \infty)$. There exists a context space $\mathcal{X}$ and a function class $\mathcal{F} \subseteq (\mathcal{X} \times \mathcal{A} \to [0, 1])$ with $K$ actions such that $|\mathcal{F}| \leq F$ and the following lower bound on cumulative regret holds:*

$$\inf_{\mathbf{A} \in \Psi_\phi} \sup_{D \in \mathcal{D}} \mathbb{E}_D \left[ \sum_{t=1}^{T} \left( r_t(\pi^*(x_t)) - r_t(a_t) \right) \right] \geq \tilde{\Omega}\left( \sqrt{\frac{K}{\phi}} \sqrt{KT \log F} \right)$$

*Here $(a_1, \ldots a_T)$ denotes the actions selected by an algorithm $A$. $\mathcal{D}$ denotes the set of environments such that $f^* \in \mathcal{F}$ and (7) hold with $(A, \lambda, \Delta) = (1, 0, 0.24)$. $\Pi$ denotes policies induced by $\mathcal{F}$. $\Psi_\phi$ denotes the set of CB algorithms that run for $T$ rounds and output a learned policy with a simple regret guarantee of $\sqrt{\phi \log F / T}$ for any instance in $\mathcal{D}$ with confidence at least $0.95$, i.e., $\Psi_\phi := \{A : \mathbb{P}(Reg(\hat{\pi}_\mathbf{A}) \leq \sqrt{\phi \log F / T}) \geq 0.95 \text{ for any instance in } \mathcal{D}\}$. Finally, $\tilde{\Omega}(\cdot)$ hides factors logarithmic in $K$ and $T$.*

**Near optimal trade-off of RAPR.** Note that the environments constructed in Theorem 3 satisfy $f^* \in \mathcal{F}$ with $\max(|\mathcal{F}|, |\Pi|) \leq F$ and also satisfy (7) with $(A, \lambda, \Delta) = (1, 0, 0.24)$. With appropriate oracles, Assumptions 1 and 2 are satisfied with $B = 0$ (i.e. $m^* = \infty$) and $\xi(n, \delta') = \mathcal{O}(\log(F/\delta')/n)$. Hence for large enough $T$, $\omega$-RAPR achieves a simple regret bound of $\tilde{\mathcal{O}}(\sqrt{(K/\omega) \log F / T})$ with probability at least $0.95$ and thus is a member of $\Psi_\phi$ for some $\phi = \tilde{\mathcal{O}}(K/\omega)$. Theorem 3 lower bounds the cumulative regret of such algorithms by $\tilde{\Omega}(\sqrt{K/\phi}\sqrt{KT \log F}) = \tilde{\Omega}(\sqrt{\omega}\sqrt{KT \log F})$. Up to logarithmic factors, this matches the cumulative regret upper bound for $\omega$-RAPR. Re-emphasizing that the trade-off observed in Theorems 1 and 2 is near optimal for large $T$.

**Simulations.** To demonstrate the computational tractability of our approach, we ran a simulation on setting within a $\mathbb{R}^2$ context space, eight arms, linear models, and an exploration horizon of 5000. Our algorithms ran in less than 9 seconds on a Macbook M1 Pro. We also compare with other baselines on simple/cumulative regret. See Appendix E.4 for details.

**Conclusion.** We develop Risk Adjusted Proportional Response (RAPR), a computationally efficient regression-based contextual bandit algorithm. It is the first contextual bandit algorithm capable of trading-off worst-case cumulative regret guarantees with instance-dependent simple regret guarantees. The versatility of our algorithm allows for general reward models, handles misspecification, extends to finite and continuous arm settings, and allows us to choose the trade-off between simple and cumulative regret guarantees. The key ideas underlying RAPR are conformal arm sets (CASs) to quantify uncertainty, proportional response principle for cumulative regret minimization, optimal cover as a surrogate for simple regret, and risk adjustment for better bounds on the optimal cover. [13]

**Limitations.** A limitation of our approach is that we do not utilize the structure of the policy class being explored. Further refining CASs with other forms of uncertainty quantification that leverage such structure can lead to significant improvements, and potentially avoid trade-offs between simple/cumulative regret when policy class structure allows for it.

---

[13]S.A. and S.K.K. are grateful for the support provided by Golub Capital Social Impact Lab and the ONR grant N00014-19-1-2468. E.B. is grateful for the support of NSF grant 2112926.

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
