# A Expanded Notations

We start with expanding our notation from Section 1.1 to include notation helpful for our proofs and expand to the continuous arm setting.

**Measure over arms.** To recap, our algorithm and analysis adapt to both discrete and continuous arm spaces, where we consider a finite measure space $(\mathcal{A}, \Sigma, \mu)$ over the set of arms (i.e. $\mu(\mathcal{A})$ is finite) to unify the notation.[14] As short hand, we use $K$ in lieu of $\mu(\mathcal{A})$. We let $\Sigma_1$ be a set of arms in $\Sigma$ with measure one, i.e. $\Sigma_1 := \{S \in \Sigma | \mu(S) = 1\}$.

**Policies.** Let $\tilde{\Pi}$ denote the universal set of policies. That is, $\tilde{\Pi}$ is the set of all functions from $\mathcal{X}$ to $\Sigma_1$. The policy class $\Pi$ is a subset of $\tilde{\Pi}$. We use $\pi(x)$ to denote the set of arms given $x \in \mathcal{X}$ and use $p_\pi(a|x) = \mathbb{I}(a \in \pi(x))$ to denote the induced probability measure over arms at $x$.[15] With some abuse of notation, we use the notation $\pi(a|x)$ in lieu of $p_\pi(a|x)$. Below is the elaboration of our notation to both discrete and continuous arm spaces.

- *Discrete arm space.* We choose $\mu$ to be the count-measure, where $\mu(S) = |S|$ for any $S \subseteq \mathcal{A}$ and $\mu(\mathcal{A}) = K$. In this case, $\Sigma_1$ contains singleton arm sets, and $\tilde{\Pi}$ denotes deterministic policies from $\mathcal{X}$ to $\mathcal{A}$ where each policy maps a context to an action.
- *Continuous arm space.* We choose $\mu$ to any finite measure, where $\mu(S) = \int_S d\mu(a)$ for any $S \subseteq \mathcal{A}$, and in particular $\mu(\mathcal{A}) = K$. In this case, $\Sigma_1$ contains arm sets that may have an infinite number of arms but with total measure be 1 with respect to $\mu$.

**Space of action selection kernels.** In this paper, we will always define our action selection kernels with respect to the reference measure $\mu$, that is $p(S|x) = \int_{a \in S} p(a|x) d\mu$ for any $S \in \Sigma$ and $x \in \mathcal{X}$. Based on the notation in Section 1.1, for any kernel $p$, we let $\mathrm{Reg}_f(p) = R_f(\pi_f) - R_f(p)$.

Now let $\mathcal{P}$ denote the set of action selection kernels such that $p(a|x) \leq 1$ for all $(x, a) \in \mathcal{X} \times \mathcal{A}$, and in particular, we have the policy class $\tilde{\Pi} \subset \mathcal{P}$. We note that all action selection kernels $(p)$ considered in this paper belong to the set $\mathcal{P}$, allowing our analysis to rely on the fact that $p(\cdot|\cdot) \leq 1$.

Note that, $\mathrm{Reg}_f(p)$ is non-negative for any $p \in \mathcal{P}$. To see this, consider any context $x$. Recall that $\pi_f(x) \in \arg\max_{S \in \Sigma_1} f(x, S)$ for all $x$. Since $p(\cdot|\cdot) \leq 1$ for any $p \in \mathcal{P}$, we have $\int_{a \in \mathcal{A}} p(a|x) f(x, a) d\mu$ is maximized when $p(a|x) = 1$ for all $a \in \pi_f(x)$. That is, $f(x, \pi_f(x)) = \max_{p \in \mathcal{P}} \mathbb{E}_{a \sim p(\cdot|x)}[f^*(x, a)]$. Hence, $\max_{p \in \mathcal{P}} R_f(p) = R_f(\pi_f)$, so $\mathrm{Reg}_f(p)$ is non-negative for any $p \in \mathcal{P}$.

**Connection to smooth regret [40].** Recall that we define cumulative regret as $\mathrm{CReg}_T := \sum_{t=1}^T \mathrm{Reg}_{f^*}(p_t)$, which measures regret w.r.t the benchmark $R_{f^*}(\pi_{f^*}) = \max_{p \in \mathcal{P}} R_{f^*}(p)$. As discussed earlier, [40] shows that smooth regret bounds are stronger than several other definitions of cumulative regret in the continuous arm setting [e.g., 27]. Hence to show that our bounds are comparable/competitive for the continuous arm setting, we argue that our definition of cumulative regret ($\mathrm{CReg}_T$) is equivalent to the definition of smooth regret in [40].

Let the loss vectors $l_t$ in [40] be given by $-r_t$. Let the smoothness parameter $h$ in [40] be given by $1/K$. And, let the base probability measure in [40] be given by $\mu/K$. Then, our benchmark ($\max_{p \in \mathcal{P}} R_{f^*}(p)$) is equal to the smooth benchmark ($\mathbb{E}[\mathrm{Smooth}_h(x)]$) considered in [40]. Hence, our definition of cumulative regret ($\mathrm{CReg}_T$) is equal to smooth regret ($\mathrm{Reg}_{\mathrm{CB},h}(T)$) when the loss, smoothness parameter, and base probability measure are given as above. This shows the equivalence in our definitions.

Hence our near-optimal cumulative regret bounds (with $\omega = 1$) recover several existing results for the stochastic contextual bandit setting up to logarithmic factors using only offline regression oracles. Our algorithm also handles reward model misspecification and does not assume realizability. We also provide instance-dependent simple regret bounds (for larger choices of $\omega$). The parameter $\omega$ allows us to trade-off between simple and cumulative regret bounds.

**Measure theoretic issues with continuous arms.** To avoid measure-theoretic issues, we require that for all models $f \in \mathcal{F} \cup \{f^*\}$, all contexts $x \in \mathcal{X}$, and all real numbers $z \in \mathbb{R}$, we have the level set of arms $\{a | f(x, a) \leq z\}$ must lie in $\Sigma$. That is the reward models $f \in \mathcal{F} \cup \{f^*\}$ are measurable at

---

[14]Here $\Sigma$ is a $\sigma$-algebra over $\mathcal{A}$ and $\mu$ is a bounded set function from $\Sigma$ to the real line.

[15]Note that for any $\pi \in \tilde{\Pi}$, we have $\int_{a \in \mathcal{A}} p_\pi(a|x) d\mu = \mu(\pi(x)) = 1$ at any $x \in \mathcal{X}$.

| | Environment distribution |
|---|---|
| $\mathcal{X}, \mathcal{A}$ | set of contexts and set of arms (respectively). |
| $D$ | joint distribution over contexts and arm rewards. |
| $D_{\mathcal{X}}$ | marginal distribution over contexts. |
| $D(p)$ | distribution over $\mathcal{X} \times \mathcal{A} \times [0,1]$ induced by action selection kernel $p$. |
| $f^*$ | true conditional expected reward, $f^*(x,a) := \mathbb{E}_D[r_t(a)|x_t = x]$. |
| | Measure space over arm sets |
| $(\mathcal{A}, \Sigma, \mu)$ | measure space over the set of arms with $K := \mu(\mathcal{A})$. |
| $\Sigma_1$ | set of measurable arm sets with measure one, $\Sigma_1 := \{S \in \Sigma | \mu(S) = 1\}$. |
| $\tilde{\Pi}$ | set of all policies (functions from $\mathcal{X}$ to $\Sigma_1$). |
| $\mathcal{P}$ | set of kernels such that $p(a|x) \leq 1$ for all $(x,a) \in \mathcal{X} \times \mathcal{A}$. |
| | Algorithm inputs |
| $\beta_{\max}$ | proportional response threshold ($\beta_{\max} = 1/2$). |
| $\omega$ | trade-off parameter ($\omega \in [1, K]$). |
| $\delta$ | confidence parameter ($\delta \in (0,1)$). |
| $\mathcal{F}, \Pi$ | give reward model class and policy class. |
| | Policy value and optimal policy |
| $R_f(p)$ | $:= \mathbb{E}_{x \sim D_{\mathcal{X}}} \mathbb{E}_{a \sim p(\cdot|x)}[f(x,a)]$, denotes the value of kernel $p$ under model $f$. |
| $R(p)$ | $:= R_{f^*}(p)$, denotes the true value of kernel $p$. |
| $\pi_f$ | $\in \arg\max_{\pi \in \tilde{\Pi}} R_f(\pi)$, denotes the best universal policy under model $f$. |
| $\pi^*$ | $\in \arg\max_{\pi \in \Pi} R(\pi)$, denotes the best policy in the class $\Pi$. |
| | Regret and cover |
| $\mathrm{Reg}_f(p)$ | $:= R_f(\pi_f) - R_f(p)$, denotes the regret of kernel $p$ under model $f$. |
| $R(p)$ | $:= R_{f^*}(p)$, denotes the true value of kernel $p$. |
| $\mathrm{Reg}_\Pi(p)$ | $:= R_f(\pi_f) - R_f(p)$, denotes the regret of kernel $p$ under model $f$. |
| $V(p,q)$ | cover $V(p,q) := \mathbb{E}_{x \sim D_{\mathcal{X}}, a \sim q(\cdot|x)}\left[q(a|x)/p(a|x)\right]$. |
| | Epochs |
| $m$ | epoch index. |
| $\xi_{m+1}$ | $:= 2\xi((\tau_m - \tau_{m-1})/3, \delta/(16m^3))$, where $\xi$ is given in Section 1.2. |
| $m^*$ | safe epoch, last epoch where $\xi_{m+1} \geq 2B$ (variance dominates bias). |
| $\hat{m}$ | last epoch that starts with **safe** set as **True**. |
| | Algorithmic parameters |
| $\hat{f}_{m+1} \in \mathcal{F}$ | fitted reward model via regressions on samples collected in epoch $m$. |
| $C_{m+1}$ | conformal arm set defined in Definition 2. |
| $\eta_{m+1}$ | risk adjustment parameter (5). |
| $\alpha_{m+1}$ | empirical bound on optimal cover ($\alpha_{m+1} = \frac{3K}{\eta_{m+1}}$). |
| $p_{m+1}$ | action selection kernel corresponding to epoch $m+1$ defined in (4). |
| $U_{m+1}$ | $:= 20\sqrt{\alpha_m \xi_{m+1}}$. |

Table 1: Table of notations

every context $x$ with the Lebesgue measure on the range of $f(x, \cdot)$ and the measure $(\mathcal{A}, \Sigma, \mu)$ on the domain of $f(x, \cdot)$. We note that this isn't a strong condition and usually trivially holds.

Moreover, we require an additional condition as follows to simplify our arguments and allow for easy construction of our uncertainty sets (see Definition 2). We require that for all models $f \in \mathcal{F}$ and all contexts $x \in \mathcal{X}$, we have $f(x, \pi_f(x))$ is equal to $\max_{a \in \mathcal{A}} f(x,a)$. This condition trivially holds for the finite-arm setting with $\mu$ as a count measure. For the continuous arm setting, this condition follows from requiring $\arg\max_{a \in \mathcal{A}} f(x,a)$ lies in $\Sigma$ and has measure of at least one.

**Additional notation.** For notational convenience, we let $U_m = 20\sqrt{\alpha_{m-1}\xi_m}$ for any epoch $m$. Note that by construction ((5) and $\alpha_m = 3K/\eta_m$) $\alpha_m$ is non-increasing in $m$. Further, from the conditions in Section 1.2, we have $\xi_{m+1} = 2\xi((\tau_m - \tau_{m-1})/3, \delta/(16m^3))$ is non-increasing in $m$. Hence $U_m$ is also non-increasing in $m$. We also let $\alpha_0 := \alpha_1 = 3K$, and let $\alpha_m := \alpha_{\hat{m}}$ for any epoch $m \geq \hat{m}$. Similarly, we let $\eta_0 := \eta_1 = 1$, and let $\eta_m := \eta_{\hat{m}}$ for any epoch $m \geq \hat{m}$. Sometimes, we use use $C_m(x, \beta, \eta)$ in lieu of $C_m(x, \beta/\eta)$.

# B Bounding Cumulative Regret

This section derives the cumulative regret bounds for $\omega$-RAPR. We start with analyzing the output of oracles described in Assumptions 1 and 2. Note that we do not make the "realizability" assumption in this work – i.e., we do not assume that $f^*$ lies in $\mathcal{F}$. Hence, as in Assumption 1, the expected squared error of our estimated models need not go to zero (even with infinite data) and may contain an unknown non-zero irreducible error term ($B$) that captures the bias of the model class $\mathcal{F}$. It is useful to split our analysis into two regimes to handle this unknown term $B$, similar to the approach in [22]. In particular, we separately analyze oracle outputs for epochs before and after a so-called "safe epoch". Where we define the safe epoch $m^*$ as the epoch where the variance of estimating from the model class $\mathcal{F}$ ($\xi_m$) is dominated by the bias of estimating from the class $\mathcal{F}$ ($B$). That is, $m^* := \arg\max\{m \geq 1 | \xi_{m+1} \geq 2B\}$.

## B.1 High Probability Events

We start with defining high-probability events under which our key theoretical guarantees hold. The first high probability event characterizes the accuracy of estimated reward models and the policy evaluation estimators, the tail bound of which can be obtained by taking a union bound of each epoch-specific event that happens with probability $1 - \frac{\delta}{4m^2}$ under assumptions in Section 1.2.

**Lemma 1.** *Suppose Assumption 1 and Assumption 2 hold. The following event holds with probability* $1 - \delta/2$,

$$
\begin{aligned}
\mathcal{W}_1 := \Bigg\{ & \forall m, \forall \pi \in \Pi \cup \{p_{m+1}\}, \forall f \in \{\hat{f}_1, \ldots, \hat{f}_{m+1}\}, \\
& \mathop{\mathbb{E}}_{x \sim D_{\mathcal{X}}} \mathop{\mathbb{E}}_{a \sim p_m(\cdot|x)} [(\hat{f}_{m+1}(x,a) - f^*(x,a))^2] \leq B + \xi_{m+1}/2 \\
& |\hat{R}_{m+1,f}(\pi) - R_f(\pi)| \leq \sqrt{\xi_{m+1}}, \\
& |\hat{R}_{m+1}(\pi) - R(\pi)| \leq \sqrt{V(p_m, \pi)\xi_{m+1}} + \xi_{m+1} / \Big( \min_{(x,a)\in\mathcal{X}\times\mathcal{A}} p_m(a|x) \Big) \Bigg\}.
\end{aligned}
$$
(9)

*Where $\xi_{m+1} = 2\xi((\tau_m - \tau_{m-1})/3, \delta/(16m^3))$.*[16]

The second high probability event characterizes the measure of conformal arm sets, which directly follows from Hoeffding's inequality and union bound.

**Lemma 2.** *The following event holds with probability* $1 - \delta/2$,

$$
\begin{aligned}
\mathcal{W}_2 := \Bigg\{ & \forall m, \forall \eta \in \Big\{ \frac{|S_{m,2}|}{n} \Big| n \in [|S_{m,2}|] \Big\}, \\
& \Bigg| \mathop{\mathbb{E}}_{x \sim D_{\mathcal{X}}} \Big[ \mu\Big( \bar{C}_{m+1}\Big(x, \frac{\beta_{\max}}{\eta}\Big) \Big) \Big] - \frac{1}{|S_{m,2}|} \sum_{t \in S_{m,2}} \mu\Big( \bar{C}_{m+1}\Big(x_t, \frac{\beta_{\max}}{\eta}\Big) \Big) \Bigg| \\
& \leq \sqrt{\frac{K^2 \ln(8|S_{m,2}|m^2/\delta)}{2|S_{m,2}|}} \Bigg\}.
\end{aligned}
$$
(10)

Together both $\mathcal{W}_1$ and $\mathcal{W}_2$ hold with probability $1 - \delta$. The rest of our analysis works under these events.

## B.2 Analyzing the Cover

In this sub-section, we upper bound the cover ($V(p_m, \cdot)$) for the action selection kernel used in epoch $m$.[17] To upper bound $V(p_m, q)$, we first lower bound $p_m(\cdot|\cdot)$. Recall that in Appendix A, we define

---

[16]Our epoch schedules will always be increasing in epoch length. Under such conditions, we have $\xi_m$ is non-increasing in $m$.

[17]Recall that $V(p,q) := \mathbb{E}_{x \sim D_{\mathcal{X}}, a \sim q(\cdot|x)}[q(a|x)/p(a|x)]$.

$U_m = 20\sqrt{\alpha_{m-1}\xi_m}$ for any epoch $m$. Starting from here, our lemmas and proofs will use $U_m$ and $20\sqrt{\alpha_{m-1}\xi_m}$ interchangeably.

**Lemma 3.** *For any epoch $m$, we have (11) holds.*

$$p_m(a|x) \geq \begin{cases} \frac{1-\beta_{\max}}{\mu(C_m(x,\beta_{\max},\eta_m))} + \frac{\beta_{\max}}{\mu(\mathcal{A})}, & \text{if } a \in C_m(x,\beta_{\max},\eta_m) \\ \frac{\eta_m}{\mu(\mathcal{A})} \min_{\bar{m}\in[m]} \frac{2\bar{m}^2 U_{\bar{m}}}{\hat{f}_{\bar{m}}(x,\pi_{\hat{f}_{\bar{m}}}(x))-\hat{f}_{\bar{m}}(x,a)}, & \text{if } a \notin C_m(x,\beta_{\max},\eta_m) \end{cases}$$
$$\geq \begin{cases} \frac{1}{\mu(\mathcal{A})}, & \text{if } a \in C_m(x,\beta_{\max},\eta_m) \\ \frac{\eta_m \min_{\bar{m}\in[m]} 2\bar{m}^2 U_{\bar{m}}}{\mu(\mathcal{A})}, & \text{if } a \notin C_m(x,\beta_{\max},\eta_m) \end{cases} \tag{11}$$

*Proof.* Recall that $p_m$ given by (12).

$$p_m(a|x) = \frac{(1-\beta_{\max})I[a \in C_m(x,\beta_{\max},\eta_m)]}{\mu(C_m(x,\beta_{\max},\eta_m))} + \int_0^{\beta_{\max}} \frac{I[a \in C_m(x,\beta,\eta_m)]}{\mu(C_m(x,\beta,\eta_m))}\mathrm{d}\beta. \tag{12}$$

We divide our analysis into two cases based on whether $a$ lies in $C_m(x,\beta_{\max},\eta_m)$, and lower bound $p_m(a|x)$ in each case.

**Case 1** ($a \in C_m(x,\beta_{\max},\eta_m)$)**.** Note that $C_m(x,\beta_{\max},\eta_m) \subseteq C_m(x,\beta,\eta_m) \subseteq \mathcal{A}$ for all $\beta \in [0,\beta_{\max}]$. Hence, $a \in C_m(x,\beta,\eta_m)$ and $\mu(C_m(x,\beta,\eta_m)) \leq \mu(\mathcal{A})$ for all $\beta \in [0,\beta_{\max}]$. Therefore, in this case, $p_m(a|x) \geq \frac{(1-\beta_{\max})}{\mu(C_m(x,\beta_{\max},\eta_m))} + \frac{\beta_{\max}}{\mu(\mathcal{A})} \geq \frac{1}{\mu(\mathcal{A})}$.

**Case 2** ($a \notin C_m(x,\beta_{\max},\eta_m)$)**.** For this case, the proof follows from (13).

$$\begin{aligned} p_m(a|x) &\geq \int_0^{\beta_{\max}} \frac{I[a \in C_m(x,\beta,\eta_m)]}{\mu(C_m(x,\beta,\eta_m))}\mathrm{d}\beta \\ &\overset{(i)}{\geq} \frac{1}{\mu(\mathcal{A})} \int_0^{\beta_{\max}} I[a \in C_m(x,\beta,\eta_m)]\mathrm{d}\beta \\ &\overset{(ii)}{\geq} \frac{I(a \notin C_m(x,\beta_{\max},\eta_m))}{\mu(\mathcal{A})} \int_0^{\beta_{\max}} I[a \in C_m(x,\beta,\eta_m)]\mathrm{d}\beta \\ &\overset{(iii)}{=} \frac{I(a \notin C_m(x,\beta_{\max},\eta_m))}{\mu(\mathcal{A})} \int_0^1 I[a \in C_m(x,\beta,\eta_m)]\mathrm{d}\beta \\ &\overset{(iv)}{=} \frac{I(a \notin C_m(x,\beta_{\max},\eta_m))}{\mu(\mathcal{A})} \int_0^1 \prod_{\bar{m}\in[m]} I\left[\hat{f}_{\bar{m}}(x,\pi_{\hat{f}_{\bar{m}}}(x)) - \hat{f}_{\bar{m}}(x,a) \leq \frac{2\bar{m}^2\eta_m U_{\bar{m}}}{\beta}\right]\mathrm{d}\beta \\ &= \frac{I(a \notin C_m(x,\beta_{\max},\eta_m))}{\mu(\mathcal{A})} \int_0^1 \prod_{\bar{m}\in[m]} I\left[\beta \leq \frac{2\bar{m}^2\eta_m U_{\bar{m}}}{\hat{f}_{\bar{m}}(x,\pi_{\hat{f}_{\bar{m}}}(x)) - \hat{f}_{\bar{m}}(x,a)}\right]\mathrm{d}\beta \\ &= \frac{\eta_m I(a \notin C_m(x,\beta_{\max},\eta_m))}{\mu(\mathcal{A})} \min_{\bar{m}\in[m]} \frac{2\bar{m}^2 U_{\bar{m}}}{\hat{f}_{\bar{m}}(x,\pi_{\hat{f}_{\bar{m}}}(x)) - \hat{f}_{\bar{m}}(x,a)} \\ &\overset{(v)}{\geq} \frac{\eta_m I(a \notin C_m(x,\beta_{\max},\eta_m))}{\mu(\mathcal{A})} \min_{\bar{m}\in[m]} 2\bar{m}^2 U_{\bar{m}} \end{aligned} \tag{13}$$

where (i) is because the measure of the conformal set $C_m$ can be no larger than the measure of the action space $\mathcal{A}$; (ii) follows from $I(a \notin C_m(x,\beta_{\max},\eta_m)) \leq 1$; (iii) follows from the fact that if $a \notin C_m(x,\beta_{\max},\eta_m)$ then $a \notin C_m(x,\beta,\eta_m)$ for all $\beta \geq \beta_{\max}$; (iv) follows from Definition 2; and (v) follows from .[18] $\qquad\square$

Using Lemma 3, we get an upper bound on $V(p_m,q)$ in terms of $\mathbb{E}[\mu(C_m(x,\beta_{\max},\eta_m))]$, $K/\eta_m$, and expected regret with respect to the models $\hat{f}_1,\ldots,\hat{f}_m$.

---

[18]Note that if $a \in \pi_{\hat{f}_m}(x)$ then $I(a \notin C_m(x,\beta_{\max},\eta_m)) = 0$.

**Lemma 4.** *For any epoch $m$ and any action selection kernel $q \in \mathcal{P}$, we have (14) holds.*

$$V(p_m, q) \leq \frac{\mathbb{E}[\mu(C_m(x, \beta_{\max}, \eta_m))]}{1 - \beta_{\max}} + \frac{K}{\eta_m} \sum_{\bar{m} \in [m]} \frac{Reg_{\hat{f}_{\bar{m}}}(q)}{2\bar{m}^2 U_{\bar{m}}}. \tag{14}$$

*Proof.* From Lemma 3, we have (15) holds. [19]

$$\frac{I(a \in C_m(x, \beta_{\max}, \eta_m))}{p_m(a|x)} \leq \frac{\mu\big(C_m(x, \beta_{\max}, \eta_m)\big)}{1 - \beta_{\max}},$$

$$\text{and} \quad \frac{I(a \notin C_m(x, \beta_{\max}, \eta_m))}{p_m(a|x)} \leq \frac{\mu(\mathcal{A})}{\eta_m} \max_{\bar{m} \in [m]} \frac{\hat{f}_{\bar{m}}(x, \pi_{\hat{f}_{\bar{m}}}(x)) - \hat{f}_{\bar{m}}(x, a)}{2\bar{m}^2 U_{\bar{m}}}. \tag{15}$$

We now bound the cover $V(p_m, q)$ as follows,

$$V(p_m, q) = \mathop{\mathbb{E}}_{x \sim D_{\mathcal{X}}, a \sim q(\cdot|x)} \left[ \frac{q(a|x)}{p_m(a|x)} \right]$$

$$\overset{(i)}{\leq} \mathop{\mathbb{E}}_{x \sim D_{\mathcal{X}}, a \sim q(\cdot|x)} \left[ \frac{1}{p_m(a|x)} \right]$$

$$= \mathop{\mathbb{E}}_{x \sim D_{\mathcal{X}}, a \sim q(\cdot|x)} \left[ \frac{I[a \in C_m(x, \beta_{\max}, \eta_m)] + I[a \notin C_m(x, \beta_{\max}, \eta_m)]}{p_m(a|x)} \right]$$

$$\overset{(ii)}{\leq} \mathop{\mathbb{E}}_{x \sim D_{\mathcal{X}}, a \sim q(\cdot|x)} \left[ \frac{\mu\big(C_m(x, \beta_{\max}, \eta_m)\big)}{1 - \beta_{\max}} \right] + \mathop{\mathbb{E}}_{x \sim D_{\mathcal{X}}, a \sim q(\cdot|x)} \left[ \frac{\mu(\mathcal{A})}{\eta_m} \max_{\bar{m} \in [m]} \frac{\hat{f}_{\bar{m}}(x, \pi_{\hat{f}_{\bar{m}}}(x)) - \hat{f}_{\bar{m}}(x, a)}{2\bar{m}^2 U_{\bar{m}}} \right]$$

$$\leq \frac{\mathbb{E}[\mu\big(C_m(x, \beta_{\max}, \eta_m)\big)]}{1 - \beta_{\max}} + \frac{\mu(\mathcal{A})}{\eta_m} \sum_{\bar{m} \in [m]} \frac{Reg_{\hat{f}_{\bar{m}}}(q)}{2\bar{m}^2 U_{\bar{m}}}. \tag{16}$$

Here (i) follows from the fact that $q \in \mathcal{P}$ and (ii) follows from (15). □

Having bounded the cover for the kernel $p_m$ in terms of $\mathbb{E}[\mu(C_m(x, \beta_{\max}, \eta_m))]$ and $K/\eta_m$. We now bound these terms with $\alpha_m$.

**Lemma 5.** *Suppose $\mathcal{W}_2$ holds. Then for any epoch $m$, we have (17) holds.*

$$\frac{\mathbb{E}[\mu(C_m(x, \beta_{\max}, \eta_m))]}{1 - \beta_{\max}} + \frac{K}{\eta_m} \leq \alpha_m \leq \frac{3K}{\eta_m}. \tag{17}$$

*Proof.* Since $\mu(C_m(x, \beta_{\max}, \eta_m)) \leq K$ and $\beta_{\max} = 1/2$, the bound trivially holds if $\eta_m = 1$. Suppose $\eta_m > 1$. Note that $\eta_{\bar{m}}$ is non-decreasing in $\bar{m}$ by construction. Let $m'$ be the smallest epoch index such that $\eta_{m'} = \eta_m$. We now have the following holds.

$$\frac{\mathbb{E}[\mu(C_m(x, \beta_{\max}, \eta_m))]}{1 - \beta_{\max}} + \frac{K}{\eta_m}$$

$$\overset{(i)}{\leq} \frac{\min\{1 + \mathbb{E}[\mu(\bar{C}_m(x, \beta_{\max}, \eta_m))], K\}}{1 - \beta_{\max}} + \frac{K}{\eta_m}$$

$$\overset{(ii)}{\leq} \frac{\min\{1 + \mathbb{E}[\mu(\bar{C}_{m'}(x, \beta_{\max}, \eta_m))], K\}}{1 - \beta_{\max}} + \frac{K}{\eta_m}$$

$$\overset{(iii)}{\leq} \frac{\lambda_{m'}(\eta_m)}{1 - \beta_{\max}} + \frac{K}{\eta_m}$$

$$\overset{(iv)}{\leq} 2\lambda_{m'}(\eta_m) + \frac{K}{\eta_m}$$

$$\overset{(v)}{\leq} \frac{3K}{\eta_m} \tag{18}$$

---

[19] Note that we require $f(x, \pi_f(x)) \geq f(x, a)$, for all $x \in \mathcal{X}$, $a \in \mathcal{A}$, and $f \in \mathcal{F}$.

Here (i) follows from the definition of CASs. (ii) follows from $\bar{C}_m \subseteq \bar{C}_{m'}$ which follows from the fact that $\bar{C}_m = \cap_{\bar{m} \in [m]} \tilde{C}_{\bar{m}}$ and $m' \leq m$. (iii) follows from $\mathcal{W}_2$ and the definition of $\lambda_{m'}$ in (5). (iv) follows from the fact that $\beta_{\max} = 1/2$. (v) follows from (5) and $\eta_{m'-1} < \eta_{m'} = \eta_m$ − note that $\eta_{m'-1} \neq \eta_{m'}$ gives us that $\eta_{m'}$ was set using the constrained maximization procedure in (5), hence the constraint $\lambda_{m'}(\eta) \leq K/\eta$ is satisfied at $\eta = \eta_{m'} = \eta_m$. $\qquad\square$

### B.3 Evaluation Guarantees Under Safe Epoch

This sub-section provides guarantees on how accurate $R_{\hat{f}_{m+1}}$ is at evaluating policies when we are within the safe epoch.

**Lemma 6.** *Suppose $\mathcal{W}_1$ holds. For all epochs $m \in [m^*]$, for any $q \in \mathcal{P}$, we have,*

$$|R_{\hat{f}_{m+1}}(q) - R(q)| \leq \sqrt{V(p_m, q)\xi_{m+1}}. \tag{19}$$

*Proof.* Consider any epoch $m \in [m^*]$ and policy $q \in \mathcal{P}$. We then have,

$$
\begin{aligned}
|R_{\hat{f}_{m+1}}(q) - R(q)| &= \Big| \underset{x \sim D_{\mathcal{X}}, a \sim q}{\mathbb{E}} [\hat{f}_{m+1}(x,a) - f^*(x,a)] \Big| \\
&\overset{(i)}{=} \Big| \underset{x \sim D_{\mathcal{X}}, a \sim p_m}{\mathbb{E}} \Big[ \frac{q(a|x)}{p_m(a|x)} \big( \hat{f}_{m+1}(x,a) - f^*(x,a) \big) \Big] \Big| \\
&\leq \underset{x \sim D_{\mathcal{X}}, a \sim p_m}{\mathbb{E}} \Big[ \frac{q(a|x)}{p_m(a|x)} |\hat{f}_{m+1}(x,a) - f^*(x,a)| \Big] \\
&= \underset{x \sim D_{\mathcal{X}}, a \sim p_m}{\mathbb{E}} \Big[ \sqrt{\Big( \frac{q(a|x)}{p_m(a|x)} \Big)^2 |\hat{f}_{m+1}(x,a) - f^*(x,a)|^2} \Big] \\
&\overset{(ii)}{\leq} \sqrt{ \underset{x \sim D_{\mathcal{X}}, a \sim p_m}{\mathbb{E}} \Big[ \Big( \frac{q(a|x)}{p_m(a|x)} \Big)^2 \Big] } \sqrt{ \underset{x \sim D_{\mathcal{X}}, a \sim p_m}{\mathbb{E}} \Big[ (\hat{f}_{m+1}(x,a) - f^*(x,a))^2 \Big] } \\
&\overset{(iii)}{=} \sqrt{ \underset{x \sim D_{\mathcal{X}}, a \sim q}{\mathbb{E}} \Big[ \frac{q(a|x)}{p_m(a|x)} \Big] } \sqrt{ \underset{x \sim D_{\mathcal{X}}, a \sim p_m}{\mathbb{E}} \Big[ (\hat{f}_{m+1}(x,a) - f^*(x,a))^2 \Big] } \\
&\overset{(iv)}{\leq} \sqrt{V(p_m, q)\xi_{m+1}},
\end{aligned}
\tag{20}
$$

where (i) and (iii) follow from change of measure arguments, (ii) follows from Cauchy-Schwartz inequality, and (iv) follows from $\mathcal{W}_1$. $\qquad\square$

By combining the guarantees of Lemma 4 and Lemma 6, we get Lemma 7.

**Lemma 7.** *Suppose $\mathcal{W}_1$ and $\mathcal{W}_2$ hold. Then for any action selection kernel $q \in \mathcal{P}$, we have:*

$$|R_{\hat{f}_{m+1}}(q) - R(q)| \leq \sqrt{\alpha_m \xi_{m+1}} + \frac{1}{2} \sqrt{\alpha_m \xi_{m+1}} \sum_{\bar{m} \in [m]} \frac{Reg_{\hat{f}_{\bar{m}}}(q)}{2\bar{m}^2 U_{\bar{m}}}. \tag{21}$$

*Proof.* From Lemma 4, we have (22) holds for any $q \in \mathcal{P}$.

$$
\begin{aligned}
&V(p_m, q) \\
&\leq \frac{\mathbb{E}[\mu(C_m(x, \beta_{\max}, \eta_m))]}{1 - \beta_{\max}} + \frac{K}{\eta_m} \sum_{\bar{m} \in [m]} \frac{Reg_{\hat{f}_{\bar{m}}}(q)}{2\bar{m}^2 U_{\bar{m}}} \\
&\leq \Big( \frac{\mathbb{E}[\mu(C_m(x, \beta_{\max}, \eta_m))]}{1 - \beta_{\max}} + \frac{K}{\eta_m} \Big) + \Big( \frac{\mathbb{E}[\mu(C_m(x, \beta_{\max}, \eta_m))]}{1 - \beta_{\max}} + \frac{K}{\eta_m} \Big) \sum_{\bar{m} \in [m]} \frac{Reg_{\hat{f}_{\bar{m}}}(q)}{2\bar{m}^2 U_{\bar{m}}} \\
&= \alpha_m + \alpha_m \sum_{\bar{m} \in [m]} \frac{Reg_{\hat{f}_{\bar{m}}}(q)}{2\bar{m}^2 U_{\bar{m}}}
\end{aligned}
\tag{22}
$$

Combining (22) with Lemma 6 we have:

$$
\begin{aligned}
&|R_{\hat{f}_{m+1}}(q) - R(q)| \\
&\leq \sqrt{V(p_m, q)\xi_{m+1}} \\
&\overset{(i)}{\leq} \frac{1}{2}\sqrt{\alpha_m \xi_{m+1}} + \frac{1}{2}\sqrt{\frac{\xi_{m+1}}{\alpha_m}}V(p_m, q) \\
&\overset{(i)}{\leq} \sqrt{\alpha_m \xi_{m+1}} + \frac{1}{2}\sqrt{\alpha_m \xi_{m+1}} \sum_{\bar{m} \in [m]} \frac{\text{Reg}_{\hat{f}_{\bar{m}}}(q)}{2\bar{m}^2 U_{\bar{m}}}.
\end{aligned}
\tag{23}
$$

Where (i) follows from AM-GM inequality and (ii) follows from (22). $\qquad \square$

## B.4 Testing Safety

The misspecification test (2) is designed to test if we are within the safe epoch. In principle, it works by comparing the accuracy of $R_{\hat{f}_{m+1}}$ (Lemma 7) and $\hat{R}_{m+1}$ (Lemma 8). Formally, Lemma 11 shows that the misspecification test in (2) fails only after $m^*$. Hence, $\hat{m} \geq m^* + 1$. Lemma 12 then describes the implication of (2) continuing to hold. In what follows, we let $\widehat{\text{Reg}}_{m+1, \hat{f}_{\bar{m}}}(\pi) := \hat{R}_{m+1, \hat{f}_{\bar{m}}}(\pi_{\hat{f}_{\bar{m}}}) - \hat{R}_{m+1, \hat{f}_{\bar{m}}}(\pi)$. We start with Lemma 8 which provides accuracy guarantees for $\hat{R}_{m+1}$ in any epoch.

**Lemma 8.** *Suppose $\mathcal{W}_1$ and $\mathcal{W}_2$ hold. Then for any epoch $m$ and all $\pi \in \Pi \cup \{p_{m+1}\}$, we have,*

$$
|\hat{R}_{m+1}(\pi) - R(\pi)| \leq \sqrt{\alpha_m \xi_{m+1}} + \frac{1}{2}\sqrt{\alpha_m \xi_{m+1}} \sum_{\bar{m} \in [m]} \frac{\text{Reg}_{\hat{f}_{\bar{m}}}(\pi)}{2\bar{m}^2 U_{\bar{m}}} + \frac{K\xi_{m+1}}{\eta_m \min_{\bar{m} \in [m]} U_{\bar{m}}}.
\tag{24}
$$

*Proof.* From Lemma 3, we have (25) holds, which provides a worst-case lower bound on $p_m$.

$$
\min_{(x,a) \in \mathcal{X} \times \mathcal{A}} p_m(a|x) \geq \min\left(\frac{1}{K}, \frac{\eta_m \min_{\bar{m} \in [m]}(2\bar{m}^2)U_{\bar{m}}}{K}\right) \geq \frac{\eta_m \min_{\bar{m} \in [m]} U_{\bar{m}}}{K}
\tag{25}
$$

Where the last inequality follows from $U_m \leq 1$. Now from $\mathcal{W}_1$, we have,

$$
\begin{aligned}
|\hat{R}_{m+1}(\pi) - R(\pi)| &\overset{(\mathcal{W}_1)}{\leq} \sqrt{V(p_m, \pi)\xi_{m+1}} + \frac{K\xi_{m+1}}{\eta_m \min_{\bar{m} \in [m]} U_{\bar{m}}} \\
&\overset{(i)}{\leq} \frac{1}{2}\sqrt{\alpha_m \xi_{m+1}} + \frac{1}{2}\sqrt{\frac{\xi_{m+1}}{\alpha_m}}V(p_m, \pi) + \frac{K\xi_{m+1}}{\eta_m \min_{\bar{m} \in [m]} U_{\bar{m}}} \\
&\overset{(ii)}{\leq} \sqrt{\alpha_m \xi_{m+1}} + \frac{1}{2}\sqrt{\alpha_m \xi_{m+1}} \sum_{\bar{m} \in [m]} \frac{\text{Reg}_{\hat{f}_{\bar{m}}}(\pi)}{2\bar{m}^2 U_{\bar{m}}} + \frac{K\xi_{m+1}}{\eta_m \min_{\bar{m} \in [m]} U_{\bar{m}}}.
\end{aligned}
\tag{26}
$$

Where (i) follows from AM-GM inequality, and (ii) follows from (22) in the proof of Lemma 7. $\qquad \square$

Lemmas 9 and 10 provide useful inequalities that help construct the misspecification test (2).

**Lemma 9.** *For any epoch $m$, policy $\pi \in \mathcal{P}$, and model $f \in \{\hat{f}_1, \hat{f}_2, \ldots, \hat{f}_{m+1}\}$. We have,*

$$
\begin{aligned}
&||R_f(\pi) - R(\pi)| - |\hat{R}_{m+1, f}(\pi) - \hat{R}_{m+1}(\pi)|| \\
&\leq |\hat{R}_{m+1}(\pi) - R(\pi)| + |R_f(\pi) - \hat{R}_{m+1, f}(\pi)|.
\end{aligned}
$$

*Proof.* The proof follows from noting that,

$$
\begin{aligned}
&|R_f(\pi) - R(\pi)| \\
&= |\hat{R}_{m+1}(\pi) - R(\pi) + R_f(\pi) - \hat{R}_{m+1, f}(\pi) + \hat{R}_{m+1, f}(\pi) - \hat{R}_{m+1}(\pi)| \\
&\leq |\hat{R}_{m+1}(\pi) - R(\pi)| + |R_f(\pi) - \hat{R}_{m+1, f}(\pi)| + |\hat{R}_{m+1, f}(\pi) - \hat{R}_{m+1}(\pi)|.
\end{aligned}
\tag{27}
$$

and from noting that,

$$|\hat{R}_{m+1}(\pi) - \hat{R}_{m+1,f}(\pi)|$$
$$= |\hat{R}_{m+1}(\pi) - R(\pi) + R_f(\pi) - \hat{R}_{m+1,f}(\pi) + R(\pi) - R_f(\pi)| \tag{28}$$
$$\leq |\hat{R}_{m+1}(\pi) - R(\pi)| + |R_f(\pi) - \hat{R}_{m+1,f}(\pi)| + |R_f(\pi) - R(\pi)|.$$
$$\square$$

**Lemma 10.** *Suppose $\mathcal{W}_1$ and $\mathcal{W}_2$ hold. Then for any epoch $m$, any model $f \in \{\hat{f}_i | i \in [m+1]\}$, and any policy $\pi \in \Pi \cup \{p_{m+1}\}$, we have,*

$$|Reg_f(\pi) - \widehat{Reg}_{m+1,f}(\pi)| \leq 2\sqrt{\xi_{m+1}}$$

*Proof.* Follows from triangle inequality and $\mathcal{W}_1$,

$$|\text{Reg}_f(\pi) - \widehat{\text{Reg}}_{m+1,f}(\pi)|$$
$$\leq |R_f(\pi_f) - \hat{R}_{m+1,f}(\pi_f)| + |R_f(\pi) - \hat{R}_{m+1,f}(\pi)| \leq 2\sqrt{\xi_{m+1}} \tag{29}$$
$$\square$$

As discussed earlier, Lemma 11 shows that the misspecification test in (2) fails only after $m^*$. Hence, $\hat{m} \geq m^* + 1$.

**Lemma 11.** *Suppose $\mathcal{W}_1$ and $\mathcal{W}_2$ hold. Now for any epoch $m \in [m^*]$ we have that,*

$$\max_{\pi \in \Pi \cup \{p_{m+1}\}} |\hat{R}_{m+1,\hat{f}_{m+1}}(\pi) - \hat{R}_{m+1}(\pi)| - \sqrt{\alpha_m \xi_{m+1}} \sum_{\bar{m} \in [m]} \frac{\widehat{Reg}_{m+1,\hat{f}_{\bar{m}}}(\pi)}{2\bar{m}^2 U_{\bar{m}}} \tag{30}$$
$$\leq 2.05\sqrt{\alpha_m \xi_{m+1}} + 1.1\sqrt{\xi_{m+1}}.$$

*Proof.* For any epoch $m \in [m^*]$ and for any $\pi \in \Pi \cup \{p_{m+1}\}$, we have,

$$|\hat{R}_{\hat{f}_{m+1}}(\pi) - \hat{R}_{m+1}(\pi)|$$
$$\overset{(i)}{\leq} |\hat{R}_{m+1}(\pi) - R(\pi)| + |R_{\hat{f}_{m+1}}(\pi) - R(\pi)| + |R_{\hat{f}_{m+1}}(\pi) - \hat{R}_{\hat{f}_{m+1}}(\pi)|$$
$$\overset{(ii)}{\leq} 2\sqrt{\alpha_m \xi_{m+1}} + \sqrt{\alpha_m \xi_{m+1}} \sum_{\bar{m} \in [m]} \frac{\text{Reg}_{\hat{f}_{\bar{m}}}(\pi)}{2\bar{m}^2 U_{\bar{m}}} + \frac{K \xi_{m+1}}{\eta_m \min_{\bar{m} \in [m]} U_{\bar{m}}} + \sqrt{\xi_{m+1}}$$
$$\overset{(iii)}{\leq} 2\sqrt{\alpha_m \xi_{m+1}} + \sqrt{\alpha_m \xi_{m+1}} \sum_{\bar{m} \in [m]} \frac{\text{Reg}_{\hat{f}_{\bar{m}}}(\pi)}{2\bar{m}^2 U_{\bar{m}}} + \frac{\alpha_m \xi_{m+1}}{U_m} + \sqrt{\xi_{m+1}}$$
$$\overset{(iv)}{\leq} 2\sqrt{\alpha_m \xi_{m+1}} + \sqrt{\alpha_m \xi_{m+1}} \sum_{\bar{m} \in [m]} \frac{\widehat{Reg}_{m+1,\hat{f}_{\bar{m}}}(\pi)}{2\bar{m}^2 U_{\bar{m}}} + \frac{2\sqrt{\alpha_m}\xi_{m+1}}{U_m} + \frac{\alpha_m \xi_{m+1}}{U_m} + \sqrt{\xi_{m+1}}$$
$$\overset{(v)}{\leq} 2.05\sqrt{\alpha_m \xi_{m+1}} + \sqrt{\alpha_m \xi_{m+1}} \sum_{\bar{m} \in [m]} \frac{\widehat{Reg}_{m+1,\hat{f}_{\bar{m}}}(\pi)}{2\bar{m}^2 U_{\bar{m}}} + 1.1\sqrt{\xi_{m+1}}$$
$$\tag{31}$$

Where (i) follows from Lemma 9. (ii) follows from Lemma 7, Lemma 8, and $\mathcal{W}_1$. (iii) follows from Equation (17) and the fact that $U_{\bar{m}}$ is non-increasing in $\bar{m}$ (giving us $\min_{\bar{m} \in [m]} U_{\bar{m}} = U_m$). (iv) follows from Lemma 10, the fact that $U_{\bar{m}}$ is non-increasing in $\bar{m}$, and the fact that $\sum_{\bar{m}=1}^{\infty} 1/(2\bar{m}^2) \leq 1$. (v) follows from $U_m = 20\sqrt{\alpha_{m-1}\xi_m}$, $\alpha_m \leq \alpha_{m-1}$, and $\xi_{m+1} \leq \xi_m$. $\square$

Lemma 12 now describes the implication of (2) continuing to hold.

**Lemma 12.** *Suppose $\mathcal{W}_1$ and $\mathcal{W}_2$ hold. Now for any epoch $m \in [\hat{m} - 1]$ and any policy $\pi \in \Pi \cup \{p_{m+1}\}$, we then have that,*

$$|R_{\hat{f}_{m+1}}(\pi) - R(\pi)| \leq 2.2\sqrt{\xi_{m+1}} + 3.1\sqrt{\alpha_m \xi_{m+1}} + \frac{3}{2}\sqrt{\alpha_m \xi_{m+1}} \sum_{\bar{m} \in [m]} \frac{Reg_{\hat{f}_{\bar{m}}}(\pi)}{2\bar{m}^2 U_{\bar{m}}}. \tag{32}$$

*Proof.*

$$
\begin{aligned}
&|R_{\hat{f}_{m+1}}(\pi) - R(\pi)| \\
&\overset{(i)}{\leq} |\hat{R}_{m+1}(\pi) - R(\pi)| + |\hat{R}_{\hat{f}_{m+1}}(\pi) - \hat{R}_{m+1}(\pi)| + |R_{\hat{f}_{m+1}}(\pi) - \hat{R}_{\hat{f}_{m+1}}(\pi)| \\
&\overset{(ii)}{\leq} 3.1\sqrt{\alpha_m \xi_{m+1}} + \frac{3}{2}\sqrt{\alpha_m \xi_{m+1}} \sum_{\bar{m} \in [m]} \frac{\operatorname{Reg}_{\hat{f}_{\bar{m}}}(\pi)}{2\bar{m}^2 U_{\bar{m}}} + 2.2\sqrt{\xi_{m+1}}
\end{aligned}
\tag{33}
$$

Where (i) follows from Lemma 9. And (ii) follows from Equation (2), Lemma 8, Lemma 10, and $\mathcal{W}_1$. $\qquad\square$

## B.5 Inductive Argument

This sub-section leverages the guarantee of Lemma 12 and applies it inductively to derive Lemma 14. This lemma bounds $\operatorname{Reg}_\Pi(\pi)$ in terms of $\operatorname{Reg}_{\hat{f}_{m+1}}(\pi)$ and vice-versa for any policy $\pi \in \Pi$. The proof of Lemma 14, relies on the following helpful lemma.

**Lemma 13.** *Consider any class of policies $\Pi' \supseteq \Pi$ and consider any fixed constants $l_1, l_2, l_3, C' > 0$. At any epoch $m$, suppose the policy evaluation guarantee of Equation (34) holds.*

$$
\forall \pi \in \Pi', \ |R_{\hat{f}_{m+1}}(\pi) - R(\pi)| \leq l_1 \sqrt{\xi_{m+1}} + l_2 \sqrt{\alpha_m \xi_{m+1}} + \frac{l_3}{C'} \sum_{\bar{m} \in [m]} \frac{z_{\bar{m},m+1} \operatorname{Reg}_{\hat{f}_{\bar{m}}}(\pi)}{2\bar{m}^2}
\tag{34}
$$

*Now consider fixed constants $C_1, C_2 \geq 0$. As an inductive hypothesis, suppose Equation (35) holds.*

$$
\forall \bar{m} \in [m], \forall \pi \in \Pi', \ \operatorname{Reg}_{\hat{f}_{\bar{m}}}(\pi) \leq \frac{4}{3}\operatorname{Reg}_\Pi(\pi) + C_1\sqrt{\xi_{\bar{m}}} + C_2\sqrt{\alpha_{\bar{m}-1}\xi_{\bar{m}}}.
\tag{35}
$$

*We then have that Equation (36) holds.*

$$
\forall \pi \in \Pi', \operatorname{Reg}_\Pi(\pi) \leq \frac{6}{5}\operatorname{Reg}_{\hat{f}_{m+1}}(\pi) + \frac{12}{5}\left(l_1 + \frac{l_3 C_1}{C'}\right)\sqrt{\xi_{m+1}} + \frac{12}{5}\left(l_2 + \frac{l_3 C_2}{C'}\right)\sqrt{\alpha_m \xi_{m+1}}.
\tag{36}
$$

*Now consider $C_3 \geq 0$ and further suppose Equation (37) holds.*

$$
\forall \bar{m} \in [m], \operatorname{Reg}_{\hat{f}_{\bar{m}}}(\pi_{\hat{f}_{m+1}}) \leq C_3\sqrt{\alpha_{\bar{m}-1}\xi_{\bar{m}}}.
\tag{37}
$$

*We then also have that Equation (38) holds,*

$$
\forall \pi \in \Pi', \operatorname{Reg}_{\hat{f}_{m+1}}(\pi) \leq \frac{7}{6}\operatorname{Reg}_\Pi(\pi) + \left(2l_1 + \frac{l_3 C_1}{C'}\right)\sqrt{\xi_{m+1}} + \left(2l_2 + \frac{l_3(C_2 + C_3)}{C'}\right)\sqrt{\alpha_m \xi_{m+1}}.
\tag{38}
$$

*Where $C' \geq 8l_3$, $z_{\bar{m},m+1} := \sqrt{\frac{\alpha_m \xi_{m+1}}{\alpha_{\bar{m}-1}\xi_{\bar{m}}}} \leq 1$, and $\alpha_m \leq \alpha_{\bar{m}-1}$ for all $\bar{m} \in [m]$.*

*Proof.* Consider any policy $\pi \in \Pi'$. Suppose (34) and (35) hold. We first show (39).

$$\text{Reg}_\Pi(\pi) - \text{Reg}_{\hat{f}_{m+1}}(\pi)$$

$$= R(\pi^*) - R(\pi) - R_{\hat{f}_{m+1}}(\pi_{\hat{f}_{m+1}}) + R_{\hat{f}_{m+1}}(\pi)$$

$$\leq R(\pi^*) - R_{\hat{f}_{m+1}}(\pi^*) + (R_{\hat{f}_{m+1}}(\pi) - R(\pi))$$

$$\overset{(i)}{\leq} 2l_1\sqrt{\xi_{m+1}} + 2l_2\sqrt{\alpha_m \xi_{m+1}} + \frac{l_3}{C'}\sum_{\bar{m}\in[m]}\frac{z_{\bar{m},m+1}}{2\bar{m}^2}\left(\text{Reg}_{\hat{f}_{\bar{m}}}(\pi) + \text{Reg}_{\hat{f}_{\bar{m}}}(\pi^*)\right)$$

$$\overset{(ii)}{\leq} 2l_1\sqrt{\xi_{m+1}} + 2l_2\sqrt{\alpha_m \xi_{m+1}}$$
$$+ \frac{l_3}{C'}\sum_{\bar{m}\in[m]}\frac{z_{\bar{m},m+1}}{2\bar{m}^2}\left(\frac{4}{3}\text{Reg}_\Pi(\pi) + 2C_1\sqrt{\xi_{\bar{m}}} + 2C_2\sqrt{\alpha_{\bar{m}-1}\xi_{\bar{m}}}\right)$$

$$= 2l_1\sqrt{\xi_{m+1}} + 2l_2\sqrt{\alpha_m \xi_{m+1}}$$
$$+ \frac{l_3}{C'}\sum_{\bar{m}\in[m]}\frac{1}{2\bar{m}^2}\left(\frac{4z_{\bar{m},m+1}}{3}\text{Reg}_\Pi(\pi) + \frac{2C_1\sqrt{\xi_{m+1}}}{\sqrt{\alpha_{\bar{m}-1}/\alpha_m}} + 2C_2\sqrt{\alpha_m \xi_{m+1}}\right)$$

$$\overset{(iii)}{\leq} \left(2l_1 + \frac{2l_3 C_1}{C'}\right)\sqrt{\xi_{m+1}} + \left(2l_2 + \frac{2l_3 C_2}{C'}\right)\sqrt{\alpha_m \xi_{m+1}} + \frac{4}{3}\frac{l_3}{C'}\text{Reg}_\Pi(\pi), \tag{39}$$

Where (i) follows from (34), (ii) follows from (35) and from $\text{Reg}_\Pi(\pi^*) = 0$, and finally (iii) follows from $z_{\bar{m},m+1} \leq 1$, $\alpha_m \leq \alpha_{\bar{m}-1}$, and $\sum_{\bar{m}\in[m]} 1/(2\bar{m}^2) \leq 1$. Now (39) immediately implies (40).

$$\left(1 - \frac{4l_3}{3C'}\right)\text{Reg}_\Pi(\pi) \leq \text{Reg}_{\hat{f}_{m+1}}(\pi) + \left(2l_1 + \frac{2l_3 C_1}{C'}\right)\sqrt{\xi_{m+1}} + \left(2l_2 + \frac{2l_3 C_2}{C'}\right)\sqrt{\alpha_m \xi_{m+1}}$$

$$\overset{(i)}{\Longrightarrow} \text{Reg}_\Pi(\pi) \leq \frac{6}{5}\text{Reg}_{\hat{f}_{m+1}}(\pi) + \frac{12}{5}\left(l_1 + \frac{l_3 C_1}{C'}\right)\sqrt{\xi_{m+1}} + \frac{12}{5}\left(l_2 + \frac{l_3 C_2}{C'}\right)\sqrt{\alpha_m \xi_{m+1}} \tag{40}$$

Where (i) follows from the fact that $C' \geq 8l_3$. Similar to (39), we will now show (41).

$$\text{Reg}_{\hat{f}_{m+1}}(\pi) - \text{Reg}_\Pi(\pi)$$

$$= R_{\hat{f}_{m+1}}(\pi_{\hat{f}_{m+1}}) - R_{\hat{f}_{m+1}}(\pi) - (R(\pi^*) - R(\pi))$$

$$\leq \left(R_{\hat{f}_{m+1}}(\pi_{\hat{f}_{m+1}}) - R(\pi_{\hat{f}_{m+1}})\right) + \left(R(\pi) - R_{\hat{f}_{m+1}}(\pi)\right)$$

$$\overset{(i)}{\leq} 2l_1\sqrt{\xi_{m+1}} + 2l_2\sqrt{\alpha_m \xi_{m+1}} + \frac{l_3}{C'}\sum_{\bar{m}\in[m]}\frac{z_{\bar{m},m+1}}{2\bar{m}^2}\left(\text{Reg}_{\hat{f}_{\bar{m}}}(\pi_{\hat{f}_{m+1}}) + \text{Reg}_{\hat{f}_{\bar{m}}}(\pi)\right)$$

$$\overset{(ii)}{\leq} 2l_1\sqrt{\xi_{m+1}} + 2l_2\sqrt{\alpha_m \xi_{m+1}}$$
$$+ \frac{l_3}{C'}\sum_{\bar{m}\in[m]}\frac{z_{\bar{m},m+1}}{2\bar{m}^2}\left(\frac{4}{3}\text{Reg}_\Pi(\pi) + C_1\sqrt{\xi_{\bar{m}}} + (C_2 + C_3)\sqrt{\alpha_{\bar{m}-1}\xi_{\bar{m}}}\right)$$

$$= 2l_1\sqrt{\xi_{m+1}} + 2l_2\sqrt{\alpha_m \xi_{m+1}}$$
$$+ \frac{l_3}{C'}\sum_{\bar{m}\in[m]}\frac{1}{2\bar{m}^2}\left(\frac{4z_{\bar{m},m+1}}{3}\text{Reg}_\Pi(\pi) + \frac{C_1\sqrt{\xi_{m+1}}}{\sqrt{\alpha_{\bar{m}-1}/\alpha_m}} + (C_2 + C_3)\sqrt{\alpha_m \xi_{m+1}}\right)$$

$$\overset{(iii)}{\leq} \left(2l_1 + \frac{l_3 C_1}{C'}\right)\sqrt{\xi_{m+1}} + \left(2l_2 + \frac{l_3(C_2 + C_3)}{C'}\right)\sqrt{\alpha_m \xi_{m+1}} + \frac{4l_3}{3C'}\text{Reg}_\Pi(\pi) \tag{41}$$

Where (i) follows from (34), (ii) follows from (35), (37), and (iii) follows from $z_{\bar{m},m+1} \leq 1$, $\alpha_m \leq \alpha_{\bar{m}-1}$, and $\sum_{\bar{m}\in[m]} 1/(2\bar{m}^2) \leq 1$. Now (41) immediately implies (42).

$$\text{Reg}_{\hat{f}_{m+1}}(\pi) \leq \left(1 + \frac{4l_3}{3C'}\right)\text{Reg}_\Pi(\pi) + \left(2l_1 + \frac{l_3 C_1}{C'}\right)\sqrt{\xi_{m+1}} + \left(2l_2 + \frac{l_3(C_2 + C_3)}{C'}\right)\sqrt{\alpha_m \xi_{m+1}}$$

$$\overset{(i)}{\Longrightarrow} \text{Reg}_{\hat{f}_{m+1}}(\pi) \leq \frac{7}{6}\text{Reg}_\Pi(\pi) + \left(2l_1 + \frac{l_3 C_1}{C'}\right)\sqrt{\xi_{m+1}} + \left(2l_2 + \frac{l_3(C_2 + C_4)}{C'}\right)\sqrt{\alpha_m \xi_{m+1}}. \tag{42}$$

Where (i) follows from the fact that $C' \geq 8l_3$. $\qquad\square$

**Lemma 14.** *Suppose $\mathcal{W}_1$ and $\mathcal{W}_2$ hold. Now for any epoch $m \in [\hat{m} - 1]$, we then have that (43) holds.*

$$\forall \pi \in \Pi,\ Reg_\Pi(\pi) \le \frac{4}{3} Reg_{\hat{f}_{m+1}}(\pi) + 6.5\sqrt{\xi_{m+1}} + 12\sqrt{\alpha_m \xi_{m+1}},$$
$$Reg_{\hat{f}_{m+1}}(\pi) \le \frac{4}{3} Reg_\Pi(\pi) + 6.5\sqrt{\xi_{m+1}} + 12\sqrt{\alpha_m \xi_{m+1}}. \tag{43}$$

*Moreover when $m \in [m^*]$, we have (43) holds for all policies $\pi \in \mathcal{P}$.*

*Proof.* Note that (43) trivially holds for $m = 0$. We will now use an inductive argument. Consider any epoch $m \in [\hat{m}]$. As an inductive hypothesis, let us assume (44) holds. (i.e. (43) holds for epoch $m - 1$.)

$$\forall \pi \in \Pi, \bar{m} \in [m],$$
$$\mathrm{Reg}_\Pi(\pi) \le \frac{4}{3}\mathrm{Reg}_{\hat{f}_{\bar{m}}}(\pi) + 6.5\sqrt{\xi_{\bar{m}}} + 12\sqrt{\alpha_{\bar{m}-1}\xi_{\bar{m}}}, \tag{44}$$
$$\mathrm{Reg}_{\hat{f}_{\bar{m}}}(\pi) \le \frac{4}{3}\mathrm{Reg}_\Pi(\pi) + 6.5\sqrt{\xi_{\bar{m}}} + 12\sqrt{\alpha_{\bar{m}-1}\xi_{\bar{m}}}.$$

Hence from (44), we have (35) holds with $C_1 = 6.5$ and $C_2 = 12$. Since $m \in [\hat{m}]$, from Lemma 12, we have (45) holds.

$$\forall \pi \in \Pi \cup \{p_{m+1}\},$$
$$|R_{\hat{f}_{m+1}}(\pi) - R(\pi)| \le \frac{22}{10}\sqrt{\xi_{m+1}} + \frac{31}{10}\sqrt{\alpha_m \xi_{m+1}} + \frac{3}{40}\sum_{\bar{m}\in[m]} \frac{z_{\bar{m},m+1}\mathrm{Reg}_{\hat{f}_{\bar{m}}}(\pi)}{2\bar{m}^2} \tag{45}$$

Hence from (45), we have (34) holds with $l_1 = 2.2$, $l_2 = 3.1$, $l_3 = 1.5$, $C' = 20$, and $\tilde{\Pi} = \Pi$. Hence from Lemma 13, we have (46) holds.

$$\forall \pi \in \Pi,$$
$$\mathrm{Reg}_\Pi(\pi) \le \frac{6}{5}\mathrm{Reg}_{\hat{f}_{m+1}}(\pi) + \frac{12}{5}\left(\frac{22}{10} + \frac{1.5*6.5}{20}\right)\sqrt{\xi_{m+1}} + \frac{12}{5}\left(\frac{31}{10} + \frac{1.5*12}{20}\right)\sqrt{\alpha_m \xi_{m+1}}$$
$$= \frac{6}{5}\mathrm{Reg}_{\hat{f}_{m+1}}(\pi) + 6.45\sqrt{\xi_{m+1}} + 9.6\sqrt{\alpha_m \xi_{m+1}}$$
$$\le \frac{4}{3}\mathrm{Reg}_{\hat{f}_{m+1}}(\pi) + 6.5\sqrt{\xi_{m+1}} + 12\sqrt{\alpha_m \xi_{m+1}} \tag{46}$$

Now from (44) and (46), we have (47) holds.

$$\forall \bar{m} \in [m],$$
$$\mathrm{Reg}_{\hat{f}_{\bar{m}}}(\pi_{\hat{f}_{m+1}}) \le \frac{4}{3}\mathrm{Reg}_\Pi(\pi_{\hat{f}_{m+1}}) + 6.5\sqrt{\xi_{\bar{m}}} + 12\sqrt{\alpha_{\bar{m}-1}\xi_{\bar{m}}}$$
$$\le \frac{4}{3}\left(0 + 6.5\sqrt{\xi_{m+1}} + 12\sqrt{\alpha_m \xi_{m+1}}\right) + 6.5\sqrt{\xi_{\bar{m}}} + 12\sqrt{\alpha_{\bar{m}-1}\xi_{\bar{m}}} \tag{47}$$
$$\le 43.2\sqrt{\alpha_{\bar{m}-1}\xi_{\bar{m}}}$$

Hence from (47), we have (37) holds with $C_3 = 43.2$. Therefore from Lemma 13 we have (48).

$$\forall \pi \in \Pi,$$
$$\mathrm{Reg}_{\hat{f}_{m+1}}(\pi) \le \frac{7}{6}\mathrm{Reg}_\Pi(\pi) + \left(2l_1 + \frac{l_3 C_1}{C'}\right)\sqrt{\xi_{m+1}} + \left(2l_2 + \frac{l_3(C_2 + C_3)}{C'}\right)\sqrt{\alpha_m \xi_{m+1}}$$
$$= \frac{7}{6}\mathrm{Reg}_\Pi(\pi) + \left(2*2.2 + \frac{1.5*6.5}{20}\right)\sqrt{\xi_{m+1}} + \left(2*3.1 + \frac{1.5(12+43.2)}{20}\right)\sqrt{\alpha_m \xi_{m+1}} \tag{48}$$
$$= \frac{7}{6}\mathrm{Reg}_\Pi(\pi) + 4.8875\sqrt{\xi_{m+1}} + 10.34\sqrt{\alpha_m \xi_{m+1}}$$
$$\le \frac{4}{3}\mathrm{Reg}_\Pi(\pi) + 6.5\sqrt{\xi_{m+1}} + 12\sqrt{\alpha_m \xi_{m+1}}$$

From (46) and (48), we have (43) holds for epoch $m$. This completes our inductive argument. $\square$

An immediate implication of Lemma 14 is that we have $\text{Reg}_{\hat{f}_m}(\pi^*) \leq U_m$ for all $m \in [\hat{m}]$. Hence, from Lemma 4 and Lemma 5, we have (49) holds.

$$V(p_m, \pi^*) \leq \frac{\mathbb{E}[\mu(C_m(x, \beta_{\max}, \eta_m))]}{1 - \beta_{\max}} + \frac{K}{\eta_m} \leq \alpha_m, \ \forall m \in [\hat{m}]. \tag{49}$$

### B.6 Bounding Exploration and Cumulative Regret

This sub-section leverages the structure of the kernel $p_{m+1}$, and the guarantees in Lemmas 12 and 14 to bound the expected regret during exploration (Lemma 17). Then, summing up these exploration regret bounds, we get our cumulative regret bound in Theorem 1. We start with Lemma 15 which leverages structure in $p_m$ to bound $\text{Reg}_{\hat{f}_{\bar{m}}}(p_m)$ for any $\bar{m} \in [m]$.

**Lemma 15.** *For any pair of epochs $m \in [\hat{m}+1]$ and $\bar{m} \in [m]$, we have that (50) holds.*

$$Reg_{\hat{f}_{\bar{m}}}(p_m) \leq 15.2\sqrt{\xi_{\bar{m}}} + 28\sqrt{\alpha_{\bar{m}-1}\xi_{\bar{m}}} + 2\bar{m}^2\eta_m U_{\bar{m}}\left(\frac{1}{\beta_{\max}} + \ln\frac{\beta_{\max}}{2\bar{m}^2\eta_m U_{\bar{m}}}\right) \tag{50}$$

*Proof.* We first make the following observation.

$$\mathbb{E}_{x \sim D_x} \mathbb{E}_{a \sim p_m(a|x)}[I(a \notin \pi_{\hat{f}_m}(x)) \cdot (\hat{f}_{\bar{m}}(x, \pi_{\hat{f}_{\bar{m}}}(x)) - \hat{f}_{\bar{m}}(x, a))]$$

$$= \mathbb{E}_{x \sim D_x}\left[\int_{a \in \mathcal{A} \setminus \pi_{\hat{f}_m}(x)} (\hat{f}_{\bar{m}}(x, \pi_{\hat{f}_{\bar{m}}}(x)) - \hat{f}_{\bar{m}}(x, a))p_m(a|x)d\mu(a)\right]$$

$$\overset{(i)}{=} \mathbb{E}_{x \sim D_x}\left[\int_{a \in \mathcal{A} \setminus \pi_{\hat{f}_m}(x)} \int_{\beta \in [0,1]} (\hat{f}_{\bar{m}}(x, \pi_{\hat{f}_{\bar{m}}}(x)) - \hat{f}_{\bar{m}}(x, a))\frac{I[a \in C_m(x, \min(\beta, \beta_{\max})/\eta_m)]}{\mu(C_m(x, \min(\beta, \beta_{\max})/\eta_m))}d\beta d\mu(a)\right]$$

$$\overset{(ii)}{\leq} \mathbb{E}_{x \sim D_x}\left[\int_{\beta \in [0,1]} \int_{a \in \mathcal{A} \setminus \pi_{\hat{f}_m}(x)} \min\left(1, \frac{2\bar{m}^2\eta_m U_{\bar{m}}}{\min(\beta, \beta_{\max})}\right)\frac{I[a \in C_m(x, \min(\beta, \beta_{\max})/\eta_m)]}{\mu(C_m(x, \min(\beta, \beta_{\max})/\eta_m))}d\mu(a)d\beta\right]$$

$$\leq \int_{\beta \in [0,1]} \min\left(1, \frac{2\bar{m}^2\eta_m U_{\bar{m}}}{\min(\beta, \beta_{\max})}\right)d\beta \leq (1 - \beta_{\max})\frac{2\bar{m}^2\eta_m U_{\bar{m}}}{\beta_{\max}} + \int_0^{\beta_{\max}} \min\left(1, \frac{2\bar{m}^2\eta_m U_{\bar{m}}}{\beta}\right)d\beta \tag{51}$$

where (i) follows from the definition of $p_m$ given in (4). (ii) follows from the fact that for any $\zeta \in (0,1)$ and $a \in C_m(x, \zeta) \setminus \pi_{\hat{f}_m}(x)$ we have $\hat{f}_{\bar{m}}(x, \pi_{\hat{f}_{\bar{m}}}(x)) - \hat{f}_{\bar{m}}(x, a) \leq \min(1, 2\bar{m}^2/\zeta)$, since $C_m(x, \zeta) \setminus \pi_{\hat{f}_m}(x) \subseteq \bar{C}_m(x, \zeta) \subseteq \tilde{C}_{\bar{m}}(x, \zeta/(2\bar{m}^2))$ by Definition 2, . We now bound $\text{Reg}_{\hat{f}_{\bar{m}}}(p_m)$.

$$\text{Reg}_{\hat{f}_{\bar{m}}}(p_m) = \mathbb{E}_{x \sim D_x} \mathbb{E}_{a \sim p_m(a|x)}[\hat{f}_{\bar{m}}(x, \pi_{\hat{f}_{\bar{m}}}(x)) - \hat{f}_{\bar{m}}(x, a)]$$

$$= \mathbb{E}_{x \sim D_x} \mathbb{E}_{a \sim p_m(a|x)}[(I(a \in \pi_{\hat{f}_m}(x)) + I(a \notin \pi_{\hat{f}_m}(x))) \cdot (\hat{f}_{\bar{m}}(x, \pi_{\hat{f}_{\bar{m}}}(x)) - \hat{f}_{\bar{m}}(x, a))]$$

$$\overset{(i)}{\leq} \text{Reg}_{\hat{f}_{\bar{m}}}(\pi_{\hat{f}_m}) + \left((1 - \beta_{\max})\frac{2\bar{m}^2\eta_m U_{\bar{m}}}{\beta_{\max}} + \int_0^{\beta_{\max}} \min\left(1, \frac{2\bar{m}^2\eta_m U_{\bar{m}}}{\beta}\right)d\beta\right)$$

$$\leq \text{Reg}_{\hat{f}_{\bar{m}}}(\pi_{\hat{f}_m}) + \left(\frac{2\bar{m}^2\eta_m U_{\bar{m}}(1 - \beta_{\max})}{\beta_{\max}} + 2\bar{m}^2\eta_m U_{\bar{m}} + 2\bar{m}^2\eta_m U_{\bar{m}}\int_{2\bar{m}^2\eta_m U_{\bar{m}}}^{\beta_{\max}} \frac{1}{\beta}d\beta\right)$$

$$= \text{Reg}_{\hat{f}_{\bar{m}}}(\pi_{\hat{f}_m}) + 2\bar{m}^2\eta_m U_{\bar{m}}\left(\frac{1}{\beta_{\max}} + \ln\frac{\beta_{\max}}{2\bar{m}^2\eta_m U_{\bar{m}}}\right) \tag{52}$$

$$\overset{(ii)}{\leq} \frac{4}{3}\text{Reg}_{\Pi}(\pi_{\hat{f}_m}) + 6.5\sqrt{\xi_{\bar{m}}} + 12\sqrt{\alpha_{\bar{m}-1}\xi_{\bar{m}}} + 2\bar{m}^2\eta_m U_{\bar{m}}\left(\frac{1}{\beta_{\max}} + \ln\frac{\beta_{\max}}{2\bar{m}^2\eta_m U_{\bar{m}}}\right)$$

$$\overset{(iii)}{\leq} \frac{4}{3}\left(6.5\sqrt{\xi_m} + 12\sqrt{\alpha_{m-1}\xi_m}\right) + 6.5\sqrt{\xi_{\bar{m}}} + 12\sqrt{\alpha_{\bar{m}-1}\xi_{\bar{m}}}$$

$$\qquad + 2\bar{m}^2\eta_m U_{\bar{m}}\left(\frac{1}{\beta_{\max}} + \ln\frac{\beta_{\max}}{2\bar{m}^2\eta_m U_{\bar{m}}}\right)$$

$$\overset{(iv)}{\leq} 15.2\sqrt{\xi_{\bar{m}}} + 28\sqrt{\alpha_{\bar{m}-1}\xi_{\bar{m}}} + 2\bar{m}^2\eta_m U_{\bar{m}}\left(\frac{1}{\beta_{\max}} + \ln\frac{\beta_{\max}}{2\bar{m}^2\eta_m U_{\bar{m}}}\right)$$

where (i) follows from (51), (ii) follows from Lemma 14, (iii) follows from Lemma 14 and $\text{Reg}_{\hat{f}_m}(\pi_{\hat{f}_m}) = 0$, and (iv) follows from $\bar{m} \leq m$. $\square$

Now from the guarantees in Lemmas 12, 14, and 15 we get the following bound on $\text{Reg}_\Pi(p_{m+1})$.

**Lemma 16.** *Suppose $\mathcal{W}_1$ and $\mathcal{W}_2$ hold. Now for any epoch $m \in [\hat{m} - 1]$, we have that (53) holds.*

$$Reg_\Pi(p_{m+1}) \leq 100(m+1)^2 \eta_{m+1} \sqrt{\alpha_m \xi_{m+1}} \left( \frac{1}{\beta_{\max}} + \ln \frac{\beta_{\max}}{40 \eta_{m+1} \sqrt{\alpha_m \xi_{m+1}}} \right) \tag{53}$$

*Proof.* Since $m \in [\hat{m}]$, from Lemma 12, we have (54) holds.

$$\forall \pi \in \Pi \cup \{p_{m+1}\},$$

$$|R_{\hat{f}_{m+1}}(\pi) - R(\pi)| \leq \frac{22}{10}\sqrt{\xi_{m+1}} + \frac{31}{10}\sqrt{\alpha_m \xi_{m+1}} + \frac{3}{40}\sum_{\bar{m} \in [m]} \frac{z_{\bar{m},m+1}\text{Reg}_{\hat{f}_{\bar{m}}}(\pi)}{2\bar{m}^2} \tag{54}$$

We will now bound $\text{Reg}_\Pi(p_{m+1})$ in terms of $\text{Reg}_{\hat{f}_{\bar{m}}}(p_{m+1})$ for $\bar{m} \in [m+1]$.

$$\text{Reg}_\Pi(p_{m+1}) - \text{Reg}_{\hat{f}_{m+1}}(p_{m+1})$$
$$= R(\pi^*) - R(p_{m+1}) - R_{\hat{f}_{m+1}}(\pi_{\hat{f}_{m+1}}) + R_{\hat{f}_{m+1}}(p_{m+1})$$
$$\leq R(\pi^*) - R_{\hat{f}_{m+1}}(\pi^*) + (R_{\hat{f}_{m+1}}(p_{m+1}) - R(p_{m+1}))$$
$$\overset{(i)}{\leq} \frac{44}{10}\sqrt{\xi_{m+1}} + \frac{62}{10}\sqrt{\alpha_m \xi_{m+1}} + \frac{3}{40}\sum_{\bar{m} \in [m]} \frac{z_{\bar{m},m+1}}{2\bar{m}^2}(\text{Reg}_{\hat{f}_{\bar{m}}}(p_{m+1}) + \text{Reg}_{\hat{f}_{\bar{m}}}(\pi^*))$$

$$\overset{(ii)}{\leq} \frac{44}{10}\sqrt{\xi_{m+1}} + \frac{62}{10}\sqrt{\alpha_m \xi_{m+1}}$$
$$+ \frac{3}{40}\sum_{\bar{m} \in [m]} \frac{z_{\bar{m},m+1}}{2\bar{m}^2}(\text{Reg}_{\hat{f}_{\bar{m}}}(p_{m+1}) + 6.5\sqrt{\xi_{\bar{m}}} + 12\sqrt{\alpha_{\bar{m}-1}\xi_{\bar{m}}}) \tag{55}$$

$$\overset{(iii)}{\leq} \frac{44}{10}\sqrt{\xi_{m+1}} + \frac{62}{10}\sqrt{\alpha_m \xi_{m+1}}$$
$$+ \frac{3}{40}\sum_{\bar{m} \in [m]} \frac{1}{2\bar{m}^2}(z_{\bar{m},m+1}\text{Reg}_{\hat{f}_{\bar{m}}}(p_{m+1}) + 6.5\sqrt{\xi_{m+1}} + 12\sqrt{\alpha_m \xi_{m+1}})$$

$$\overset{(iv)}{\leq} 4.9\sqrt{\xi_{m+1}} + 7.1\sqrt{\alpha_m \xi_{m+1}} + \frac{3}{40}\sum_{\bar{m} \in [m]} \frac{z_{\bar{m},m+1}}{2\bar{m}^2}\text{Reg}_{\hat{f}_{\bar{m}}}(p_{m+1}).$$

Where (i) follows from (54), (ii) follows from Lemma 14 and from $\text{Reg}_\Pi(\pi^*) = 0$, (iii) follows from $z_{\bar{m},m+1} := \sqrt{\frac{\alpha_m \xi_{m+1}}{\alpha_{\bar{m}-1}\xi_{\bar{m}}}}$ and $\alpha_m \leq \alpha_{\bar{m}-1}$, finally (iv) follows from $\sum_{\bar{m} \in [m]} 1/(2\bar{m}^2) \leq 1$. We now simplify the last term in the upper bound of (55).

$$\sum_{\bar{m} \in [m]} \frac{z_{\bar{m},m+1}}{2\bar{m}^2}\text{Reg}_{\hat{f}_{\bar{m}}}(p_{m+1})$$

$$\overset{(i)}{\leq} \sum_{\bar{m} \in [m]} \frac{z_{\bar{m},m+1}}{2\bar{m}^2}\left(15.2\sqrt{\xi_{\bar{m}}} + 28\sqrt{\alpha_{\bar{m}-1}\xi_{\bar{m}}} + 2\bar{m}^2 \eta_{m+1} U_{\bar{m}}\left(\frac{1}{\beta_{\max}} + \ln \frac{\beta_{\max}}{2\bar{m}^2 \eta_{m+1} U_{\bar{m}}}\right)\right)$$

$$\overset{(ii)}{\leq} \sum_{\bar{m} \in [m]} \frac{1}{2\bar{m}^2}\left(15.2\sqrt{\xi_{m+1}} + 28\sqrt{\alpha_m \xi_{m+1}}\right.$$

$$\left. + 40\bar{m}^2 \eta_{m+1}\sqrt{\alpha_m \xi_{m+1}}\left(\frac{1}{\beta_{\max}} + \ln \frac{\beta_{\max}}{40\eta_{m+1}\sqrt{\alpha_m \xi_{m+1}}}\right)\right)$$

$$\overset{(iii)}{\leq} 15.2\sqrt{\xi_{m+1}} + 28\sqrt{\alpha_m \xi_{m+1}} + 20m\eta_{m+1}\sqrt{\alpha_m \xi_{m+1}}\left(\frac{1}{\beta_{\max}} + \ln \frac{\beta_{\max}}{40\eta_{m+1}\sqrt{\alpha_m \xi_{m+1}}}\right)$$
$$\tag{56}$$

Where (i) follows from Lemma 15, (ii) follows from $z_{\bar{m},m+1} := \sqrt{\frac{\alpha_m \xi_{m+1}}{\alpha_{\bar{m}-1} \xi_{\bar{m}}}}$, choice of $U_m$, and $\alpha_m \leq \alpha_{\bar{m}-1}$, finally (iii) follows from $\sum_{\bar{m} \in [m]} 1/(2\bar{m}^2) \leq 1$. By combining (55), (56), and Lemma 15, we get our final result.

$$
\begin{aligned}
&\mathrm{Reg}_\Pi(p_{m+1}) \\
&\overset{(i)}{\leq} \mathrm{Reg}_{\hat{f}_{m+1}}(p_{m+1}) + 6.04\sqrt{\xi_{m+1}} + 9.2\sqrt{\alpha_m \xi_{m+1}} \\
&\quad + 1.5 m \eta_{m+1} \sqrt{\alpha_m \xi_{m+1}} \left( \frac{1}{\beta_{\max}} + \ln \frac{\beta_{\max}}{40 \eta_{m+1} \sqrt{\alpha_m \xi_{m+1}}} \right) \\
&\overset{(ii)}{\leq} 21.3\sqrt{\xi_{m+1}} + 37.2\sqrt{\alpha_m \xi_{m+1}} \\
&\quad + 41.5(m+1)^2 \eta_{m+1} \sqrt{\alpha_m \xi_{m+1}} \left( \frac{1}{\beta_{\max}} + \ln \frac{\beta_{\max}}{40 \eta_{m+1} \sqrt{\alpha_m \xi_{m+1}}} \right)
\end{aligned}
\tag{57}
$$

Where (i) follows from (55) and (56), and (ii) follows from Lemma 15. $\qquad\square$

The earlier bound on $\mathrm{Reg}_\Pi(p_{m+1})$ now immediately gives us the following bound on $\mathrm{Reg}_{f^*}(p_{m+1})$.

**Lemma 17.** *Suppose $\mathcal{W}_1$ and $\mathcal{W}_2$ hold. Now for any epoch $m \in [\hat{m}-1]$, we have that* (58)

$$
Reg_{f^*}(p_{m+1}) \leq 2\sqrt{KB} + 100(m+1)^2 \eta_{m+1} \sqrt{\alpha_m \xi_{m+1}} \left( \frac{1}{\beta_{\max}} + \ln \frac{\beta_{\max}}{40 \eta_{m+1} \sqrt{\alpha_m \xi_{m+1}}} \right)
\tag{58}
$$

*Proof.* From Assumption 1 (properties of EstOracle), we know the bias of the model class $\mathcal{F}$ is bounded by $B$. In particular, we know there exists $g \in \mathcal{F}$ such that $\mathbb{E}_{x \sim D_\mathcal{X}, a \sim \mathrm{Unif}(\mathcal{A})} \left[ (g(x,a) - f^*(x,a))^2 \right] \leq B$. Hence, we have, the following.

$$
\begin{aligned}
\mathrm{Reg}_{f^*}(\pi^*) &\overset{(i)}{\leq} \mathrm{Reg}_{f^*}(\pi_g) = R(\pi_{f^*}) - R(\pi_g) \\
&= (R(\pi_{f^*}) - R_g(\pi_{f^*})) - \mathrm{Reg}_g(\pi_{f^*}) + (R_g(\pi_g) - R(\pi_g)) \\
&\overset{(ii)}{\leq} |R(\pi_{f^*}) - R_g(\pi_{f^*})| + |R_g(\pi_g) - R(\pi_g)| \\
&\overset{(iii)}{\leq} \left( \sqrt{\mathbb{E}_{x \sim D_\mathcal{X}, a \sim \pi_{f^*}} \left[ \frac{\pi_{f^*}(a|x)}{1/K} \right]} + \sqrt{\mathbb{E}_{x \sim D_\mathcal{X}, a \sim \pi_g} \left[ \frac{\pi_g(a|x)}{1/K} \right]} \right) \\
&\qquad \cdot \sqrt{\mathbb{E}_{x \sim D_\mathcal{X}, a \sim \mathrm{Unif}(\mathcal{A})} \left[ (g(x,a) - f^*(x,a))^2 \right]} \\
&\overset{(iv)}{\leq} 2\sqrt{KB}.
\end{aligned}
\tag{59}
$$

Here (i) follows from the fact that $\pi_g \in \Pi$ since $g \in \mathcal{F}$. (ii) follows from triangle inequality and the fact that $\mathrm{Reg}_g(\pi_{f^*}) \geq 0$. (iii) follows from the proof of Lemma 7. And (iv) follows from $\mathbb{E}_{x \sim D_\mathcal{X}, a \sim \mathrm{Unif}(\mathcal{A})} \left[ (g(x,a) - f^*(x,a))^2 \right] \leq B$ and $\pi_f(a|x) = I(a \in \pi_f(x))$.

Since $\mathrm{Reg}_{f^*}(p_{m+1}) = R(\pi_{f^*}) - R(\pi^*) + R(\pi^*) - R(p_{m+1}) = \mathrm{Reg}_{f^*}(\pi^*) + \mathrm{Reg}_\Pi(p_{m+1})$, the result follows from combining the above with Lemma 16. $\qquad\square$

We now get our final cumulative regret bound by summing up the exploration regret bounds in Lemma 17.

**Theorem 1.** *Suppose Assumptions 1 and 2 hold. Then with probability $1 - \delta$, $\omega$-RAPR attains the following cumulative regret guarantee. Here $\xi_{m+1} = \xi((\tau_m - \tau_{m-1})/3, \delta/(16m^3)$ for all $m$.*

$$
CReg_T \leq \tilde{\mathcal{O}}\left( \sum_{t=\tau_1+1}^{T} \sqrt{\frac{K}{\alpha_{m(t)}} \frac{\alpha_{m(t)-1}}{\alpha_{m(t)}}} \left( \sqrt{KB} + \sqrt{K\xi_{m(t)}} \right) \right)
\tag{6a}
$$

$$
\leq \tilde{\mathcal{O}}\left( \sqrt{\omega KBT} + \sum_{t=\tau_1+1}^{T} \sqrt{\omega K \xi_{m(t)}} \right).
\tag{6b}
$$

*Where we use $\tilde{\mathcal{O}}$ to hide terms logarithmic in $T, K, \xi(T, \delta)$.*

*Proof.* From Appendix B.1, both $\mathcal{W}_1$ and $\mathcal{W}_2$ hold with probability $1 - \delta$. We prove our cumulative regret bounds under these events. Under $\mathcal{W}_1$, from Lemma 11, we have $\hat{m} \geq m^* + 1$. Further, from conditions in Section 1.2, we have $\xi_m$ is non-increasing in $m$. Since $\xi(n, \delta')$ scales polynomially in $1/n$ and $\log(1/\delta')$, there exists a constant $Q_0 > 1$ such that the doubling epoch structure ensures $\xi_m \leq Q_0 \xi_{m+1}$ for all $m$. Hence $\xi_{\hat{m}} \leq Q_0 \xi_{\hat{m}+1} \leq Q_0 \xi_{m^*+2} \leq 2 Q_0 B$. Let $m'(t) = \min(m(t), \hat{m})$. Hence, $\xi_{m'(t)} \leq \max(\xi_{m(t)}, \xi_{\hat{m}}) \leq 2 Q_0 B + \xi_{m(t)}$. Therefore, by summing up the bounds in Lemma 17, we have the following cumulative regret bound.

$$
\begin{aligned}
\mathrm{CReg}_T &\leq \sum_{t=1}^{T} \mathrm{Reg}_{f^*}(p_{m'(t)}) \\
&\leq \tau_1 + \sum_{t=\tau_1+1}^{T} \Bigg( 2\sqrt{KB} \\
&\qquad + 100(m'(t))^2 \eta_{m'(t)} \sqrt{\alpha_{m'(t)-1}\xi_{m'(t)}} \bigg( \frac{1}{\beta_{\max}} + \ln \frac{\beta_{\max}}{40 \eta_{m'(t)}\sqrt{\alpha_{m'(t)-1}\xi_{m'(t)}}} \bigg) \Bigg) \quad (60) \\
&\leq \tilde{\mathcal{O}}\Bigg( \sum_{t=\tau_1+1}^{T} \bigg( \eta_{m'(t)}\sqrt{\alpha_{m'(t)-1}\xi_{m'(t)}} \bigg) \Bigg) = \tilde{\mathcal{O}}\Bigg( \sum_{t=\tau_1+1}^{T} \eta_{m(t)}\sqrt{\frac{\alpha_{m(t)-1}}{K}}\Big(\sqrt{K\xi_{m'(t)}}\Big) \Bigg) \\
&\leq \tilde{\mathcal{O}}\Bigg( \sum_{t=\tau_1+1}^{T} \eta_{m(t)}\sqrt{\frac{\alpha_{m(t)-1}}{K}}\Big(\sqrt{KB} + \sqrt{K\xi_{m(t)}}\Big) \Bigg)
\end{aligned}
$$

Now the theorem follows from the fact that we have:

$$
\eta_m \sqrt{\frac{\alpha_{m-1}}{K}} \overset{Lemma\ 5}{\leq} 3\sqrt{\frac{K}{\alpha_m}\frac{\alpha_{m-1}}{\alpha_m}} \overset{Lemma\ 5}{\leq} 3\eta_m\sqrt{\frac{\alpha_{m-1}}{K}} \overset{(5)}{\leq} 3\sqrt{\omega}. \quad (61)
$$

$\square$

## C  Bounding Simple Regret

In this section, we prove our simple regret bound (Theorem 2). Our analysis starts with Lemma 18, which provides instance dependent bounds on $\mathbb{E}_{x \sim D_{\mathcal{X}}}\left[\mu\Big(C_m(x, \beta, \eta)\Big)\right]$. We will later use Lemma 18 to derive instance-dependent bounds on $\alpha_m$. This bound then helps us derive instance-dependant bounds on simple regret.

**Lemma 18.** *For some environment parameters $\lambda \in (0, 1)$, $\Delta > 0$, and $A \in [1, K]$, consider an instance where (7) holds.*

$$
\mathbb{P}_{x \sim D_{\mathcal{X}}}\left( \mu\big(\{a \in \mathcal{A} : f^*(x, \pi_{f^*}(x)) - f^*(x, a) \leq \Delta\}\big) \leq A \right) \geq 1 - \lambda. \quad (62)
$$

*Suppose $\mathcal{W}_1$ and $\mathcal{W}_2$ hold. For all epochs $m$, suppose the action selection kernel is given by eq. (4), and suppose (2) holds for all $\bar{m} \in [m]$. Then for any epoch $m \in [m^*]$, we have (63) holds.*

$$
\mathbb{E}_{x \sim D_{\mathcal{X}}}\left[ \mu\Big(C_m(x, \beta, \eta)\Big) \right] \leq \Big(1 + A + K\lambda\Big) + 25\frac{K}{\Delta}\frac{\eta}{\beta}\sqrt{\alpha_{m-1}\xi_m}. \quad (63)
$$

*For any $\beta \in (0, 1/2]$ and $\eta \in [1, K]$.*

*Proof.* Consider any epoch $m \in [m^*]$. In this proof, for short-hand, let $C := \mathbb{E}[\mu(C_m(x, \beta, \eta))]$. We then have,

$$
\begin{aligned}
C &= \mathbb{E}[\mu(C_m(x, \beta, \eta))] \\
&\leq (A+1)P(\mu(C_m(x, \beta, \eta)) \leq A+1) + KP(\mu(C_m(x, \beta, \eta)) > A+1) \\
&\leq A + 1 + KP(\mu(C_m(x, \beta, \eta)) > A+1) \\
&\leq A + 1 + K - KP(\mu(C_m(x, \beta, \eta)) \leq A+1)
\end{aligned}
$$

The above immediately implies (64).

$$P\big(\mu(C_m(x,\beta,\eta)) \leq A+1\big) \leq \frac{A+1+K-C}{K}. \tag{64}$$

Let $\pi_0 \in \tilde{\Pi}$ be defined by (65).

$$\forall x \in \mathcal{X}, \ \pi_0(x) \in \underset{S \in \Sigma_1 | S \subseteq C_m(x,\beta,\eta)}{\arg\min} f^*(x,S). \tag{65}$$

Since $\pi_0$ only selects arms in $C_m(x,\beta,\eta)$, from Definition 2, we have (66).

$$\text{Reg}_{\hat{f}_m}(\pi_0) \leq \frac{\eta}{\beta} U_m. \tag{66}$$

We can lower bound the regret of $\pi_0$ as follows,

$$\begin{aligned}
&\text{Reg}_{f^*}(\pi_0)\\
&\geq P(f^*(x,\pi_{f^*}(x)) - f^*(x,\pi_0(x)) > \Delta) \cdot \Delta\\
&\overset{(i)}{=} P(\exists\, S \in \Sigma_1 | \, S \subseteq C_m(x,\beta,\eta), \ f^*(x,\pi_{f^*}(x)) - f^*(x,S) > \Delta) \cdot \Delta\\
&\overset{(ii)}{\geq} P\Big(\mu\big(C_m(x,\beta,\eta)\big) \geq A+1 \text{ and } \mu\big(\{a : (f^*(x,\pi_{f^*}(x)) - f^*(x,a) > \Delta\}\big) \geq K-A\Big) \cdot \Delta\\
&= P\Big(\mu\big(C_m(x,\beta,\eta)\big) \geq A+1 \text{ and } \mu\big(\{a : (f^*(x,\pi_{f^*}(x)) - f^*(x,a) \leq \Delta\}\big) \leq A\Big) \cdot \Delta\\
&= \Big(1 - P\Big(\mu\big(C_m(x,\beta,\eta)\big) < A+1 \text{ or } \mu\big(\{a : (f^*(x,\pi_{f^*}(x)) - f^*(x,a) \leq \Delta\}\big) > A\Big)\Big) \cdot \Delta\\
&\overset{(iii)}{\geq} \Big(1 - P\Big(\mu\big(C_m(x,\beta,\eta)\big) < A+1\Big) - P\Big(\mu\big(\{a : (f^*(x,\pi_{f^*}(x)) - f^*(x,a) \leq \Delta\}\big) > A\Big)\Big) \cdot \Delta\\
&= \Big(P\Big(\mu\big(\{a : (f^*(x,\pi_{f^*}(x)) - f^*(x,a) \leq \Delta\}\big) \leq A\Big) - P\Big(\mu\big(C_m(x,\beta,\eta)\big) < A+1\Big)\Big) \cdot \Delta\\
&\overset{(iv)}{\geq} \Big(1 - \lambda - \frac{A+1+K-C}{K}\Big)\Delta = \Big(\frac{C-A-1}{K} - \lambda\Big)\Delta.
\end{aligned} \tag{67}$$

where (i) is because by construction $\pi_0(x) \subseteq C_m(x,\beta,\eta)$ for all $x$, (ii) is by the fact that $\mu$ is a finite measure with $\mu(\mathcal{A}) =: K$, (iii) follows from union bound, and (iv) follows from (64) and (7).

We will now work towards upper bounding $\text{Reg}_{f^*}(\pi_0)$, and use this bound in conjunction with (67) to obtain our desired bound on $C$. To upper bound $\text{Reg}_{f^*}(\pi_0)$ using Lemma 14, we will upper bound $\text{Reg}_{\hat{f}_{m-1}}(\pi_0)$ and $\text{Reg}_{\hat{f}_{m-1}}(\pi_{f^*})$.

$$\begin{aligned}
\text{Reg}_{\hat{f}_{m-1}}(\pi_0) &\overset{(i)}{\leq} \frac{4}{3}\text{Reg}_{\Pi}(\pi_0) + 12\sqrt{\alpha_{m-2}\xi_{m-1}} + 6.5\sqrt{\xi_{m-1}}\\
&\overset{(ii)}{\leq} \frac{4}{3}\Big(\frac{4}{3}\text{Reg}_{\hat{f}_m}(\pi_0) + 12\sqrt{\alpha_{m-1}\xi_m} + 6.5\sqrt{\xi_m}\Big) + 12\sqrt{\alpha_{m-2}\xi_{m-1}} + 6.5\sqrt{\xi_{m-1}}\\
&\overset{(iii)}{\leq} \frac{16}{9}\text{Reg}_{\hat{f}_m}(\pi_0) + 28\sqrt{\alpha_{m-2}\xi_{m-1}} + \frac{91}{6}\sqrt{\xi_{m-1}}\\
&\overset{(iv)}{\leq} \frac{16}{9}\text{Reg}_{\hat{f}_m}(\pi_0) + \frac{259}{6}\sqrt{\alpha_{m-2}\xi_{m-1}}.
\end{aligned} \tag{68}$$

Where (i) and (ii) follow from Lemma 14, (iii) follows from $z_{m-1} = \sqrt{\frac{\alpha_{m-1}\xi_m}{\alpha_{m-2}\xi_{m-1}}} \leq 1$, and (iv) follows from $\alpha_{m-2} \geq 1$.

$$\begin{aligned}
\text{Reg}_{\hat{f}_{m-1}}(\pi_{f^*}) &\overset{(i)}{\leq} 12\sqrt{\alpha_{m-2}\xi_{m-1}} + 6.5\sqrt{\xi_{m-1}}\\
&\overset{(ii)}{\leq} \frac{37}{2}\sqrt{\alpha_{m-2}\xi_{m-1}}.
\end{aligned} \tag{69}$$

Where (i) follows from Lemma 14, (ii) follows from $\alpha_{m-2} \geq 1$.

$$\text{Reg}_{f^*}(\pi_0)$$
$$= R(\pi_{f^*}) - R(\pi_0)$$
$$= \big(R(\pi_{f^*}) - R_{\hat{f}_m}(\pi_{f^*})\big) - \big(R(\pi_0) - R_{\hat{f}_m}(\pi_0)\big) + \big(R_{\hat{f}_m}(\pi_{f^*}) - R_{\hat{f}_m}(\pi_0)\big)$$
$$\overset{(i)}{\leq} 2\sqrt{\alpha_{m-1}\xi_m} + \frac{1}{2}\sqrt{\alpha_{m-1}\xi_m} \sum_{\bar{m}\in[m]} \frac{1}{2\bar{m}^2 U_{\bar{m}}} \Big(\text{Reg}_{\hat{f}_{m-1}}(\pi_{f^*}) + \text{Reg}_{\hat{f}_{m-1}}(\pi_0)\Big) + \text{Reg}_{\hat{f}_m}(\pi_0)$$
$$\overset{(ii)}{\leq} 2\sqrt{\alpha_{m-1}\xi_m} + \frac{1}{40} \sum_{\bar{m}\in[m]} \frac{z_{\bar{m},m-1}}{2\bar{m}^2} \Big(\frac{16}{9}\text{Reg}_{\hat{f}_m}(\pi_0) + \Big(37 + 18.5*\frac{4}{3}\Big)\sqrt{\alpha_{m-2}\xi_{m-1}}\Big) + \text{Reg}_{\hat{f}_m}(\pi_0)$$
$$\overset{(iii)}{\leq} 3.6\sqrt{\alpha_{m-1}\xi_m} + \frac{47}{45}\text{Reg}_{\hat{f}_m}(\pi_0) \overset{(iv)}{\leq} \sqrt{\alpha_{m-1}\xi_m}\Big(3.6 + \frac{47}{45}*20\frac{\eta}{\beta}\Big) \leq 25\frac{\eta}{\beta}\sqrt{\alpha_{m-1}\xi_m}$$
$$(70)$$

Where (i) follows from Lemma 7. (ii) follows from (68), (69), and $U_{m-1} = 20\sqrt{\alpha_{m-2}\xi_{m-1}}$, (iii) follows from $z_{m-1} \leq 1$, and (iv) follows from (66) and $U_m = 20\sqrt{\alpha_{m-1}\xi_m}$. Finally, combining (67) and (70), we have,

$$\Big(\frac{C-A-1}{K} - \lambda\Big)\Delta \leq \text{Reg}_{f^*}(\pi_0) \leq 25\frac{\eta}{\beta}\sqrt{\alpha_{m-1}\xi_m}$$
$$\implies C \leq A + 1 + K\lambda + 25\frac{K}{\Delta}\frac{\eta}{\beta}\sqrt{\alpha_{m-1}\xi_m}.$$
$$(71)$$

$\square$

In Lemma 19, we use the bound from Lemma 18 to derive instance-dependent bounds on $\alpha_m$. Corollary 2 is an immediate implication of Lemma 19, and provides a bound on $\alpha_m$ that doesn't depend on $\alpha_{m-1}$. Finally, Corollary 2 is used to derive our instance-dependant bound on simple regret.

**Lemma 19.** *For some environment parameters $\lambda \in (0,1)$, $\Delta > 0$, and $A \in [1,K]$, consider an instance where (7) holds. Suppose $\mathcal{W}_1$ and $\mathcal{W}_2$ hold, and $\eta_m$ is chosen using (5). For all epochs $m$, suppose the action selection kernel is given by eq. (4), suppose eq. (17) holds, and suppose (2) holds for all $\bar{m} \in [m]$. Then for any epoch $m \in [m^*]$, we have (72) holds.*

$$\alpha_m \leq \mathcal{O}\Big(\max\Big(\sqrt{\frac{K\alpha_{m-1}}{\omega}}, A + K\lambda + \frac{\sqrt{K^3\omega\xi_m}}{\Delta}\Big)\Big)$$
$$(72)$$

*Proof.* Suppose $\eta_m \leq \sqrt{\frac{K\omega}{\alpha_{m-1}}} - 1/|S_{m-1,2}|$, we then have,

$$\frac{K}{\eta_m} \overset{(i)}{\leq} \frac{|S_{m-1,2}|+1}{|S_{m-1,2}|}\frac{K}{\eta_m + \frac{1}{|S_{m-1,2}|}}$$
$$\overset{(ii)}{\leq} \frac{|S_{m-1,2}|+1}{|S_{m-1,2}|}\lambda_m\Big(\eta_m + \frac{1}{|S_{m-1,2}|}\Big)$$
$$\overset{(iii)}{\leq} \frac{|S_{m-1,2}|+1}{|S_{m-1,2}|}\Big(1 + \mathbb{E}\Big[\mu\Big(C_m\Big(x_t, \beta_{\max}, \eta_m + \frac{1}{|S_{m-1,2}|}\Big)\Big)\Big] + \sqrt{\frac{2K^2\ln(8|S_{m-1,2}|m^2/\delta)}{|S_{m-1,2}|}}\Big)$$
$$\overset{(iv)}{\leq} \frac{|S_{m-1,2}|+1}{|S_{m-1,2}|}\Big((2+A+K\lambda) + 25\frac{K}{\Delta}\frac{\eta_m + \frac{1}{|S_{m-1,2}|}}{\beta_{\max}}\sqrt{\alpha_{m-1}\xi_m} + \sqrt{\frac{2K^2\ln(8|S_{m-1,2}|m^2/\delta)}{|S_{m-1,2}|}}\Big)$$
$$\overset{(v)}{\leq} \frac{|S_{m-1,2}|+1}{|S_{m-1,2}|}\Big((1+A+K\lambda) + \frac{50}{\Delta}\sqrt{K^3\omega\xi_m} + \sqrt{\frac{2K^2\ln(8|S_{m-1,2}|m^2/\delta)}{|S_{m-1,2}|}}\Big)$$
$$(73)$$

Where (i) follows from $\eta_m \geq 1$, (ii) follows from (5), (iii) follows from $\mathcal{W}_2$, (iv) follows from Lemma 18, and (v) follows from (5) and the fact that $\beta_{\max} = 0.5$. Finally, the result now follows from Lemma 5. $\square$

**Corollary 2.** *For some environment parameters $\lambda \in (0,1)$, $\Delta > 0$, and $A \in [1, K]$, consider an instance where (7) holds. Suppose $\mathcal{W}_1$ and $\mathcal{W}_2$ hold. For all epochs $m$, suppose the action selection kernel is given by eq. (4), suppose eq. (17) holds, and suppose suppose (2) holds for all $\bar{m} \in [m]$. Then for any epoch $m \in [m^*]$, we have (74) holds.*

$$\alpha_m \leq \mathcal{O}\left( \frac{K}{\omega} + A + K\lambda + \frac{\sqrt{K^3 \omega \xi_{m - \lceil \log_2 \log_2(K) \rceil}}}{\Delta} \right) \tag{74}$$

*Where for notational convenience, we let $\xi_i = 1$ for $i \leq 0$.*

*Proof.* By repeatedly applying Lemma 19, we have:

$$\alpha_m$$
$$\leq \mathcal{O}\left( \max\left( \left(\frac{K}{\omega}\right)^{\frac{1}{2} + \frac{1}{4} + \cdots + \frac{1}{2^{\lceil \log_2 \log_2(K) \rceil}}} K^{0.5 \lceil \log_2 \log_2(K) \rceil}, A + K\lambda + \frac{\sqrt{K^3 \omega \xi_{m - \lceil \log_2 \log_2(K) \rceil}}}{\Delta} \right) \right)$$
$$\overset{(i)}{\leq} \mathcal{O}\left( \max\left( \left(\frac{K}{\omega}\right) K^{0.5 \lceil \log_2 \log_2(K) \rceil}, A + K\lambda + \frac{\sqrt{K^3 \omega \xi_{m - \lceil \log_2 \log_2(K) \rceil}}}{\Delta} \right) \right)$$
$$\overset{(ii)}{\leq} \mathcal{O}\left( \max\left( \left(\frac{K}{\omega}\right), A + K\lambda + \frac{\sqrt{K^3 \omega \xi_{m - \lceil \log_2 \log_2(K) \rceil}}}{\Delta} \right) \right)$$
$$\leq \mathcal{O}\left( \frac{K}{\omega} + A + K\lambda + \frac{\sqrt{K^3 \omega \xi_{m - \lceil \log_2 \log_2(K) \rceil}}}{\Delta} \right)$$
$$\tag{75}$$

where (i) follows from $\sum_{i=1}^{\infty} 1/2^i = 1$, and (ii) follows from $K^{1/2 \lceil \log_2 \log_2(K) \rceil} \leq K^{1/2 \log_2 \log_2(K)} = K^{1/\log_2 K} = K^{\log_K 2} = 2$. $\qquad\square$

We now re-state and prove Theorem 2. As discussed earlier, this result relies on the bound in Corollary 2.

**Theorem 2.** *Suppose Assumptions 1 and 2 hold. For some $(\lambda, \Delta, A) \in [0,1] \times (0,1] \times [1, K]$, consider instances where for $1 - \lambda$ fraction of contexts at most $A$ arms are $\Delta$ optimal (i.e. (7) holds).*

$$\mathbb{P}_{x \sim D_{\mathcal{X}}}\left( \mu\left(\{a \in \mathcal{A} : f^*(x, \pi_{f^*}(x)) - f^*(x, a) \leq \Delta\}\right) \leq A \right) \geq 1 - \lambda. \tag{7}$$

*Let $m' = \min(\hat{m}, m(T)) - 1$. Let the learned policy $\hat{\pi}$ be given by (8) (equivalent to variance penalized policy optimization).*

$$\hat{\pi} \in \arg\max_{\pi \in \Pi} \hat{R}_{m(T)}(\pi) - \frac{1}{2}\sqrt{\alpha_{m'} \xi_{m(T)}} \sum_{\bar{m} \in [m']} \frac{\hat{R}_{m(T), \hat{f}_{\bar{m}}}(\pi_{\hat{f}_{\bar{m}}}) - \hat{R}_{m(T), \hat{f}_{\bar{m}}}(\pi)}{40 \bar{m}^2 \sqrt{\alpha_{\bar{m}-1} \xi_{\bar{m}}}}. \tag{8}$$

*Then with probability $1 - \delta$, $\omega$-RAPR has the following simple regret bound when $T$ samples.*

$$Reg_{\Pi}(\hat{\pi}) \leq \mathcal{O}\left( \sqrt{\alpha_{m'} \xi_{m(T)}} \right)$$
$$\leq \mathcal{O}\left( \sqrt{\xi_{m(T)} \min\left( K, A + K\lambda + \frac{K}{\omega} + \frac{K^{3/2} \omega^{1/2}}{\Delta} \sqrt{\xi_{\min(m^*, m(T)-1) - \lceil \log_2 \log_2(K) \rceil}} \right)} \right).$$

*Proof.* From Appendix B.1, both $\mathcal{W}_1$ and $\mathcal{W}_2$ hold with probability $1 - \delta$. We prove our simple regret bounds under these events. Let $m = m(T)$, we then have the following bound.

$$
R(\hat{\pi}) \overset{(i)}{\geq} \hat{R}_m(\hat{\pi}) - \sqrt{\alpha_{m'}\xi_m} - \frac{1}{2}\sqrt{\alpha_{m'}\xi_m} \sum_{\bar{m}\in[m']} \frac{\mathrm{Reg}_{\hat{f}_{\bar{m}}}(\hat{\pi})}{2\bar{m}^2 U_{\bar{m}}} - \frac{K\xi_m}{\eta_{m'}\min_{\bar{m}\in[m']} U_{\bar{m}}}
$$

$$
\overset{(ii)}{\geq} \hat{R}_m(\hat{\pi}) - \sqrt{\alpha_{m'}\xi_m} - \frac{1}{2}\sqrt{\alpha_{m'}\xi_m} \sum_{\bar{m}\in[m']} \frac{\widehat{\mathrm{Reg}}_{m,\hat{f}_{\bar{m}}}(\hat{\pi})}{2\bar{m}^2 U_{\bar{m}}} - \frac{2\sqrt{\alpha_{m'}}\xi_m}{U_{m'}} - \frac{\alpha_{m'}\xi_m}{U_{m'}}
$$

$$
\overset{(iii)}{\geq} \hat{R}_m(\pi^*) - \sqrt{\alpha_{m'}\xi_m} - \frac{1}{2}\sqrt{\alpha_{m'}\xi_m} \sum_{\bar{m}\in[m']} \frac{\widehat{\mathrm{Reg}}_{m,\hat{f}_{\bar{m}}}(\pi^*)}{2\bar{m}^2 U_{\bar{m}}} - \frac{2\sqrt{\alpha_{m'}}\xi_m}{U_{m'}} - \frac{\alpha_{m'}\xi_m}{U_{m'}} \qquad (76)
$$

$$
\overset{(iv)}{\geq} \hat{R}_m(\pi^*) - \sqrt{\alpha_{m'}\xi_m} - \frac{1}{2}\sqrt{\alpha_{m'}\xi_m} \sum_{\bar{m}\in[m']} \frac{\mathrm{Reg}_{\hat{f}_{\bar{m}}}(\pi^*)}{2\bar{m}^2 U_{\bar{m}}} - \frac{4\sqrt{\alpha_{m'}}\xi_m}{U_{m'}} - \frac{\alpha_{m'}\xi_m}{U_{m'}}
$$

$$
\overset{(v)}{\geq} R(\pi^*) - 2\sqrt{\alpha_{m'}\xi_m} - \sqrt{\alpha_{m'}\xi_m} \sum_{\bar{m}\in[m']} \frac{\mathrm{Reg}_{\hat{f}_{\bar{m}}}(\pi^*)}{2\bar{m}^2 U_{\bar{m}}} - \frac{4\sqrt{\alpha_{m'}}\xi_m}{U_{m'}} - \frac{2\alpha_{m'}\xi_m}{U_{m'}}
$$

$$
\overset{(vi)}{\geq} R(\pi^*) - 3.3\sqrt{\alpha_{m'}\xi_m}.
$$

Here (i) follows from Lemma 8. (ii) follows from Lemma 10, Lemma 5, and the fact that $U_{m'} \leq U_{\bar{m}}$ for any $\bar{m} \in [m']$. (iii) follows from (8). (iv) follows from Lemma 10. (v) follows from Lemma 8, Lemma 5, and the fact that $U_{m'} \leq U_{\bar{m}}$ for any $\bar{m} \in [m']$. Finally, (vi) follows from Lemma 14 and $U_{m'} \leq 20\sqrt{\alpha_{m'}\xi_m}$. Hence $\mathrm{Reg}_\Pi(\hat{\pi}) \leq \mathcal{O}(\sqrt{\alpha_{m'}\xi_m})$. Now the final bound follows from the fact that $\alpha_{m'} \leq \alpha_1 = 3K$, $\alpha_{m'} \leq \alpha_{\min(m^*,m(T)-1)}$, and Corollary 2. □

# D   Lower bound

**Theorem 3.** *Given parameters $K, F, T \in \mathbb{N}$ and $\phi \in [1, \infty)$. There exists a context space $\mathcal{X}$ and a function class $\mathcal{F} \subseteq (\mathcal{X} \times \mathcal{A} \to [0,1])$ with $K$ actions such that $|\mathcal{F}| \leq F$ and the following lower bound on cumulative regret holds:*

$$
\inf_{\mathbf{A}\in\Psi_\phi} \sup_{D\in\mathcal{D}} \mathbb{E}_D\left[ \sum_{t=1}^T \left( r_t(\pi^*(x_t)) - r_t(a_t) \right) \right] \geq \tilde{\Omega}\left( \sqrt{\frac{K}{\phi}} \sqrt{KT\log F} \right)
$$

*Here $(a_1, \ldots a_T)$ denotes the actions selected by an algorithm $A$. $\mathcal{D}$ denotes the set of environments such that $f^* \in \mathcal{F}$ and (7) hold with $(A, \lambda, \Delta) = (1, 0, 0.24)$. $\Pi$ denotes policies induced by $\mathcal{F}$. $\Psi_\phi$ denotes the set of CB algorithms that run for $T$ rounds and output a learned policy with a simple regret guarantee of $\sqrt{\phi\log F/T}$ for any instance in $\mathcal{D}$ with confidence at least $0.95$, i.e., $\Psi_\phi := \{A : \mathbb{P}(Reg(\hat{\pi}_\mathbf{A}) \leq \sqrt{\phi\log F/T}) \geq 0.95 \text{ for any instance in } \mathcal{D}\}$. Finally, $\tilde{\Omega}(\cdot)$ hides factors logarithmic in $K$ and $T$.*

We prove theorem 3 in the following sub-sections.

## D.1   Basic Technical Results

The following result is established in [31], with this version taken from the proof of Lemma D.2 in [13].

**Lemma 20** (Fano's inequality with reverse KL-divergence)**.** *Let*

$$
\mathcal{H} = (x_1, a_1, r_1(a_1)), \ldots, (x_T, a_T, r_T(a_T)),
$$

*and let $\{\mathbb{P}^{(i)}\}_{i\in[M]}$ be a collection of measures over $\mathcal{H}$, where $M \geq 2$. Let $\mathcal{Q}$ be any reference measure over $\mathcal{H}$, and let $\mathbb{P}$ be the law of $(m^*, \mathcal{H})$ under the following process:*

- *Sample $m^*$ uniformly from $[M]$.*

- *Sample $\mathcal{H} \sim \mathbb{P}^{(m^*)}$.*

*Then for any function $\hat{m}(\mathcal{H})$, if $\mathbb{P}(\hat{m} = m^*) \geq 1 - \delta$, then*

$$\left(1 - \frac{1}{M}\right) \log(1/\delta) - \log 2 \leq \frac{1}{M} \sum_{i=1}^{M} D_{KL}(Q || \mathbb{P}^{(i)}). \tag{77}$$

## D.2 Construction

If $K \leq 10$ or $T \leq 152^2 K \log F$ or $\phi \geq K$, our lower bound directly follows from the cumulative regret lower bound in [13]. Hence, without loss of generality, we can assume $K \geq 10, T \geq 152^2 K \log F$, and $\phi \leq K$.

The following construction closely follows lower bound arguments in [13]. Let $\mathcal{A} = \{a^{(1)}, a^{(2)}, \ldots, a^{(K)}\}$ be an arbitrary set of discrete actions. Let $k = \lfloor 1/\epsilon \rfloor$, and $d$ be parameters that will be fixed later. With $\epsilon \in (0, 1)$, note that $1/(2\epsilon) \leq k \leq 1/\epsilon$. We will now define the context set $\mathcal{X}$ as the union of $d$ disjoint partitions $\mathcal{X}^{(1)}, \mathcal{X}^{(2)}, \ldots, \mathcal{X}^{(d)}$, where $\mathcal{X}^{(i)} = \{x^{(i,0)}, x^{(i,1)}, \ldots, x^{(i,k)}\}$ for all $i \in [d]$. Hence, we have $\mathcal{X} = \cup \mathcal{X}^{(i)}$ and $|\mathcal{X}| = d(k+1)$.

For each partition index $i \in [d]$, we construct a policy class $\Pi^{(i)} \subseteq (\mathcal{X}^{(i)} \to \mathcal{A})$ as follows. First we let $\pi^{(i,0)} : \mathcal{X}^{(i)} \to \mathcal{A}$ be the policy that always selects arm $a^{(1)}$, and let $\pi^{(i,l,b)} : \mathcal{X}^{(i)} \to \mathcal{A}$ be defined as follows for all $l \in [k]$ and $b \in \mathcal{A}_0 := \mathcal{A} \setminus \{a^{(1)}\}$,

$$\forall x^{(i,j)} \in \mathcal{X}^{(i)}, \ \pi^{(i,l,b)}(x^{(i,j)}) = \begin{cases} a^{(1)}, & \text{if } j \neq l, \\ b, & \text{if } j = l. \end{cases} \tag{78}$$

Construct $\Pi^{(i)} := \{\pi^{(i,l,b)} | l \in [k] \text{ and } b \in \mathcal{A}_0\} \cup \{\pi^{(i,0)}\}$.[20] Finally, we let $\Pi := \Pi^{(1)} \times \Pi^{(2)} \times \cdots \times \Pi^{(d)}$. We will now construct a reward model class $\mathcal{F}$ that induces $\Pi$.

Let $\Delta := 1/4$. For each partition index $i \in [d]$, we construct a reward model class $\mathcal{F}^{(i)} \subseteq (\mathcal{X}^{(i)} \times \mathcal{A} \to [0, 1])$ as follows. First we let $f^{(i,0)} : \mathcal{X}^{(i)} \times \mathcal{A} \to [0, 1]$ be defined as follows,

$$\forall (x^{(i,j)}, a) \in \mathcal{X}^{(i)} \times \mathcal{A}, \ f^{(i,0)}(x^{(i,j)}, a) = \begin{cases} \frac{1}{2} + \Delta, & \text{if } a = a^{(0)}, \\ \frac{1}{2}, & \text{if } a \in \mathcal{A}_0. \end{cases} \tag{79}$$

For all $l \in [k]$ and $b \in \mathcal{A}_0$, we define $f^{(i,l,b)} : \mathcal{X}^{(i)} \times \mathcal{A} \to [0, 1]$ as follows,

$$\forall (x^{(i,j)}, a) \in \mathcal{X}^{(i)} \times \mathcal{A}, \ f^{(i,l,b)}(x^{(i,j)}, a) = \begin{cases} \frac{1}{2} + \Delta, & \text{if } a = a^{(0)} \\ \frac{1}{2} + 2\Delta, & \text{if } j = l \text{ and } a = b, \\ \frac{1}{2}, & \text{otherwise.} \end{cases} \tag{80}$$

Note that $f^{(i,l,b)}$ differs from $f^{(i,0)}$ only at context $(x^{(i,l)}, b)$. Construct $\mathcal{F}^{(i)} := \{f^{(i,l,b)} | l \in [k] \text{ and } b \in \mathcal{A}_0\} \cup \{f^{(i,0)}\}$. Finally, we let $\mathcal{F} := \mathcal{F}^{(1)} \times \mathcal{F}^{(2)} \times \cdots \times \mathcal{F}^{(d)}$. Hence, we have,

$$|\mathcal{F}| = |\mathcal{F}^{(i)}|^d \leq (k \cdot K)^d \implies d \geq \frac{\log |\mathcal{F}|}{\log(K \cdot k)} \geq \frac{\log |\mathcal{F}|}{\log(K/\epsilon)}. \tag{81}$$

We choose $d$ to be the largest value such that $F \geq (k \cdot K)^d$. Hence we choose $d = \lfloor \log F / \log(K \cdot k) \rfloor \geq \log F / (2 \log(K \cdot k))$.

To use lemma 20, we will describe a collection of environments that share a common distribution over contexts and only differ in the reward distribution. The context distribution $D_{\mathcal{X}}$ is given by $D_{\mathcal{X}} := \frac{1}{d} \sum_i D_{\mathcal{X}}^{(i)}$, where $D_{\mathcal{X}}^{(i)}$ is a distribution over $\mathcal{X}^{(i)}$, with $\epsilon$ probability of sampling each context in $\mathcal{X}^{(i)} \setminus \{x^{(i,0)}\}$, and $1 - k\epsilon$ probability of sampling the context $x^{(i,0)}$.

For each block $\mathcal{X}^{(i)}$, we let $\mathbb{P}^{(i,0)}$ denote the law given by the reward distribution $r(a) \sim \text{Ber}(f^{(i,0)}(x, a))$ for all $x \in \mathcal{X}^{(i)}$. Further, for any $l \in [k]$ and $b \in \mathcal{A}_0$, we let $\mathbb{P}^{(i,l,b)}$ denote the law given by the reward distribution $r(a) \sim \text{Ber}(f^{(i,l,b)}(x, a))$ for all $x \in \mathcal{X}^{(i)}$. For any policy $\pi \in \Pi^{(i)}$, we let $R^{(i,l,b)}(\pi) = \mathbb{E}_{\mathbb{P}^{(i,l,b)}}[r(\pi(x))]$ denote expected reward under $\mathbb{P}^{(i,l,b)}$, and let $\text{Reg}^{(i,l,b)}(\pi) = R^{(i,l,b)}(\pi^{(i,l,b)}) - R^{(i,l,b)}(\pi)$ denote expected simple regret under $\mathbb{P}^{(i,l,b)}$.

---

[20] Here $[k] = \{1, 2, \ldots, k\}$

We use $\rho$ to index environments. Here $\rho = (\rho_1, \ldots, \rho_d)$, where $\rho_i = (l_i, b_i)$ for $l_i \in \{0, 1, \ldots, k\}$ and $b_i \in \mathcal{A}_0$. We let $\mathbb{P}_\rho$ denote an environment with the law $\mathbb{P}^{(i,l_i,b_i)}$ for contexts in $\mathcal{X}^{(i)}$.[21] Finally let $\pi_\rho$ denote the optimal policy under $\mathbb{P}_\rho$, and let $\pi_\rho{}^{(i)}$ denote its restriction to $\mathcal{X}^{(i)}$. Let $\mathbb{E}_\rho[\cdot]$ denote the expectation under $\mathbb{P}_\rho$. Let $R_\rho(\pi) = \mathbb{E}_\rho[r(\pi(x))]$ denote the expected reward of $\pi$ under $\mathbb{P}_\rho$, and let $\mathrm{Reg}_\rho(\pi) = R_\rho(\pi_\rho) - R_\rho(\pi)$ denote the simple regret of $\pi$ under $\mathbb{P}_\rho$.

### D.3 Lower bound argument

We sample $\rho$ from a distribution $\nu$ defined as follows. For each $i \in [d]$, set $l_i = 0$ with probability 0.5, otherwise $l_i$ is selected uniformly from $[k]$. Select $b_i$ uniformly from $\mathcal{A}_0$. Note that when $l_i = 0$, we disregard the value of $b_i$.

We let $\hat{\pi}_{\mathbf{A}} \in \Pi$ denotes the policy recommended by the contextual bandit algorithm $\mathbf{A}$ at the end of $T$ rounds, and let $\hat{\pi}_{\mathbf{A}}^{(i)} \in \Pi^{(i)}$ be the restriction of $\hat{\pi}_{\mathbf{A}}$ to block $\mathcal{X}^{(i)}$. Note that the policy recommended by $\mathbf{A}$ will depend on the environment $\rho$.

Let $\mathcal{I} := \{i \in [d] | \mathrm{Reg}^{(i,l_i,b_i)}(\pi_{\mathbf{A}}{}^{(i)}) \leq 19\sqrt{\phi \log F/T}\}$. Since we only consider algorithms that guarantee the following with probability at least $19/20$,

$$\frac{1}{d} \sum_{i=1}^{d} \mathrm{Reg}^{(i,l_i,b_i)}(\pi_{\mathbf{A}}{}^{(i)}) = \mathrm{Reg}_\rho(\hat{\pi}_{\mathbf{A}}) \leq \sqrt{\phi \log F/T}. \tag{82}$$

Under this event, we have that at most $d/19$ block indices satisfy $\mathrm{Reg}^{(i,l_i,b_i)}(\pi_{\mathbf{A}}{}^{(i)}) > 19\sqrt{\phi \log F/T}$. Therefore, we have $|\mathcal{I}| \geq 18d/19$.
Define event $M_i = \{i \in \mathcal{I}\}$. We have

$$\sum_{i=1}^{d} P(M_i) = \sum_{i=1}^{d} \mathbb{E}[1\{i \in \mathcal{I}\}] \geq \frac{19}{20} \mathbb{E}[|\mathcal{I}||\mathrm{Reg}_\rho(\hat{\pi}_{\mathbf{A}}) \leq \sqrt{\phi \log F/T}] \geq \frac{9d}{10}. \tag{83}$$

Consider any fixed index $i$, under the event $M_i$, we have the following. First observe for any $(l, b) \neq (l', b')$, we have $\mathrm{Reg}^{(i,l,b)}(\pi^{(i,l',b')}) \geq \epsilon\Delta$. Let $(\hat{l}_i, \hat{b}_i)$ be indices such that $\pi_{\mathbf{A}}{}^{(i)} = \pi^{(i,\hat{l}_i,\hat{b}_i)}$. We now choose $\epsilon$ such that,

$$\epsilon = \frac{38}{\Delta}\sqrt{\frac{\phi \log F}{T}} \iff \frac{\epsilon\Delta}{2} = 19\sqrt{\frac{\phi \log F}{T}}. \tag{84}$$

Hence from definition of $\mathcal{I}$, we have $\mathrm{Reg}^{(i,l_i,b_i)}(\pi_{\mathbf{A}}{}^{(i)}) \leq \epsilon\Delta/2$. Further since $\mathrm{Reg}^{(i,l_i,b_i)}(\pi) \geq \epsilon\Delta$ for all $\pi \in \Pi^{(i)} \setminus \{\pi^{(i,l_i,b_i)}\}$, we have $(\hat{l}_i, \hat{b}_i) = (l_i^*, b_i^*)$.

Restating the above result, we have the following. For any $i \in \mathcal{I}$, with probability $1 - 1/16$, we have $(\hat{l}_i, \hat{b}_i) = (l_i^*, b_i^*)$. Hence from lemma 20, we have,

$$\left(1 - \frac{1}{(K-1)k}\right) \log\left(1/P(\overline{M_1})\right) - \log 2$$

$$\leq \frac{1}{(K-1)k} \sum_{l=1}^{k} \sum_{b \in \mathcal{A}_0} D_{KL}(\mathbb{P}^{(i,0)} || \mathbb{P}^{(i,l,b)})$$

$$\overset{(i)}{=} \frac{1}{(K-1)k} \sum_{l=1}^{k} \sum_{b \in \mathcal{A}_0} D_{KL}(\mathrm{Ber}(1/2) || \mathrm{Ber}(1/2 + 2\Delta)) \underset{\mathbb{P}^{(i,0)}}{\mathbb{E}}[|\{t | x_t = x^{(i,l)}, a_t = b\}|] \tag{85}$$

$$\overset{(ii)}{\leq} \frac{1}{(K-1)k} \sum_{l=1}^{k} \sum_{b \in \mathcal{A}_0} 4\Delta^2 \underset{\mathbb{P}^{(i,0)}}{\mathbb{E}}[|\{t | x_t = x^{(i,l)}, a_t = b\}|]$$

$$= \frac{4\Delta^2}{(K-1)k} \underset{\mathbb{P}^{(i,0)}}{\mathbb{E}}[|\{t | x_t \in \mathcal{X}^{(i)} \setminus \{x^{(i,0)}\}, a_t \in \mathcal{A}_0\}|].$$

---

[21] Here $\mathbb{P}^{(i,0)} \equiv \mathbb{P}^{(i,0,b)}$ for all $b \in \mathcal{A}_0$.

Where (i) follows from the fact that $\mathbb{P}^{(i,0)}$ and $\mathbb{P}^{(i,l,b)}$ are identical unless $x_t = x^{(i,l)}$ and $a_t = b$, and (ii) follows from $\Delta \leq 1/4$. Clearly we have:

$$
\underset{\rho \sim \nu}{\mathbb{E}} \underset{\rho}{\mathbb{E}} \left[ \sum_{t=1}^{T} \left( r_t(\pi^*(x_t)) - r_t(a_t) \right) \right]
$$

$$
\geq \Delta \underset{\rho \sim \nu}{\mathbb{E}} \underset{\rho}{\mathbb{E}} \left[ \sum_{t=1}^{T} \sum_{i=1}^{d} \mathbb{I}\left( \left\{ x_t \in \mathcal{X}^{(i)} \setminus \{x^{(i,0)}\}, a_t \in \mathcal{A}_0, l_i = 0 \right\} \right) \right]
$$

$$
\overset{(i)}{\geq} \frac{\Delta}{2} \sum_{i=1}^{d} \underset{\mathbb{P}^{(i,0)}}{\mathbb{E}} \left[ \left| \left\{ t \mid x_t \in \mathcal{X}^{(i)} \setminus \{x^{(i,0)}\}, a_t \in \mathcal{A}_0 \right\} \right| \right]
$$

$$
\overset{(ii)}{\geq} \frac{\Delta}{2} \sum_{i=1}^{d} \left( - \left( 1 - \frac{1}{k(K-1)} \right) \log \left( P(\overline{M_i}) \right) - \log 2 \right) \cdot \frac{(K-1)k}{4\Delta^2}
$$

$$
= \sum_{i=1}^{d} \left( - \left( 1 - \frac{1}{k(K-1)} \right) \log \left( 1 - P(M_i) \right) - \log 2 \right) \cdot \frac{(K-1)k}{8\Delta} \tag{86}
$$

$$
\overset{(iii)}{\geq} \sum_{i=1}^{d} \left( \left( 1 - \frac{1}{k(K-1)} \right) P(M_i) - \log 2 \right) \cdot \frac{(K-1)k}{8\Delta}
$$

$$
\overset{(iv)}{\geq} \left\{ \left( 1 - \frac{1}{k(K-1)} \right) \frac{9}{10} - \log 2 \right\} \cdot \frac{(K-1)kd}{8\Delta}
$$

$$
\overset{(v)}{\geq} \frac{Kkd}{100\Delta} \overset{(vi)}{\geq} \frac{dK}{200\Delta\epsilon} \overset{(vii)}{=} \frac{1}{7600} \sqrt{\frac{Td^2K^2}{\phi \log F}} \overset{(viii)}{\geq} \frac{1}{15200} \sqrt{\frac{K^2 T \log F}{\phi \log^2(K \cdot k)}}
$$

$$
\overset{(ix)}{\geq} \frac{1}{15200} \sqrt{\frac{K^2 T \log F}{\phi \log^2(K \cdot T)}}.
$$

Where (i) follows from the fact that $\nu(l_i = 0) = 1/2$, (ii) follows from (85) and that $|\mathcal{I}| \geq d/2$, (iii) uses $\log(1 + x) \leq x$, for $x > -1$, , (iv) uses (83), (v) follows from $k \geq 1$ and $K \geq 10$, (vi) follows from $k \geq 1/(2\epsilon)$, (vii) follows from choice of $\epsilon$, (viii) follows from (81), and (ix) since $k \leq 1/\epsilon \overset{84}{=} \frac{1}{152} \sqrt{\frac{T}{\phi \log F}} \leq T$. This completes the proof of theorem 3.

# E    Additional Details

## E.1    Conformal Arm Sets

The below lemma shows that, for any given policy $\pi$, the conformal arm sets given in definition 2 can be probabilistically relied on (over the distribution of contexts) to contain arms recommended by $\pi$, with low regret under the models estimated up to epoch $m$. Recall we earlier define $U_m = 20\sqrt{\alpha_{m-1}\xi_m}$.

**Lemma 21** (Conformal Uncertainty). *For any policy $\pi$ and epoch $m$, we have:*

$$
\underset{x \sim D_{\mathcal{X}}, a \sim \pi(\cdot|x)}{\Pr}(a \in C_m(x, \zeta)) \geq 1 - \zeta \sum_{\bar{m} \in [m]} \frac{Reg_{\hat{f}_{\bar{m}}}(\pi)}{(2\bar{m}^2)U_{\bar{m}}} \tag{87}
$$

*Proof.* For any policy $\pi$, we have (88) holds.

$$
\Pr_{x \sim D_{\mathcal{X}}, a \sim \pi(\cdot|x)} (a \notin C_m(x, \zeta))
$$

$$
\leq \Pr_{x \sim D_{\mathcal{X}}, a \sim \pi(\cdot|x)} \Big( \bigcup_{\bar{m} \in [m]} \{a \notin \tilde{C}_{\bar{m}}(x, \zeta/(2\bar{m}^2))\} \Big)
$$

$$
\overset{(i)}{\leq} \sum_{\bar{m} \in [m]} \Pr_{x \sim D_{\mathcal{X}}, a \sim \pi(\cdot|x)} \Big( \hat{f}_{\bar{m}}(x, \pi_{\hat{f}_{\bar{m}}}(x)) - \hat{f}_{\bar{m}}(x, a) > \frac{(2\bar{m}^2)U_{\bar{m}}}{\zeta} \Big) \tag{88}
$$

$$
\overset{(ii)}{\leq} \sum_{\bar{m} \in [m]} \frac{\mathrm{Reg}_{\hat{f}_{\bar{m}}}(\pi)}{(2\bar{m}^2)U_{\bar{m}}/\zeta}.
$$

Where (i) follows from union bound and (ii) follows from Markov's inequality. $\qquad\square$

Recall that right after Lemma 14, we show that $\mathrm{Reg}_{\hat{f}_{\bar{m}}} \leq U_{\bar{m}}$ with high-probability for any $\bar{m} \in [\hat{m}]$. Hence, Lemma 21 gives us that with high-probability we have $\Pr_{x \sim D_{\mathcal{X}}, a \sim \pi^*(\cdot|x)}(a \in C_m(x, \zeta)) \geq 1 - \zeta$. While we don't directly use Lemma 21, this lemma helps demonstrate the utility of CASs.

### E.2 Argument for Surrogate Objective

Lemma 22 is a self-contained result proving that guarantying tighter bounds on the optimal cover leads to tighter simple regret bounds for any contextual bandit algorithm. Hence the optimal cover is a valid surrogate objective for simple regret. This lemma is not directly used in the analysis of $\omega$-RAPR, however similar results (see Theorem 2) were proved and used. Note that the parameters below (including $\alpha$) are not directly related to parameters maintained by $\omega$-RAPR.

**Lemma 22** (Valid Surrogate Objective). *Suppose $\Pi$ is a finite class and suppose a contextual bandit algorithm collects $T$ samples using kernels $(p_t)_{t \in [T]}$ such that $p_t(\cdot|\cdot) \geq \sqrt{\frac{\ln(4|\Pi|/\delta)}{\alpha T}}$. Further suppose that the following condition holds with some $\alpha \in [1, \infty)$:*

$$
\frac{1}{T} \sum_{t=1}^{T} V(p_t, \pi^*) \leq \alpha \tag{89}
$$

*Then we can estimate a policy $\hat{\pi} \in \Pi$ such that with probability at least $1 - \delta$, we have:*

$$
|R(\pi^*) - R(\hat{\pi})| \leq \mathcal{O}\bigg( \sqrt{\frac{\alpha \ln(4|\Pi|/\delta)}{T}} \bigg). \tag{90}
$$

*Proof.* WOLG we assume $T \geq \ln(4|\Pi|/\delta)$, since otherwise the result trivially holds. Now consider any policy $\pi$. Let $y_t := \frac{r_t \mathbb{I}(\pi(x_t) = a_t)}{p_t(\pi(x_t)|x_t)}$. Now note that:

$$
\mathrm{Var}_t[y_t] \leq \mathbb{E}_{D(p_t)}[y_t^2] = \mathbb{E}_{(x_t, a_t, r_t) \sim D(p_t)} \bigg[ \frac{r_t^2 \mathbb{I}(\pi(x_t) = a_t)}{p_t^2(\pi(x_t)|x_t)} \bigg] \leq \mathbb{E}_{x \sim D_{\mathcal{X}}} \bigg[ \frac{1}{p_t(\pi(x_t)|x_t)} \bigg] = V(p_t, \pi). \tag{91}
$$

Then from from a Freedman-style inequality [See theorem 13 in 8], we have with probability at least $1 - \delta/(2|\Pi|)$ that the following holds:

$$
\bigg| \sum_{t=1}^{T} (y_t - R(\pi)) \bigg| \leq 2 \max \bigg\{ \sqrt{\sum_{t=1}^{T} \mathrm{Var}(y_t) \ln(4|\Pi|/\delta)}, \frac{\ln(4|\Pi|/\delta)}{\sqrt{\frac{\ln(4|\Pi|/\delta)}{\alpha T}}} \bigg\}
$$

$$
\overset{(i)}{\Longrightarrow} \bigg| \frac{1}{T} \sum_{t=1}^{T} \frac{r_t \mathbb{I}(\pi(x_t) = a_t)}{p_t(\pi(x_t)|x_t)} - R(\pi) \bigg| \leq 2 \sqrt{\frac{\ln(4|\Pi|/\delta)}{T} \max \bigg\{ \frac{1}{T} \sum_{t=1}^{T} V(p_t, \pi), \alpha \bigg\}} \tag{92}
$$

Here (i) follows from (91). Similarly with probability at least $1 - \delta/(2|\Pi|)$ the following holds:

$$
\left| \frac{1}{T} \sum_{t=1}^{T} \frac{1}{p_t(\pi(x_t)|x_t)} - \frac{1}{T} \sum_{t=1}^{T} V(p_t, \pi) \right|
$$

$$
\overset{(i)}{\leq} \frac{2}{T} \max \left\{ \sqrt{\sum_{t=1}^{T} \mathrm{Var}\left( \frac{1}{p_t(\pi(x_t)|x_t)} \right) \ln(4|\Pi|/\delta)}, \frac{\ln(4|\Pi|/\delta)}{\sqrt{\frac{\ln(4|\Pi|/\delta)}{\alpha T}}} \right\} \tag{93}
$$

$$
\overset{(ii)}{\leq} \frac{2}{T} \max \left\{ \sqrt{T \frac{\alpha T}{\ln(4|\Pi|/\delta)} \ln(4|\Pi|/\delta)}, \sqrt{\alpha T \ln(4|\Pi|/\delta)} \right\} \overset{(iii)}{\leq} 2\sqrt{\alpha} \overset{(iv)}{\leq} 2\alpha.
$$

Here (i) follows from Freedman's inequality, (ii) follows from the lower bound on $p_t$, (iii) follows from $T \geq \ln(4|\Pi|/\delta)$, and (iv) follows from $\alpha \geq 1$. Hence the above events hold with probability at least $1 - \delta$ for all policies $\pi \in \Pi$. Now let $\hat{\pi}$ be given as follows.

$$
\hat{\pi} \in \arg\max_{\pi \in \Pi} \frac{1}{T} \sum_{t=1}^{T} \frac{r_t \mathbb{I}(\pi(x_t) = a_t)}{p_t(\pi(x_t)|x_t)} - 2\sqrt{\frac{\ln(4|\Pi|/\delta)}{T} \left( 2\alpha + \frac{1}{T} \sum_{t=1}^{T} \frac{1}{p_t(\pi(x_t)|x_t)} \right)} \tag{94}
$$

We then have the following lower bound on $R(\hat{\pi})$ using the definition of $\hat{\pi}$ and the above to high-probability events.

$$
R(\hat{\pi}) \overset{(i)}{\geq} \frac{1}{T} \sum_{t=1}^{T} \frac{r_t \mathbb{I}(\hat{\pi}(x_t) = a_t)}{p_t(\hat{\pi}(x_t)|x_t)} - 2\sqrt{\frac{\ln(4|\Pi|/\delta)}{T} \max \left\{ \frac{1}{T} \sum_{t=1}^{T} V(p_t, \hat{\pi}), \alpha \right\}}
$$

$$
\overset{(ii)}{\geq} \frac{1}{T} \sum_{t=1}^{T} \frac{r_t \mathbb{I}(\hat{\pi}(x_t) = a_t)}{p_t(\hat{\pi}(x_t)|x_t)} - 2\sqrt{\frac{\ln(4|\Pi|/\delta)}{T} \left( 2\alpha + \frac{1}{T} \sum_{t=1}^{T} \frac{1}{p_t(\hat{\pi}(x_t)|x_t)} \right)} \tag{95}
$$

$$
\overset{(iii)}{\geq} \frac{1}{T} \sum_{t=1}^{T} \frac{r_t \mathbb{I}(\pi^*(x_t) = a_t)}{p_t(\pi^*(x_t)|x_t)} - 2\sqrt{\frac{\ln(4|\Pi|/\delta)}{T} \left( 2\alpha + \frac{1}{T} \sum_{t=1}^{T} \frac{1}{p_t(\pi^*(x_t)|x_t)} \right)}
$$

$$
\overset{(iv)}{\geq} R(\pi^*) - 4\sqrt{\frac{\ln(4|\Pi|/\delta)}{T} \left( 4\alpha + \frac{1}{T} \sum_{t=1}^{T} V(p_t, \pi^*) \right)} \geq R(\pi^*) - 4\sqrt{\frac{5\alpha \ln(4|\Pi|/\delta)}{T}}
$$

Here (i) follows from (92), (ii) follows from (93), (iii) follows from (94), and (iv) follows from (92) and (93). This completes the proof. $\qquad\square$

### E.3 Testing Misspecification via CSC

We restate the misspecification test that is used at the end of epoch $m$ and argue how this test can be solved via two calls to a cost sensitive classification solver. First, let us restate the test in (96).

$$
\max_{\pi \in \Pi \cup \{p_{m+1}\}} |\hat{R}_{m+1, \hat{f}_{m+1}}(\pi) - \hat{R}_{m+1}(\pi)| - \sqrt{\alpha_m \xi_{m+1}} \sum_{\bar{m} \in [m]} \frac{\hat{R}_{m+1, \hat{f}_{\bar{m}}}(\pi_{\hat{f}_{\bar{m}}}) - \hat{R}_{m+1, \hat{f}_{\bar{m}}}(\pi)}{40 \bar{m}^2 \sqrt{\alpha_{\bar{m}-1} \xi_{\bar{m}}}}
$$

$$
\leq 2.05 \sqrt{\alpha_m \xi_{m+1}} + 1.1 \sqrt{\xi_{m+1}}, \tag{96}
$$

We are interested in calculating the value of the maximization problem in (96). To calculate this maximum, we need to fix our estimators. Let $\hat{R}_{m+1, f}(\pi) := \frac{1}{|S_{m,3}|} \sum_{t \in S_{m,3}} f(x_t, \pi(x_t)) = \frac{1}{|S_{m,3}|} \sum_{t \in S_{m,3}} \mathbb{E}_{a \sim \pi(\cdot|x_t)} f(x_t, a)$ for any policy $\pi$ and reward model $f$, which is the only obvious estimator we could think off for $R_f(\pi)$. Also let us use IPS estimaton for policy evaluation (the same argument works for DR), $\hat{R}_{m+1}(\pi) := \frac{1}{|S_{m,3}|} \sum_{t \in S_{m,3}} \frac{\pi(a_t|x_t) r_t(a_t)}{p_m(a_t|x_t)}$. [22] [23] Note that the value of

---

[22] When evaluating a general kernel $q$, we use the natural extension of these estimators of policy value. In particular, simply replace $\pi(\cdot|x)$ with $q(\cdot|x)$ in their formulas.

[23] Up to constant factors, these estimators give us the best rates in Assumption 2 with finite classes. These estimators are also used in several contextual bandit papers [e.g., 2, 26].

the maximization problem in (96) is equal to $\max(L_1, L_2, L_3)$, where $\{L_i | i \in [3]\}$ are defined as follows.

$$L_1 := \max_{\pi \in \Pi} \hat{R}_{m+1,\hat{f}_{m+1}}(\pi) - \hat{R}_{m+1}(\pi) - \sqrt{\alpha_m \xi_{m+1}} \sum_{\bar{m} \in [m]} \frac{\hat{R}_{m+1,\hat{f}_{\bar{m}}}(\pi_{\hat{f}_{\bar{m}}}) - \hat{R}_{m+1,\hat{f}_{\bar{m}}}(\pi)}{40\bar{m}^2 \sqrt{\alpha_{\bar{m}-1}\xi_{\bar{m}}}}$$

$$L_2 := \max_{\pi \in \Pi} \hat{R}_{m+1}(\pi) - \hat{R}_{m+1,\hat{f}_{m+1}}(\pi) - \sqrt{\alpha_m \xi_{m+1}} \sum_{\bar{m} \in [m]} \frac{\hat{R}_{m+1,\hat{f}_{\bar{m}}}(\pi_{\hat{f}_{\bar{m}}}) - \hat{R}_{m+1,\hat{f}_{\bar{m}}}(\pi)}{40\bar{m}^2 \sqrt{\alpha_{\bar{m}-1}\xi_{\bar{m}}}}$$

$$L_3 := |\hat{R}_{m+1,\hat{f}_{m+1}}(p_{m+1}) - \hat{R}_{m+1}(p_{m+1})| - \sqrt{\alpha_m \xi_{m+1}} \sum_{\bar{m} \in [m]} \frac{\hat{R}_{m+1,\hat{f}_{\bar{m}}}(\pi_{\hat{f}_{\bar{m}}}) - \hat{R}_{m+1,\hat{f}_{\bar{m}}}(p_{m+1})}{40\bar{m}^2 \sqrt{\alpha_{\bar{m}-1}\xi_{\bar{m}}}}$$

(97)

Note that $L_3$ doesn't involve any optimization and can be easily calculated. Substituting value of these estimators for $L_1$ and $L_2$, we get.

$$L_1 = \max_{\pi \in \Pi} \sum_{t \in S_{m,3}} \frac{1}{|S_{m,3}|} \left( \hat{f}_{m+1}(x_t, \pi(x_t)) - \frac{\pi(a_t|x_t)r_t(a_t)}{p_m(a_t|x_t)} \right.$$
$$\left. - \sqrt{\alpha_m \xi_{m+1}} \sum_{\bar{m} \in [m]} \frac{\hat{f}_{\bar{m}}(x_t, \pi_{\hat{f}_{\bar{m}}}(x_t)) - \hat{f}_{\bar{m}}(x_t, \pi(x_t))}{40\bar{m}^2 \sqrt{\alpha_{\bar{m}-1}\xi_{\bar{m}}}} \right)$$

(98)

$$L_2 = \max_{\pi \in \Pi} \sum_{t \in S_{m,3}} \frac{1}{|S_{m,3}|} \left( \frac{\pi(a_t|x_t)r_t(a_t)}{p_m(a_t|x_t)} - \hat{f}_{m+1}(x_t, \pi(x_t)) \right.$$
$$\left. - \sqrt{\alpha_m \xi_{m+1}} \sum_{\bar{m} \in [m]} \frac{\hat{f}_{\bar{m}}(x_t, \pi_{\hat{f}_{\bar{m}}}(x_t)) - \hat{f}_{\bar{m}}(x_t, \pi(x_t))}{40\bar{m}^2 \sqrt{\alpha_{\bar{m}-1}\xi_{\bar{m}}}} \right)$$

Clearly, both $L_1$ and $L_2$ are cost-sensitive classification problems [see 21, for problem definition]. In both, we need to find a policy (classifier) that maps contexts to arms (classes), incurring a score (cost) for each decision such that the total score (cost) is maximized (minimized). Hence the misspecification test we use only requires two calls to CSC solvers.

### E.4 Simulation

We ran uniform RCT, LinUCB, LinTS, 1-RAPR, and 4-RAPR with linear function classes and an exploration horizon of 5000 on a synthetic data generating process (DGP).[24]

**Data generating process.** We consider four arms, i.e., $\mathcal{A} = [8]$. The context $x = (x_1, x_2)$ is uniformly sampled from four regions on the two-dimensional unit ball; and in specific, $x$ is generated via the following distribution:

1. $\tilde{x}_1 \sim \text{Uniform}(0.8, 1.0)$
2. $\tilde{x}_2 = \sqrt{1 - \tilde{x}_1^2} \cdot z$, where $z \sim \text{Uniform}\{-1, 1\}$.
3. Sample region index $r \sim \text{Uniform}\{0, 1, 2, 3\}$:
   - if $r = 0$: $\{x_1, x_2\} = \{\tilde{x}_1, \tilde{x}_2\}$.
   - if $r = 1$: $\{x_1, x_2\} = \{\tilde{x}_2, \tilde{x}_1\}$.
   - if $r = 2$: $\{x_1, x_2\} = \{-\tilde{x}_1, -\tilde{x}_2\}$.
   - if $r = 3$: $\{x_1, x_2\} = \{-\tilde{x}_1, -\tilde{x}_2\}$.
4. The reward for each arm is $0.4$ plus a linear function of the contexts. The linear parameters for the 8 arms are $\{(a, b) | |a| + |b| = 1, |a|, |b| \in \{0, 0.4, 0.6, 1\}\}$. Hence, the conditional expected rewards lies in the range $[0.2, 0.6]$. Finally, the noise was sampled uniformly at random from $[-0.4, 0.4]$.

---

[24]The LinUCB scaling parameter was set to a default of 0.25, we similarly let $\sqrt{\xi(T, 0.5)} = 0.25 \times \sqrt{d/T}$. We also set the bloated constant of 20 in Definition 2 to be 1.

A simulation run takes less than 9 seconds for any of these algorithms on a laptop with 16GB RAM and an Apple M1 Pro chip, demonstrating the computational tractability of this approach. We provide a scatter plot (aggregating results from 50 runs) showing (i) the value of the average reward during exploration (as a proxy for cumulative regret, $x$-axis) and (ii) the value of the learned policy at the end of the experiment (as a proxy for simple regret of learned policy, $y$-axis). On simple regret, we see that 4-RAPR $\approx$ 1-RAPR $>$ RCT $>$ LinUCB $>$ LinTS. On cumulative regret, we see that LinUCB $>$ 1-RAPR $>$ 4-RAPR $>$ LinTS $>$ RCT. The RAPR algorithms achieve the best simple regret perfor-

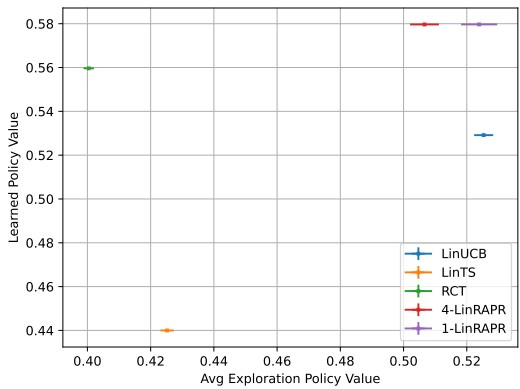

mance and achieve competitive performance on cumulative regret for this DGP. However, the fact that both RAPR algorithms learn policies of similar values suggests that our CASs are larger than necessary for at least some risk levels on this DGP. Further refining CAS is an important direction of future work.