# OpenReview forum: "Proportional Response: Contextual Bandits for Simple and Cumulative Regret Minimization"
_NeurIPS.cc/2023/Conference — NeurIPS 2023 poster_

### Official Review · Reviewer_G9eA · 2023-07-02

**Soundness:** 2 fair
**Presentation:** 2 fair
**Contribution:** 2 fair
**Rating:** 5
**Confidence:** 4

**Summary:**

The paper proposes a novel contextual bandit algorithm that balances cumulative and simple regret guarantees using a surrogate objective and optimal cover. By constructing the conformal arms sets (CASs) which improve the coverage guarantees for the optimal policy, the algorithm efficiently finds the policy that minimizes simple regret while balancing with the cumulative regret. Cumulative and simple regret upper bounds are considered in the case when the model can be misspecified and the hardness result shows that the trade-off between simple and cumulative regret is unavoidable.

**Strengths:**

The paper provides a theoretical analysis of the balance and trade-off between cumulative and simple regrets, which are not considered in the previous literature. The definition of cover and conformal arm sets is new and paves the way to characterize simple regret minimization. Generalizing regret bounds to the case when the model can be misspecified extends the currently existing literature.

**Weaknesses:**

W1. More discussions on theoretical results are required. Specifically, (a) the role of the action selection kernel q in Assumption 2, (b) what happens when $\min_{(x,a)\in\mathcal{X}\times\mathcal{A}} p(a|x)=0$in Assumption 2 (line 180), (c) how the bias $B$ is multiplied to $\sqrt{1/T}$ term rather than $O(1)$, (d) examples of the setting that satisfies the assumptions in Theorem 3.

W2. The experiment section does not include the balance of cumulative regret and simple regret, which is the key part of the paper. More diverse settings are required to evaluate the algorithm.

**Questions:**

Q1. My main concern is in Theorem 3. The assumption in Theorem 3 implies a positive gap between the optimal and the second-best arm (e.g., Section 5.2 in [1] ) which enables the algorithm to attain zero simple regrets after $O(\log T)$ exploration rounds and thus logarithmic cumulative regret bound. Does this result contradict Theorem 3?

Q2. I couldn't understand the sentence "it (the algorithm) does not utilize the structure of the policy class being explored" in line 349. Could you elaborate more on this?

Q3. In line 109, the integral in $f(x,S)=\int_{a} f(x,a) d\mu(a) / \mu(S)$ ranges over $S$? Should it be $f(x,S)=\int_{S} f(x,a) d\mu(a) / \mu(S)$?


[1] Abbasi-Yadkori, Yasin, Dávid Pál, and Csaba Szepesvári. "Improved algorithms for linear stochastic bandits." Advances in neural information processing systems 24 (2011).

**Limitations:**

Yes. This work is mostly theoretical and the negative societal impact of the work is unseen.

---

> ### Author Rebuttal · Authors · 2023-08-09
>
> Q1: The results in [1] do not contradict Theorem 3. We will make this clear in our discussion. The log(T) cumulative regret guarantee in [1] relies on the linear realizability assumption, while our framework here addresses more general model classes where the instance-dependent cumulative regret bounds are impossible to obtain unless the bound depends exponentially on the policy class complexity, as pointed out in [2]. We provide a detailed explanation below.
> - Our lower bound in Theorem 3 indeed considers environments with a positive gap between the optimal and the second-best arm at every context. Under linear realizability, [1] develops an algorithm with gap-dependent cumulative regret guarantees, implying strong simple regret guarantees. While the environments we consider in Theorem 3 satisfy realizability, the model class $\mathcal{F}$ we construct is not linear. Hence, this is not a contradiction.
> - It is worth noting that linear realizability is a strong structural condition that facilitates the design and analysis of some optimistic algorithms. For a general function class $\mathcal{F}$, even under realizability, [2] show a lower bound of $\Omega(|\mathcal{F}|/\Delta)$ on cumulative regret. Since this lower bound is much larger than $O(\log|\mathcal{F}|/\Delta)$, in a reasonably strong sense, meaningful gap-dependent cumulative regret bounds are impossible in general. This also implies that gap-dependent simple regret guarantees can not be achieved as a byproduct of gap-dependent cumulative regret guarantees. Theorem 3 strengthens this observation by showing that cumulative regret guarantees get worse than minimax rates when attempting to attain gap-dependent simple regret bounds.
>
> [1] Abbasi-Yadkori, Yasin, Dávid Pál, and Csaba Szepesvári. "Improved algorithms for linear stochastic bandits." Advances in neural information processing systems 24 (2011).
>
> [2] Dylan J Foster and Alexander Rakhlin. Beyond UCB: Optimal and efficient contextual bandits with regression oracles. International Conference on Machine Learning (ICML), 2020.
>
> Q2: Beyond policy class complexity, our algorithm does not leverage any additional structure in the policy class we want to explore. For example, consider the class of policies that recommend the same arm at every context ($\{\pi|\pi(x)=\pi(x’),\forall x,x’\}$). Consider an arm with a high-expected reward for some sub-population of the context distribution but a low estimated average reward (under the context and reward distribution). This arm need not be considered for further exploration at any context. However, CASs do not leverage such structure and will continue accounting for cases where this arm may perform well for some sub-population. Ideally, we should use information regarding the explored policy class to reduce the size of CASs we rely on. If it were feasible in terms of memory/compute, we would ideally maintain a set of potentially optimal policies and then restrict our exploration at any context to arms that correspond to one of the policies in this set.
>
> Q3: You are right, it should be $f(x,S):=\int_{a\in S}f(x,a)d\mu(a)/\mu(S)$.
>
> W1 [More discussions on theoretical results]: We thank the reviewer for pointing out our limitations on the exposition, and we will address all these points in the updated paper.
> - (a)  [the role of ... kernel q in Assumption 2,] In assumption 2, the action selection kernel q is an input to the evaluation oracle. The oracle adds q to the set of policies it evaluates. Evaluating a single extra kernel does not significantly increase the statistical complexity of evaluating these policies. Our algorithm adds the action selection kernel for the next epoch ($p_{m+1}$) as this additional input to the evaluation oracle. Evaluating $p_{m+1}$ helps us verify that our reward model class's bias did not negatively affect the construction of $p_{m+1}$.
> - (b)[ what happens … Assumption 2 (line 180),]  We will clarify in Assumption 2 that we treat $1/0$ as $\infty$. Hence, if $\min_{(x,a)\in\mathcal{X}\times\mathcal{A}}p(a|x)=0$, then the evaluation guarantees of Assumption 2 are trivial. Assumption 2 specifies the requirements of the evaluation sub-routine we use. The weaker these requirements are, the easier they are to satisfy. The reason we let the bounds in Assumption 2 depend on $1/\min_{(x, a)\in\mathcal{X}\times\mathcal{A}}p(a|x)$ is to (i) allow usage of policy evaluation methods like inverse propensity scores that average random variables with inverse propensity terms, and (ii) allow for confidence bounds that require these random variables be bounded.
> - (c) [how the bias is multiplied] In the definition of simple regret (line 118), we compare the learned policy with the best policy in the class $\Pi$. Note that $\Pi$ may not contain the universal optimal policy $\pi_{f^*}$ induced by the ground truth reward model $f^*$. Hence the bound in Theorem 2 can go to zero asymptotically and does not depend on the bias of the policy class. Theorem 2 depends on the bias of the reward model class $B$ because our exploration relies on the estimated reward model. However, this bias can only reduce the benefits of our adaptive exploration.
> In contrast, our definition of cumulative regret (line 123) compares our action selection kernels used for exploration against the universal optimal policy $\pi_{f^*}$. Hence Theorem 1 has a $O(\sqrt{\omega KB}T)$ term. We work with different definitions of cumulative and simple regret because we use a regression-based approach that tries to estimate $f^*$ for exploration; however, we use cost-sensitive classification for learning the final policy. It is worth noting that there is no difference in our definitions if we assume realizability.
> - (d) [examples ... Theorem 3] The proof of Theorem 3 explicitly constructs environments that satisfy the conditions of Theorem 3 and demonstrate the tradeoff.
>
> W2: We are adding plots showing balance between the objectives. See joint response above.

---

> > ### Author Response · Authors · 2023-08-14
> >
> > We hope this message finds you well. We would greatly appreciate the opportunity to address any further questions or concerns you might have regarding the validity of our theoretical findings. Kindly refer to our response to Q1 for further clarity. Thank you for your time and consideration.

---

> ### Comment · Reviewer_G9eA · 2023-08-15
>
> I appreciate the helpful responses to my questions and comments. My main concern about the correctness of Theorem 3 is resolved. But I cannot change the score more than `borderline accept' based on the following reasons:
>
> (1) The trade-off between simple regret minimization (SRM) and cumulative regret minimization (CRM) in Theorem 3 holds for (not *all*) but *some* contexts space, function class, and set of environments. It is not sure how many kinds of environments have such a trade-off. Although it might be impossible to prove a similar result for *all* kinds of environments, it would be better if the authors could provide some examples of the environments (other than the one in the proof) to intuitively understand how many kinds of environments have the inevitable trade-offs. But for now, Theorem 3 itself does not support the trade-off between SRM and CRM for general cases and the authors should clearly mention this limitation. Because this limitation could make it difficult to understand why the trade-off between SRM and CRM is challenging and I think there should be more explanation about the trade-off in the main text.
>
> (2) In the response, I understand that ``gap-dependent simple regret guarantees can not be achieved as a byproduct of gap-dependent cumulative regret guarantees.''. But is it possible that gap-dependent cumulative regret bounds can be achieved by gap-dependent simple regret bounds on some function classes or environments? I might miss some points, but still, I am not convinced why achieving both SRM and CRM is challenging and the trade-off analysis is significant.
>
> (3) While rechecking the paper, I found some presentation and clarity issues as Reviewer VVeg mentioned. It took a long time to find definitions for the terms in the algorithm and in the proof. I think the presentation should be improved to provide an accurate understanding of the analysis.

---

> > ### Author Response · Authors · 2023-08-15
> >
> > We thank the reviewer for their valuable time and for engaging with us in this discussion.
> > - Yes, it should be possible to achieve better guarantees for specific environments.
> > - Our paper studies general-purpose contextual bandits with any user-specified model/policy class and does not impose additional assumptions like realizability. While this does help with the generality of Theorem 1 and 2 (cumulative and simple regret guarantees), we agree that restricting the environments we study can lead to better bounds and less stringent trade-offs between simple/cumulative regret.
> > - The lower bound (Theorem 3) only helps show that the trade-off between Theorem 1 and 2 is close to the best we can hope for without additional restrictions.
> > - Identifying minimal additional restrictions to ensure better guarantees is an excellent direction for future work.

---

### Official Review · Reviewer_FhEq · 2023-07-05

**Soundness:** 3 good
**Presentation:** 3 good
**Contribution:** 3 good
**Rating:** 7
**Confidence:** 4

**Summary:**

This paper studies contextual bandits and develops a flexible algorithm that can be adapted to both cumulative regret minimization and simple regret minimization. New techniques are developed to analyze the proposed algorithm, e.g., constructing the conformal arm sets. The authors also prove a negative result stating that one cannot simultaneously achieve optimal simple regret and cumulative regret.

**Strengths:**

This paper proposes a novel algorithm called RAPR, which utilizes two computational subroutines, an estimation subroutione and an evaluation subroutine. The RAPR algorithm takes a hyperparameter as input, which can be tuned to adapt to either cumulative regret minimization or simple regret minimization. This result seems quite interesting to me. The authors also complement the theoretical upper bound with a lower bound stating that one cannot simultaneously achieve optimal cumulative regret bound and simple regret bound.

**Weaknesses:**

I have questions regarding some wording used in the paper. More specifically:
1. The paper claims that they develop a new computationally efficient algorithm. However, it seems that the proposed algorithm is not entirely efficient under model misspecification since a cost-sensitive classification (CSC) subroutine is used in this case. It is known that CSC oracle can be NP-hard to implement even in simple cases.
2. The paper claims instance-dependent guarantees for simple regret minimization. However, it seems that the guarantees stated in Theorem 2 have a worst-case flavor, compared to instance-dependent results stated in Li et al, 2022.

**Questions:**

See the weaknesses part.

**Limitations:**

As stated on Page 9.

---

> ### Author Rebuttal · Authors · 2023-08-09
>
> **Regarding claims of computational efficiency:** We thank the reviewer for raising this point and apologize for the confusion. The term “computationally efficient” is indeed ambiguous. In the revised version, we will be more precise in our explanation by emphasizing that we only make $O(1)$ oracle calls at the end of every epoch and do not need cost-sensitive classification (CSC) under realizability. Below is a detailed explanation:
> - As discussed in lines 280-281, we don't need CSC if we assume realizability or only focus on the cumulative regret objective.
> - To the best of our knowledge, under model misspecification, the computational challenge of solving problems like CSC is unavoidable in the policy learning literature (e.g. [2], see discussion in Section 2). Even policy learning under RCT (randomized control trials) data requires a CSC solver in settings without realizability assumptions. Despite CSC being NP-hard, several heuristic CSC solvers (e.g., vowpalwabbit) and exact brute-force solvers (e.g., policytree) have been developed.
> - Noting this, papers that propose regression-free algorithms (e.g., [1]) claim computational efficiency by arguing that they only make polynomially many calls to such CSC solvers at every epoch (under doubling epochs). However, as the epochs get large, the number of CSC problems they need to solve and outputs they need to save grows to be impractically large.
> - In contrast, we only make two calls to CSC solvers at the end of every epoch. While this is a computationally expensive step, our approach represents a sizable improvement in terms of computational efficiency over prior work. The improvement is sizable enough to make our algorithm sufficiently computationally tractable/efficient to be implemented (even under model misspecification).
>
> **Regarding the instance-dependence claim:** Thank you for allowing us to clarify why the guarantees in Theorem 2 are instance dependent.
> - Our regression-based approach exploits the gap between optimal and sub-optimal arms under the true conditional expected reward model $f^*$. Most prior literature on instance-dependent guarantees (e.g., [3], see Section 5.2) attempts to use similar gaps between arms under the true reward model.
> - Eq 8 (see line 306 in Thm 2) considers instances where for at least $1-\lambda$ fraction of contexts from the context distribution, the instance has at most $A$ arms that are $\Delta$-optimal. Notably, while RAPR does not need these parameters as input, the guarantees of Theorem 2 surpass  minimax guarantees by leveraging these instance-dependent parameters.
> - There is a worst-case component $\sqrt{Kd/T}$ in our simple regret bound in Theorem 2. The reason for this is that the RAPR’s upper bound on simple regret highly relies on the size of the conformal arm set. By design, the set’s size is at most $K$ but can be smaller than $K$ when using RAPR in instances that are characterized in Theorem 2.
> - [1] propose a regression-free algorithm. Their upper bound is stated in terms of the value of a min-max optimization problem which makes our guarantees hard to compare. Both our bounds seem similar with regard to utilizing gaps between optimal and sub-optimal arms under $f^*$. Nevertheless, the value of the optimization problem in their upper bound can further exploit structure in the explored policy class. We will better emphasize this point as a limitation of our regression-based approach in the revised version.
>
>
> [1] Zhaoqi Li, Lillian Ratliff, Houssam Nassif, and Kevin Jamieson. Instance-optimal PAC Algorithms for Contextual Bandits. Advances in neural information processing systems. 2022.
>
> [2] Wang, Lequn, Akshay Krishnamurthy, and Aleksandrs Slivkins. "Oracle-Efficient Pessimism: Offline Policy Optimization in Contextual Bandits." arXiv preprint arXiv:2306.07923 (2023).
>
> [3] Abbasi-Yadkori, Yasin, Dávid Pál, and Csaba Szepesvári. "Improved algorithms for linear stochastic bandits." Advances in neural information processing systems 24 (2011).

---

> > ### Comment · Reviewer_FhEq · 2023-08-18
> > **After rebuttal**
> >
> > I thank the authors for their response. I suggest the authors add these related discussion into the main paper during revision. I'd like to keep my score and lean towards accepting the paper.

---

### Official Review · Reviewer_X2Xo · 2023-07-05

**Soundness:** 3 good
**Presentation:** 3 good
**Contribution:** 4 excellent
**Rating:** 8
**Confidence:** 4

**Summary:**

This paper focus on the continues objective between simple regret minmization (SRM) and cumulative regret minimization (CRM) in contextual bandits, and propose an algorithm that can be either optimal in SRM, or optimal in CRM. Meanwhile, a new uncertainty quantification is propsed. Numerical simulations validate the efficacy.

**Strengths:**

 - Novelty: the paper focus on the objective between best arm identification (BAI) and regret minimization (RM), which is rarely studied but of great practical significance, as most works focus on either RM or BAI.
- Technical contribution: the conformal arm set as a new type of confidence bound is new, and is of independent interests beyond the topic.
- The results are strong, as a single algorithm can optimally change from one objective to another is not easy. It's also valuable for multi-objective related topics.

**Weaknesses:**

- The geometic covering in the algorithm (doubling the epochs) is common in RM, however, when it comes to simple regret minimization, there might be the case that only limited round $T$ is given, and thus requires an any-time algorithm.

- $\beta_{max}$ as the threshold of $\beta$ constrains that $\beta$ can not be too big, and restrains the smallest possible CAS. I'm confused with the counter-intuitive parameter, as for the objective of BAI, CAS should be as small as possible eventually. Why do we need the trucation threshold and why it's selected as $1/4$ should be explained.

- Though it is mentioned that the optimal cover can be seen as the surrogate objective for BAI, it is mainly used to estimate arbitrary policy $\pi$, while how it can be optimized to achieve a better $\pi$ is not clear.

**Questions:**

See weaknesses above.

**Limitations:**

\

---

> ### Author Rebuttal · Authors · 2023-08-02
>
> We thank the reviewer for their encouraging remarks.
>
> **Regarding doubling epochs:** We thank the reviewer for raising this point and will discuss doubling epochs as a limitation of our algorithm. To clarify, our algorithm does not assume a fixed given $T$. If we pause exploration in the middle of an epoch, then the simple regret guarantees in Theorem 2 depend on the data collected in the previous epoch. Hence while doubling epochs lead to larger constants in our simple/cumulative regret guarantees (Note the previous epoch contains at least one-fourth of the total number of samples), our algorithm is still anytime. That is, doubling epochs makes our algorithm less adaptive, leading to worse constants in our theoretical bounds. This is an artifact of us using offline oracles (requiring large batches of i.i.d. data). While needing only offline oracles improves the generality of our algorithm, using online oracles would help us avoid the need to use doubling epochs and would make our algorithm more adaptive. Developing better-optimized algorithms for the fixed budget setting could be an interesting direction for future research.
>
> **Clarifying the role of the optimal cover:** We apologize we did not make this clearer. As briefly discussed in the paper, optimal cover is a surrogate objective for simple regret. We approximately minimize an upper bound on the optimal cover when choosing $\eta_{m+1}$ (See eq 5 at line 270.). This step is critical to the simple regret guarantees of Theorem 2.
> - Eq 49 (line 635-636) shows that the optimal cover $V(p_{m+1},\pi^*)$ is upper bounded by $\frac{E_{x\sim D_X}[|C_{m+1}(x,\beta_{\max}/\eta_{m+1})|]}{1-\beta_{\max}}+\frac{K}{\eta_{m+1}}$.
> - Empirically, this quantity is further upper bounded by  $\frac{\lambda_{m+1}(\eta_{m+1})}{1-\beta_{\max}}+\frac{K}{\eta_{m+1}}$.
> - We approximately minimize this objective by selecting $\eta_{m+1}$ to be the largest $\eta$ that satisfies $\lambda_{m+1}(\eta)\leq K/\eta$ (subject to constraints needed to achieve the desired cumulative regret). Since $\lambda_{m+1}(\eta)$ is non-decreasing with $\eta$, we can show that $\eta_{m+1}$ approximately optimizes the above objective.
> - The proof of Theorem 2 (see section C) critically relies on the fact that our algorithm ensures the optimal cover is well-bounded.
> - We will expand on this in the paper.
>
> **Regarding the need for thresholding with $\beta_{\max}$:** We apologize we did not make this clearer. For simple regret minimization, two factors are of primary concern: (a) the size of the CAS we use and (b) does the set contain the optimal arm. Thresholding helps with (b).
> - When the optimal arm lies in $C_m(x,\beta_{\max}/\eta_m)$, thresholding ensures the optimal arm is sampled with probability at least $(1-\beta_{\max})/|C_m(x,\beta_{\max}/\eta_m)|$ (See eq 14 in the proof of lemma 3 in section B.2.). Without thresholding, this probability may be smaller since the optimal arm may not lie in smaller sets ($C_m(x,\beta/\eta_m)$ with $\beta>\beta_{\max}$). If the threshold parameter is too close to one, the numerator term ($1-\beta_{\max}$) in the above bound would be trivial.
> - Hence, in our analysis, thresholding helps lock in the benefits of $E[|C_m(x,\beta_{\max}/\eta_m)|]$ being small (see proof of Lemma 3) by avoiding the risk of the optimal arm not lying in smaller sets.
> - The numerator term of $1-\beta_{\max}$ appears due to selecting $\beta$ proportionally. Eq 13 in the proof of Lemma 3 shows that selecting $\beta$ proportionally helps bound the probability of sampling the optimal arm when the optimal arm does not lie in $C_m(x,\beta_{\max}/\eta_m)$.
> - Our analysis can be made to work with $\beta_{\max}$ set to be any constant that is not too close to zero or one. The current heuristic choice of $1/4$ can be further optimized in our analysis. It may be that thresholding with $\beta_{\max}$ is an artifact of our analysis and can be avoided with other proof techniques.
> - We will expand our discussion of this in the paper.

---

> > ### Comment · Reviewer_X2Xo · 2023-08-16
> >
> > Thanks for clarifying, and good luck with the final decision.

---

### Official Review · Reviewer_VVeg · 2023-07-19

**Soundness:** 2 fair
**Presentation:** 1 poor
**Contribution:** 2 fair
**Rating:** 6
**Confidence:** 3

**Summary:**

This paper provides a computational efficient algorithm for contextual bandit that is capable of trading off between near-optimal minimax cumulative regret guarantees and instance-dependent simple regret guarantees. Additionally, It handles model misspecification, works in the continuous arm setting, and provides a lower bound that shows there is always a trade off in achieving simple regret and the cumulative regret.

**Strengths:**

- A significant contribution in achieving a trade-off between simple regret and cumulative regret.

- One advantage of the algorithm is that it is regression based which makes it computationally efficient, while it does not rely on the realizability assumption, as it can adapt to model misspecification.


**Weaknesses:**

The weaknesses mainly lie in the paper's poor writing and presentation, as well as in the insufficiently conducted simulations, detailed as bellow.

- The motivation in the introduction lacks conviction and the main story is postponed to the related work section.

- The introduction lacks a clear image of the achieved bounds and the bounds of related works, which could have been presented together in a single table along with all limitations.

- in Section Preliminaries, the notations are listed in the footnote, which make it difficult for readers to follow the setup. In general, the flow in this section is confusing and needs improvement. Moreover, the authors used $\Pi$ and $\mathcal{F}$ in the "objectives" subsection without defining them before in the problem setup. Also definition of "CB algorithm" is missing.

- Lack of clarity regarding the specific location in the appendix for referenced proofs and details.

- Algorithm section is tedious with numerous links to various definitions, making it challenging to follow the flow. Besides that, the provided intuition is not sufficient for a non-expert reader to understand the complex formulas used for variables such as $\lambda$, $\eta$, and $\alpha$. This can hinder comprehension for readers. Moreover, line 15 of algorithm $\omega$-RAPR seems to be incorrect.

- There are many arXiv papers listed in the references, whereas for most of them there are already the published version in some journal or conference.

- The simulation part is extremely brief and lacks comparison to existing baselines, as well as an illustration of the results.

I believe the paper requires a major revision due to various identified weaknesses.




**Questions:**

- I request the authors to elaborate further on the optimality of the achieved trade-off between simple regret and cumulative regret. Additionally, I am curious to understand how the provided lower bound sheds light on this trade-off and its implications in the context of the paper's contributions.

---

> ### Author Rebuttal · Authors · 2023-08-09
>
> **Regarding theoretical guarantees and optimality of the achieved trade-off:** The algorithm RAPR comes with a tuning parameter $\omega\in[1,K]$. This tuning parameter helps us trade-off guarantees on simple and cumulative regret. We discuss our theoretical results and recap the optimality of our trade-off.
> - The simple regret guarantees of RAPR are never worse than the minimax optimal rates (Theorem 2). Depending on the instance, RAPR achieves simple regret guarantees that are up to $O(1/\sqrt{\omega})$ times smaller compared to minimax optimal rates (Theorem 2). This improvement factor of $O(1/\sqrt{\omega})$ over minimax optimal rates is asymptotically achieved for instances where realizability holds and where the gap between the best and second best arm in terms of conditional expected reward is at least $\Delta>0$ at every context (best-case instance in Theorem 2). RAPR achieves these instance-dependent guarantees without being provided any instance information as input to the algorithm.
> - Unfortunately, the corresponding cumulative regret for the above instances is a factor of $O(\sqrt{\omega})$ times larger compared to minimax optimal rates (Theorem 1). The cumulative regret guarantees of our algorithm only degrade relative to the minimax optimal rate if the instance allows for better simple regret guarantees.
> - Our lower bound (Theorem 3) considers the instances described above with $\Delta=0.24$ (the gap between the best and second best arm in terms of conditional expected reward). Theorem 3 shows that (by setting $\alpha=\Theta(K/\omega)$ in the Theorem statement) if an algorithm bounds simple regret on these instances by $O(1/\sqrt{\omega})$ times minimax optimal rates, the cumulative regret is lower bounded by $\Omega(\sqrt{\omega})$ times minimax optimal rates. Note that RAPR achieves this trade-off when the number of rounds $T$ is large enough. Hence in this sense, our algorithm achieves the optimal trade-off between guarantees on simple vs cumulative regret when $T$ is large enough.
>
> **Regarding simulations:** We are adding plots comparing with other baselines and showing the balance between the objectives. See the joint response above.
>
> **A new outline on the intro::** We thank the reviewer for their valuable feedback on the introduction. Reflecting on this, we have decided to rewrite the introduction with the following outline:
> - We study the stochastic contextual bandit setting and consider two goals. Define/motivate simple regret and cumulative regret. This paper proposes a new algorithm, RAPR, that has a tuning parameter $\omega\in[1, K]$ that governs the weight placed on the two objectives. Discuss the theoretical guarantees of RAPR and optimality of achieved trade-off (detailed earlier).
> - Define regression-based approaches and discuss computational benefits of regression-based approaches over the algorithm proposed in [1]. State that RAPR is the first general-purpose (works with any user-specified reward/policy class) regression-based algorithm with attractive simple regret guarantees.
> - We next describe the RAPR algorithm in more detail. (We will include similar intuition for the algorithms section)
>   - We first define a surrogate objective for simple regret, the optimal cover, which is inversely proportional to the probability that the (unknown) optimal policy is used by the bandit. The optimal cover bounds the variance of evaluating the unknown optimal policy under our exploration policy. This surrogate objective can be minimized by appropriately designing our exploration policy/action selection kernels.
>   - Since RAPR is regression-based, we cannot construct an explicit distribution over policies as that distribution would have a large support and would be computationally and memory intensive to maintain. We need to be able to minimize the optimal cover by directly constructing a distribution over arms for each arriving context.
>   - To reason about how well we explore the unknown optimal policy, having a general purpose uncertainty quantification at each context would be helpful. Much of the existing literature constructs  UCB-style confidence intervals, but existing approaches to constructing them rely on assumptions like linear realizability. For general function classes, these intervals may be too wide and are often computationally expensive to construct.
>   - To overcome this issue, we develop conformal arm sets (CASs), which are a set of potentially optimal arms at each context. This uncertainty quantification is regression-based and computationally easy to construct, is general-purpose, and shrinks at ``fast rates” (with square-root dependency on expected squared error bounds for regression). Unfortunately, these sets come with some risk of not containing the arm recommended by the optimal policy at every context. Nevertheless, we can use this uncertainty quantification to construct a distribution over arms at each context that helps us minimize the optimal cover by balancing the benefits and risks of relying on these CASs. The unavoidable trade-off between our simple and cumulative regret guarantees is an artifact of these risky sets.
>   - Beyond allowing us to trade off simple and cumulative regret guarantees, the flexibility of the approach also helps us extend to the continuous arm setting and allows us to handle model misspecification.
> - We then briefly discuss other related work that is outside the scope of this ``main story".
>
> **Other writing comments:** We will fix the issue regarding our reference containing old arXiv papers. We have improved the intuition provided in the algorithms section and will give our variables intuitive names to help with the flow. We will add proper references to the specific locations in the appendix. We will also improve the flow and readability of the preliminaries.
>
> [1] Zhaoqi Li, Lillian Ratliff, Houssam Nassif, and Kevin Jamieson. Instance-optimal PAC Algorithms for Contextual Bandits

---

> ### Comment · Reviewer_VVeg · 2023-08-15
> **Score Adjustment: One-Point Increase**
>
> Thank the authors for addressing my questions. I appreciate their effort to improve the clarity of the manuscript. I am therefore increasing my score by one point.

---

### Author Rebuttal · Authors · 2023-08-10

We thank all the reviewers for their valuable feedback.

**Regarding feedback on simulations:** We ran an additional simulation to compare our algorithm (RAPR) against other baselines and to better understand the balance between the two objectives we study. On a randomly generated linearly realizable data-generating process (DGP) with eight arms, two-dimensional context, and an exploration horizon of $40000$ rounds; we simulate LinUCB, BootstrapLinTS (linear thompson sampling), RCT (uniformly sampling arms), and $\omega$-RAPR with $\omega\in \{1,8\}$.
- Since we are comparing against algorithms based on linear realizability, we used linear realizability to further refine the size of CASs used in our implementation. Note that tightening the uncertainty quantification we rely on does not break our theoretical guarantees. For the other algorithms, we use vowpalwabbit implementations.
- We provide a scatter plot showing (i) the value of the learned policy for each algorithm (as a proxy for simple regret of learned policy) and (ii) the value of the last exploration policy used by each algorithm (as a proxy for regret incurred during exploration). We compare policy values to make it easier to see the trade-off. The current plot averages over results from 10 simulation runs.
  - The ordering for the value of the learned policy is $8$-RAPR > $1$-RAPR > RCT > LinUCB > BootstrapLinTS.
  - The ordering for the value of the last exploration policy is LinUCB > $1$-RAPR > $8$-RAPR > BootstrapLinTS > RCT. As one might expect, cumulative rewards also have the same ordering.
  - Hence we see some trade-off between the two objectives (less stark than Theorem 3 since we restrict attention to the linear case).
  - We also see that the algorithmic techniques introduced in this paper can help practitioners achieve better policy learning guarantees. However, algorithms like LinUCB can outperform RAPR on cumulative reward.
  - Finally, we note that each simulation run of RAPR took about 40 seconds on an M1 Macbook Pro laptop. Demonstrating the computational efficacy of our approach.
- Data-generating process (DGP): Contexts are 2 dimensional, and number of arms is 8. The DGP was randomly generated and satisfies linear realizability. At every round, each feature of the context is sampled from $[0,0.5]$. The final reward is the average of a uniform noise term sampled from [0,1], a linear reward, and an arm-specific reward. The linear reward parameters for each arm are sampled from a uniform distribution and are normalized to have an L1 norm of 1. The arm-specific rewards for each arm are sampled uniformly from [0,0.5].

---

### Decision · Program_Chairs · 2023-09-21

**Decision:**

Accept (poster)

**Comment:**

This paper studies the contextual bandit problem under model misspecification and with access to regression oracles.
The proposed algorithm can be instantiated with a tradeoff parameter that interpolates between optimizing the cumulative regret or the last-iterate simple regret. For the second objective, it is the first optimal contextual bandit algorithm that is efficient under the regression oracle assumption.

The reviewers generally agree that this is a solid work and it is a clear accept.

The reason why this is not above a poster is that it is well known, see Bandit algorithms (Lattimore and Szepesvari, Chapter 33) that there is an unavoidable trade-off between cumulative regret and simple regret in bandits. The lower bound presented here is folklore knowledge for me (at least for K-armed bandits). The reason why the literature studies the settings in isolation, is because 1) the trade-off curve does not contain too many surprising insights, 2) nor is it especially hard to obtain a point on the trade-off curve.